# On the Paradox of Generalizable Logical Reasoning in Large Language Models

## Abstract

The emergent few-shot reasoning capabilities of Large Language Models (LLMs) have excited the natural language and machine learning community over recent years. Despite the numerous successful applications, it remains an open question whether LLMs have generalizable logical reasoning abilities. In this work, we expose a surprising failure of generalization in logical reasoning tasks (deduction, induction, and abduction)—when semantics are decoupled from the language reasoning process (*i.e.*, replacing semantic words with pure symbols), LLMs tend to perform much worse. We hypothesize that the learned *semantics* of language tokens do the most heavy lifting during the reasoning process but fail to imitate the basic formal reasoning abilities of humans. Furthermore, we also attempt to fine-tune Llama-2 on pure symbolic reasoning tasks to narrow the gap. However, the results indicate that FT-Llama2 can utilize similar template matching to respond to reasoning queries, but it falls short of generalizing to novel logic rules. These surprising observations question whether modern LLMs have mastered the inductive, deductive, and abductive reasoning abilities as in human intelligence, and motivate research on unveiling the magic existing within the black-box LLMs and evaluating and improving language models' reasoning abilities.

## 1 Introduction

Logical reasoning, a mental activity that aims to arrive at a conclusion using evidence, arguments, and logic in a rigorous way (Seel, 2011; Honderich, 1995), is fundamental for the discovery of new knowledge and explainability. It is a key aspect for the development of problem solving, decision making, and critical thinking (Bronkhorst et al., 2020; Poole et al., 1987; Dowden, 1993; McCarthy, 1959). Neural Logical Machine (Dong et al., 2019) firstly proposed a neural-symbolic architecture to conduct first-order reasoning. However, its application is restricted to specific domains and it relies on manually defined logical rules. Recently, Large Language Models (LLMs) also explore the similar direction by introducing a external tool or symbolic solver (Pan et al., 2023; Schick et al., 2023; Gao et al., 2023). Despite the impressive performance on a variety of natural language tasks, *can LLMs themselves perform like generalizable neural logical machines?*

We define the ability of *generalizable logical reasoning* as two featured aspects: i) using rules of inference (*e.g.*, deductive, inductive and abductive reasoning as in Fig. 1) to derive valid conclusions given premises; and ii) the "true" logical reasoning ability that can generalize across facts, rules, domains, and representations, regardless of what the premises are and in which forms the premises are presented. Previous works (Rae et al., 2022; Liu et al., 2023) have carried out evaluations on logical reasoning tasks with different large language models (*e.g.*, ChatGPT, GPT-4 (OpenAI, 2023) and RoBERTa (Liu et al., 2019) fine-tuning) and concluded that LLMs have limitation in natural language inference tasks requiring logical reasoning. Various benchmarks on logical reasoning (Yang et al., 2022; Saparov & He, 2022; Han et al., 2022) have also been proposed to challenge LLMs. However, the basic problem they have in common is that the in-context prompts they input are all in natural language representations. These representations contain rich semantic[1] information, creating strong connections among tokens, which might help LLMs to compose a superficial logical chain (shortcut) instead of really performing the formal reasoning process (Elazar et al., 2021; McCoy

---

[1] Unless otherwise specified, the term "semantics" used in this paper refers to linguistic semantics.

et al., 2019), preventing us from evaluating their true (generalizable) logical reasoning ability. Ideally, a general-intelligence agent is expected to be able to master generalizable logical reasoning, meaning they can reason with any given information to solve new, unfamiliar problems, regardless of any prior knowledge or different knowledge representations (Zhang et al., 2022; Singley & Anderson, 1989). Inspired by this, in this work, we examine the logical reasoning abilities of LLMs thoroughly by investigating the following questions:

**Question1:** Can pre-trained LLMs perform logical reasoning agnostic to semantics?

**Question2:** Can fine-tuned LLMs achieve generalizable logical reasoning?

The examination process is formally organized as follows. **First**, we systematically study the in-context reasoning ability of state-of-the-art pre-trained LLMs by decoupling the semantics from the language reasoning process, *i.e.*, we replace words with semantics with pure symbols. We test three kinds of reasoning abilities (*i.e.*, deduction, induction, abduction) on the two different settings (semantics, symbols) to study whether there is a gap in performance (§3.1). If LLMs master generalizable logical reasoning, there should be a little gap between the two settings. **Secondly,** we fine-tune LLMs on pure symbolic reasoning tasks to study whether fine-tuning on symbols helps narrow the gap, or even helps LLMs really master logical reasoning by generalizing to arbitrary new facts, rules, and domains, regardless of whether the knowledge is presented in semantic or symbolic forms (§3.2). Fig. 3 illustrates our ideas. Our extensive experiments reveal that (1) LLMs tend to perform significantly worse when semantics are decoupled, indicating that LLMs might rely on semantic representations to create superficial logical chains (shortcuts), instead of really performing the formal reasoning process; and (2) although fine-tuning achieves shallow generalization, *i.e.*, leveraging template matching to generalize to unseen facts, it fails to generalize to unseen logic rules—that is, LLMs, even heavily fine-tuned on semantics-decoupled data, still fall short of really mastering human's generalizable logical reasoning abilities.

Our analysis uncovers the paradox of generalizable logical reasoning in LLMs and questions whether modern LLMs have mastered the logical reasoning abilities as in human intelligence. We hope that our findings can provide a novel perspective on the evaluation of LLMs' reasoning abilities and inspire further research on unveiling the magic inside the black-box LLMs.

## 2 RELATED WORKS

**Logical reasoning abilities of LLMs**   Logical reasoning is of social importance and a great proportion of NLP tasks require logical reasoning. Prior work contextualizes the problem of logical reasoning by proposing reasoning- dependent datasets and studies solving the tasks with neural models (Johnson et al., 2017; Sinha et al., 2019; Yu et al., 2020; Wald et al., 2021; Tian et al., 2021; Han et al., 2022; Tafjord et al., 2020; Saparov & He, 2022). However, most studies focus on solving a single task, and the datasets either are designed for a specific domain or are rich in semantics, which indicating LLMs can rely on shallow semantic association for prediction rather than truly formal logical reasoning. There has been also some work to explore the effect of semantics on reasoning. For example, Dasgupta et al. (2022) evaluate three logical reasoning tasks, namely natural language inference (NLI), syllogisms and Wason selection based on whether the content of the problem is aligned with prior knowledge, concluding that LLMs show human-like content effects on reasoning. Schlegel et al. (2022) also reach the similar conclusion. Wei et al. (2023b) investigate the effects of semantic priors and input-label mapping on in-context learning using different-scale and instruction-tuned models. Wu et al. (2023) conducts "counterfactual" task variants and suggests that current LMs often rely on narrow, non-transferable procedures for task-solving. These findings highlight the gap in reasoning performance under different semantic settings and verify our conclusions. In fact, we attempted to narrow this gap by fine-tuning language models, but we found that LLMs still fall short of generalizing to completely new rules and facts, indicating that current language models still struggle to truly master logical reasoning abilities.

**Solving logical reasoning tasks**   Another line studies leveraging LLMs to solve complex logical reasoning problems. For example, "chain-of-thought (CoT)" (Wei et al., 2022b; Wang et al., 2022; Kojima et al., 2022) is proposed to facilitate models to generate a reasoning path that decomposes complex reasoning into multiple easier steps. Creswell et al. (2022) solve multi-step reasoning tasks by interacting between selection and inference to generate immediate reasoning steps. This sig-

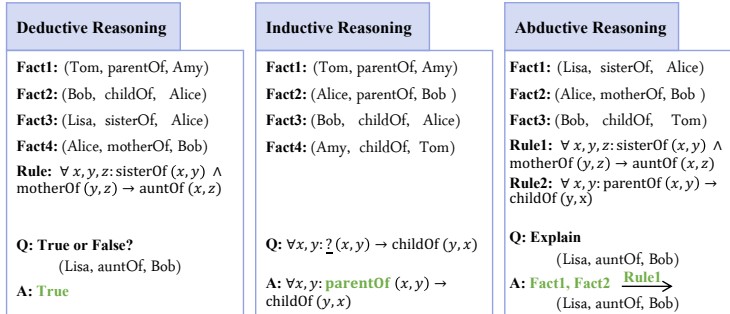

Figure 1: Task Definitions. **Deductive**: predicting the correctness of the predicted fact given rules and facts. **Inductive**: generating a rule based on multiple facts with similar patterns. **Abductive**: explaining the predicted fact based on given rules and facts.

nificantly improves the performance on arithmetic, commonsense, and symbolic reasoning benchmarks (Roy & Roth, 2016; Talmor et al., 2018; Geva et al., 2021; Wei et al., 2022b). Wei et al. (2023a) enhance the reasoning capabilities of a language model by fine-tuning it with input-label mappings that are unrelated to semantic priors. Besides, LLMs' reasoning abilities are closely related to in-context learning (ICL). ICL refers to the ability of language models to adapt and learn from a few prompt examples during the inference process. There has been a focus on exploring how to improve the performance of ICL (Su et al., 2022; Sanh et al., 2022; Wei et al., 2022a) and study the underlying mechanisms of ICL (Liu et al., 2021; Akyürek et al., 2022; Webson & Pavlick, 2022)

**Symbolic Reasoning** Symbolic reasoning has long been studied in the field of artificial intelligence (Boole, 1847; McCarthy, 1960; Fuhr, 2000; Eiter et al., 2009; Yi et al., 2018) and cognitive science (McCarthy & Hayes, 1981; Lavrac & Dzeroski, 1994; Newell & Simon, 2007; Landy et al., 2014; Fowler, 2021). It involves manipulating symbols and applying logical rules to perform deduction (Johnson-Laird, 1999), induction (Lavrac & Dzeroski, 1994), and abduction (Kovács & Spens, 2005). Recently, there has been some work to explore LLMs' ability of symbolic reasoning. Qian et al. (2022) evaluate a set of simple symbolic manipulation tasks to uncover the difficulty of the LLMs in handling (out-of-distribution) OOD symbolic generalization, highlighting the limitation in arithmetic and symbolic induction. BIG-Bench (Srivastava et al., 2022) contain the symbol_interpretation task aimed at reasoning and interpreting a simple scene consisting of some defined objects. However, we focus on decoupling semantics to compare the pure symbolic logical reasoning capabilities of LLMs with semantics setting. Furthermore, there is some work to use symbols (*e.g.*, code) can improve reasoning to a certain extent Shin et al. (2018); Gao et al. (2023) explore using LLM-based models for program synthesis to improve LLMs' reasoning abilities. Lample & Charton (2019) propose a framework to combine deep learning with symbolic mathematics to perform algebraic reasoning, equation solving, and theorem proving. Pallagani et al. (2022) use LLMs for automated planning. Gaur & Saunshi (2023) replace all occurrences of numbers in the problem statement with newly introduced variables to better evaluate the faithfulness of reasoning.

## 3 SYSTEMATIC EVALUATION OF GENERALIZABLE LOGICAL REASONING

To begin, we first introduce the definition of logical reasoning and provide task descriptions for each. More details are provided in Appendix G.

**Logical reasoning** is a type of problem-solving that involves working through a set of rules that govern a scenario (Wason & Johnson-Laird, 1972; Wason, 2007; Fagin et al., 2004; Walton, 1990). As an umbrella term, it encompasses a variety of aspects, including:

- *Deductive reasoning* is a logical process in which a conclusion can be derived from given premises or principles, meaning predicting new facts based on existing facts and logical rules. For example, given the two facts (Lisa, sisterOf, Alice) and (Alice, motherOf, Bob) along with a logical rule $\forall x, y, z : \text{sisterOf}(x, y) \land \text{motherOf}(y, z) \rightarrow \text{auntOf}(x, z)$, the new fact (Lisa, auntOf, Bob) can be derived through deductive reasoning. The task is to predict the True/False of a predicted fact given facts and rules. The accuracy is the proportion of correct predictions.

- *Inductive reasoning* involves making generalizations based on specific observations or evidence. In other words, Given multiple facts with similar patterns and a rule template, the goal is to induce a rule that entails these facts. For instance, given a set of observations that person A is the parent of person B and person B is the child of person A, inductive reasoning is to generate the logical rule $\forall x, y : \text{parentOf}(x, y) \rightarrow \text{childOf}(y, x)$. We perform the *rule generation* task. The precision is the proportion of generated rules exactly matching grounding truth rules.

- *Abductive reasoning* is a logical process of seeking a hypothesis that best fits or explains a set of observations. For example, given facts (Lisa, sisterOf, Alice) and (Alice, motherOf, Bob), along with logical rule $\forall x, y, z : \text{sisterOf}(x, y) \wedge \text{motherOf}(y, z) \rightarrow \text{auntOf}(x, z)$, if we observe Lisa is Bob's aunt, one possible explanation is that Lisa is Alice's sister and Alice is Bob's mother. We use *explanation generation* to evaluate the abductive reasoning ability. Given a *theory* including facts and logical rules, the task is to select specific facts and a logical rule from the given theory to explain the *observation*. We use Proof Accuracy (PA) as an evaluation metric, *i.e.*, the fraction of examples where the generated proof matches exactly any of the gold proofs.

## 3.1 PARADOX 1: LLMs ARE IN-CONTEXT SEMANTIC REASONERS RATHER THAN SYMBOLIC REASONERS

### 3.1.1 DECOUPLING SEMANTICS FROM IN-CONTEXT REASONING

To examine whether pre-trained LLMs can perform logical reasoning agnostic to semantics (*i.e.*, *Q1*), we decouple semantics from the in-context reasoning process and compare the performance gap between two different settings. Specifically, we evaluate LLMs on two easily-parseable datasets, Symbolic Trees (Hohenecker & Lukasiewicz, 2020) and ProofWriter (Tafjord et al., 2020), which need to infer the unknown facts by selecting relevant facts and logical rules.

**Symbolic Tree** is an artificially close-world and noise-free symbolic dataset generated with family-domain logical rules. Each generated tree shares the same set of logic rules. Given randomly sampled "*basic facts*", including gender information and "parentOf" relations, the datasets allow for reasoning about other different types of family relations (*e.g.*, auntOf), as *inferred facts*. Note that Symbolic Tree is a close-world dataset, which means that any facts not presented in the dataset are assumed to be false. Thus, we construct the false facts by replacing the head or tail entities with a random entity as negative examples. More descriptions are provided in Appendix J. **ProofWriter** (Tafjord et al., 2020) provide artificial facts and rules expressed in natural language. For our experiments, we use a subset of the ProofWriter Open World Assumption (OWA) dataset with a depth of 1, 2, 3 and 5 (there is no depth 4 task), which contains many small rulebases of facts and rules, expressed in English and do not exist in LLMs' knowledge base.

To decouple the semantics within the dataset, we replace the relation names (such as "parent") with hand-crafted symbols (*e.g.*, "r1", "r2", ...) or replace the entities names (such as "Alice") with entity ID (*e.g.*, "e1", "e2",...). Fig. 2 provides an illustrative example. During the symbol generation process, we also try to randomly sample some letters as relation names (*e.g.*, "lnqgv" instead of "r1"), but we observe that LLMs struggle to understand garbled characters, which may negatively affect performance (further discussion is provided in Appendix O).

**Experiment setup**  We primarily evaluate the performance of ChatGPT and GPT-4 by using the API provided by OpenAI, with a temperature of 0 to generate output. Additionally, we set the frequency penalty to zero and top p to 1. We also conduct human studies to evaluate human performance on these reasoning tasks (we only choose the most difficult setting, zero-shot *Symbols* for human). Please refer to Appendix H for the details.

**Baseline**  We consider zero-shot, zero-shot CoT, few-shot CoT and zero-plus-few-shot-CoT as baselines. To generate explanations for few-shot CoT experiments, for deductive reasoning, we use zero-shot CoT (*i.e.*, Let's think step by step) to generate explanations given the random questions; for abductive reasoning, we randomly select five examples and manually design their demonstrations. We provide all prompts and CoT demonstrations in Appendix A.

**Evaluation**  We use the accuracy of various tasks as the reasoning result, including deducing the correctness of a conclusion, inducing correct rules, or finding correct explanations for hypotheses. For the two settings, we refer to the raw data, where semantics are retained, as *Semantics*. When semantics are decoupled using symbols, we refer to this setting as *Symbols*. For the Symbolic Tree dataset, we experiment with 10 sampled trees and report the average results, where facts and rules

Table 1: The Reasoning Accuracy of Symbolic Tree. Results are in %.

| Category | Model | Baseline | deduction | induction | abduction |
|----------|-------|----------|-----------|-----------|-----------|
| **Symbols** | **ChatGPT** | Zero-Shot | 52.6±2.24 | 6.10±3.66 | 1.50±0.77 |
| | | Zero-Shot-CoT | 55.7±2.11 | 7.86±4.74 | 4.90±1.56 |
| | | Few-Shot-CoT | 54.8±1.40 | - | 18.2±2.28 |
| | | Zero-Plus-Few-Shot-CoT | 55.7±1.96 | - | 16.8±4.3 |
| | **GPT-4** | Zero-Shot | 68.8 | 9.28±2.37 | 25.0 |
| | | Zero-Shot-CoT | 71.1 | 8.93±5.59 | 31.2 |
| | | Few-Shot-CoT | 67.6 | - | 44.2 |
| | **Human** | Zero-Shot | 93.9 | 60.6 | 68.2 |
| **Semantics** | **ChatGPT** | Zero-Shot | 66.1±2.86 | 36.4±5.27 | 2.94±0.77 |
| | | Zero-Shot-CoT | 65.5±1.10 | 32.2±4.51 | 3.40±1.18 |
| | | Few-Shot-CoT | 67.1±3.47 | - | 21.8±3.28 |
| | | Zero-Plus-Few-Shot-CoT | 67.2±2.61 | - | 20.9±3.48 |
| | **GPT-4** | Zero-Shot | 79.2 | 52.5±2.28 | 27.3 |
| | | Zero-Shot-CoT | 86.2 | 53.9±2.48 | 33.4 |
| | | Few-Shot-CoT | 91.1 | - | 69.2 |
| | **Random** | - | 50.1±1.48 | 3.57 | - |

can be represented as logical language and natural language text as the input of LLMs. For example, the fact "motherOf(Alice, Bob)" can be represented as "Alice is Bob's mother"; the rule "$\forall x, y :$ parentOf$(x, y) \rightarrow$ childOf$(y, x)$" can be represented as "If x is parent of y, then y is child of x.". Through numerous trials, we find that for the *Symbols* setting, LLMs tend to perform better when using logic language representations. Conversely, for the *Semantics* setting, LLMs tend to perform better when using natural language representations. We select the representation that yields better performance for each setting. Additional results are presented in Appendix N.

**Results** From Tab. 1, we observe that in all reasoning scenarios, *Symbols* setting significantly underperforms *Semantics* setting. In other words, LLMs show significantly worse performance when semantics are decoupled. Specifically, in the deductive and abductive experiments, *Symbols* achieves approximately 25% lower absolute accuracy compared to *Semantics* setting while in the inductive scenarios, the performance declination goes up to 40%. In contrast, the human performance on symbolic deductive reasoning tasks stands at a 94% accuracy, far surpassing the 67.6% presented by GPT-4. These phenomena answer our question *Q1*—pre-trained LLMs fail to perform logical reasoning agnostic to semantics. The failure indicates that pre-trained LLMs fail to invoke the basic formal reasoning abilities of human, but instead may rely on the shallow semantic associations for prediction. We list our main observations as follows:

*(1) Induction and abduction under-perform deduction:* We compare the reasoning abilities of LLMs across induction and abduction tasks and find that they perform notably worse compared to deduction, regardless of whether semantics or symbols are used. When semantics are decoupled, the drop in performance is even more significant. These findings highlight the considerable room for improvement in LLMs' reasoning abilities.

Table 2: The deduction accuracy of ProofWriter (ChatGPT). Results are in %.

| Category | Baseline | depth-1 | depth-2 | depth-3 | depth-5 |
|----------|----------|---------|---------|---------|---------|
| **Symbols** | Zero-Shot | 69.1 | 62.3 | 59.4 | 52.8 |
| | Zero-Shot-CoT | 56.2 | 49.4 | 45.2 | 38.6 |
| | Few-Shot-CoT | 65.8 | 58.1 | 57.8 | 45.9 |
| **Semantics** | Zero-Shot | 69.0 | 63.5 | 60.3 | 51.4 |
| | Zero-Shot-CoT | 51.5 | 45.8 | 40.3 | 30.9 |
| | Few-Shot-CoT | 62.5 | 56.7 | 56.9 | 47.8 |

*(2) Shorter in-context knowledge enhances reasoning performance:* To examine the influence of context length on reasoning, we conducted an abductive reasoning experiment using a smaller Symbolic Tree, containing approximately 12 entities and 100 facts. The results, provided in Appendix Q, show that abductive reasoning with a shorter context leads to better performance compared to a longer context. Besides, we also conduct deduction and induction experiments where LLMs are directly provided with the relevant facts related to the predicted fact or the predicted rule. The results are presented in Appendix L. This finding suggests that LLMs struggle with processing excessively long in-context information, particularly in reasoning tasks. The length of the context influences reasoning performance, as shorter contexts make it easier to select relevant and useful information while minimizing the impact of unrelated content.

*(3) Effectiveness of commonsense expressed in natural language:* We explore the representation of knowledge in natural language and logic language forms in our experiments. The results, presented in Appendix N, indicate that for tasks involving semantics, natural language descriptions are more effective than logical language representations. Conversely, for symbolic and counter-commonsense tasks, logic language performs better. This observation suggests that natural language representations better stimulate the semantic understanding capabilities of LLMs, while logical language representations are more conducive to symbolic reasoning.

*(5) Utilizing internal knowledge outperforms external in-context knowledge:* To explore the ability of LLMs to utilize internal and external knowledge, we conduct an additional experiment where we provide LLMs with only the relevant facts related to the predicted fact. We compare the performance of *Removing rules* (leveraging internal knowledge) with *Semantics* (providing external logical rules). Surprisingly, we find that *Removing rules* performed better than *Semantics*. This suggests that LLMs possess the necessary internal knowledge to support answering questions and reasoning tasks, and leveraging this internal knowledge is more effective for reasoning than relying on external logical rules. Detailed results and case studies can be found in Appendix L.1.

**Analysis about Semantics** The aforementioned experiments offer initial evidence highlighting the significance of semantics in the reasoning of LLMs. To further investigate this observation, we hypothesize the influence to logical reasoning abilities comes from prior knowledge stored within semantics representation. Specifically, we test it by exploring three aspects:

Table 3: Semantics, Removing rules/facts and Counter-Commonsense reasoning experiments (ChatGPT and GPT-4). Results are in %.

|  | deductive (Few-Shot-CoT) | | inductive (Zero-Shot-CoT) | |
|---|---|---|---|---|
|  | ChatGPT | GPT-4 | ChatGPT | GPT-4 |
| Semantics | 71.8 | 90.0 | 25.0 | 53.6 |
| Symbols | 53.7 | 67.6 | 7.14 | 21.4 |
| Remove R/F | 70.1 | 90.4 | 7.14 | 35.7 |
| Counter-CS | 48.9 | 73.4 | 7.14 | 17.8 |

*When semantics are consistent with prior knowledge?* First, we examine the influence of prior knowledge stored within LLMs on their semantic reasoning performance. To achieve this, we remain the semantics (as semantics can encompass prior knowledge) and remove all given logical rules (in deduction) and facts (in induction). Please refer to Appendix A for prompts. This forces the LLMs to rely solely on their prior commonsense knowledge to infer the answers and allows us to assess the extent to which LLMs can leverage their internal knowledge to reason effectively without explicit in-context knowledge. As shown in Tab. 3, in the deductive reasoning experiment, *Removing rules/facts* achieves comparable results to *Semantics*; in the inductive reasoning experiment, *Removing rules/facts* outperforms *Symbols*, achieving 35.7% in GPT-4. These findings suggest that LLMs can perform deductive reasoning comparably by leveraging their stored commonsense knowledge without using the provided semantic knowledge.

Given a set of rules and facts, you have to reason whether a statement is true or false. Here are some facts and rules:

The bear likes the dog.
The cow is round.
The cow likes the bear.
The cow needs the bear.
The dog needs the squirrel.
The dog sees the cow.
The squirrel needs the dog.
If someone is round then they like the squirrel.
If the bear is round and the bear likes the squirrel then the squirrel needs the bear.
If the cow needs the dog then the cow is cold.

Does it imply that the statement "The cow likes the squirrel." is True?

Given a set of rules and facts, you have to reason whether a statement is true or false. Here are some facts and rules:

The e4 likes the e5.
The e14 is e2.
The e14 likes the e4.
The e14 needs the e4.
The e5 needs the e26.
The e5 sees the e14.
The e26 needs the e5.
If someone is e2 then they like the e26.
If the e4 is e2 and the e4 likes the e26 then the e26 needs the e4.
If the e14 needs the e5 then the e14 is e1.

Does it imply that the statement "The e14 likes the e26." is True?

Figure 2: Decoupling semantics from the ProofWriter task. In the original ProofWriter task, entities are represented by their names (left). However, in our decoupled setting, we replace the entity names with unique entity IDs (right).

*When semantics are not consistent with prior knowledge?* There has been some work pointing out LLMs struggle with reasoning with premises that are inconsistent with commonsense (Dasgupta et al., 2022; Saparov & He, 2022). Here, we undertake a similar study by introducing counter-commonsense logical rules. We implement it by shuffling relations as new relation labels to construct a new counter-commonsense dataset. As shown in Tab. 3, in comparison to *Semantics* and *Symbols*, we find that *Counter-Commonsense* performs worse than *Semantics*, even *Symbols*. These findings

Figure 3: Generalization of logical reasoning

suggest that when the in-context new knowledge conflicts with commonsense, LLMs struggle to accurately reason and predict.

*When semantics are irrelevant to prior knowledge?* We use the ProofWriter tasks to test whether unmeaningful semantics are still useful. The results are shown in Tab. 2. The results reveal that *Symbols* setting performs comparably to the *Semantics* setting in the zero-shot setting, suggesting that when semantics are irrelevant to commonsense, they have little effect on the reasoning abilities of LLMs. In other words, when the task does not require deep semantic understanding or relies minimally on commonsense knowledge, the presence or absence of semantics does not significantly impact the performance of LLMs. However, in the CoT settings, we observe that *Semantics* is significantly worse than *Symbols*. This might be because step-by-step reasoning magnifies the disturbing effect brought by weird semantics such as "The squirrel needs the dog". Additionally, we observe that the CoT settings even perform worse than the zero-shot setting, with a higher frequency of the answer "Cannot be determined.". Similar phenomenons are also observed in Tab. 1, indicating that CoT may not be always helpful for reasoning tasks with in-context new knowledge.

## 3.2 Paradox 2: LLMs rely on template matching for prediction but struggle to generalize to novel logic rules

We have studied the in-context reasoning abilities of pre-trained LLMs in Section 3.1, and showed that there is a huge performance decline when semantic words are replaced with symbols, indicating that state-of-the-art LLMs do not really invoke a similar reasoning mechanism like human. However, one may wonder whether this infeasibility to generalize to symbolic setting is because the adopted LLMs were not pre-trained on such symbolic data. In this section, using deduction tasks as the test bed, we fine-tune a Llama2-13b-chat model on our symbolic datasets. The results suggest that supervised fine-tuning indeed greatly narrows down the performance gap between symbolic and semantic settings. However, does this mean that fine-tuning on symbolic data can enable *generalizable logical reasoning* (Q2)? Our experiments, unfortunately, give a firm "no" as the answer. Specifically, we found LLMs solely leverage a *template matching* mechanism to generalize to unseen facts, rather than really executing the formal reasoning process. When generalizing to unseen logical rules, the performance of the fine-tuned model significantly drops, indicating that generalizable logical reasoning is still difficult to achieve for today's LLMs.

To test our hypotheses, we use Symbolic Tree, ProofWriter, RuDaS (Cornelio & Thost, 2020) and FOLIO (Han et al., 2022). RuDaS is a synthetic symbolic dataset containing facts and rules. FOLIO is a complex and diverse dataset for logical reasoning in natural language. We continue to use accuracy as the evaluation metric (predicting the correctness of unknown facts).

### 3.2.1 FT-LMs sometimes struggle with learning to reason

In this section, we aim to examine the difficulty and efficiency of learning to reason by fine-tuning Llama2-13B-chat model on different datasets (two versions of Symbolic Tree and RuDaS) to evaluate the training and testing accuracy. Training accuracy shows the fitting capability of model. Testing accuracy shows the in-domain generalization ability.

Given that language models are mostly pre-trained on a substantial corpus with rich language semantics, we initially fine-tune the Llama2 model on the *Semantics* version of Symbolic Tree to align with the pre-training representations. We select five sampled Symbolic Trees for fine-tuning and another three for testing. Note that the testing trees share the same logical rules as the training ones but the facts differ. Details are in Appendix H. Refer to the blue lines (train and test on *Semantics* version of Symbolic Tree) in Fig. 4 for our findings: (1) the solid line converges slowly, indicating that only after adequate fine-tuning epochs (about 13 epochs), the FT-model can completely fit the

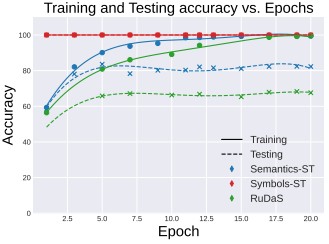 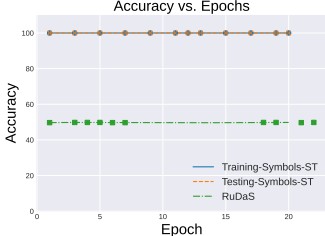 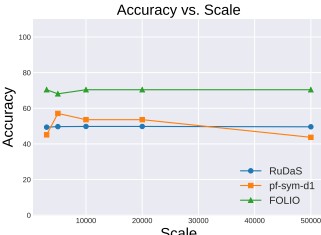

Figure 4: Training (solid lines) and testing (dashed lines) accuracy at each epoch. Colors denote different datasets.

Figure 5: Accuracy of different testing data at different epochs. Colors denote different datasets.

Figure 6: Accuracy on different testing data with different fine-tuning data scale. Colors denote different datasets

Table 4: The Symbolic Tree results of fine-tuning on Llama2-13B-chat with *Symbols* version of Symbolic Tree. Results are in %.

|  | FT-Llama2 |
|---|---|
| training | 100 |
| testing | 100 |
| ent_words | 100 |
| rel_commonsense | 100 |
| rel_counter | 87.4 |
| all_other_symbol | 100 |
| all_words | 92.4 |
| Symbols (natural language) | 50.7 |

Table 5: The generalization accuracy of Symbolic Tree tasks (Llama2-13B-chat) in %.

|  | FT-Llama2 | Random |
|---|---|---|
| RuDaS | 49.4 | 50.3 |
| pf-sym-d1 | 39.2 | 50.2 |
| pf-sym-d2 | 34.2 | 49.8 |
| pf-sym-d1 w/o negation | 35.4 | 50.0 |
| pf-sym-d2 w/o negation | 45.1 | 50.3 |
| pf-sem-d1 | 55.1 | 50.1 |
| pf-sem-d2 | 52.5 | 49.6 |
| FOLIO | 70.4 | 33.3 |

training data with approximately 100% accuracy; (2) there is a gap between the solid and dashed lines, meaning that FT-Llama2 struggles with generalizing to unseen facts.

Similarly, we also use the *Symbols* version of Symbolic Tree to fine-tune Llama2-13B-chat. We still use five trees for training and the other three for testing. Refer to the red lines in Fig. 4: (1) Comparing red solid line and blue solid line, we find that fine-tuning on *Symbols* coverages faster than *Semantics*, achieving 100% training accuracy with one single epoch. This is probably because symbolic representations are indeed more abstract and concise than semantic data, and are easier to fit. (2) The red solid line and the red dashed line overlap (achieving 100% accuracy), indicating the fine-tuned Llama2 on *Symbols* Tree can achieve generalization to unseen facts.

In addition, we also train and test on RuDaS, which contains more predicates and entities. Refer to the green solid lines in Fig. 4. Compared to *Semantics* (blue solid line) and *Symbols* (red solid line) versions of Symbolic Tree, fine-tuning Llama2 on RuDaS is more difficult and the gap between the training (green sold line) and testing accuracy (green dashed line) is wider. These observations reveal that FT-LMs sometimes struggle with learning to reason from complex reasoning datasets. As training epochs increase, FT-model rolls over the same data repeatedly, which is likely to perform memorizing rather than learning, thus showing worse performance on generalization.

### 3.2.2 FT-LMs LEVERAGE TEMPLATE MATCHING FOR PREDICTION

We have found that fine-tuning Llama2 with *Symbols* version of Symbolic Tree can generalize to unseen facts (in symbolic forms), but fine-tuning on *Semantics* version fails to generalize to unseen facts (in semantic forms). In this section, we further answer the question whether fine-tuning LMs on *Symbols* can generalize to *Semantics* setting. Thus, we still use the *Symbols* version of five trees as the fine-tuning data. To evaluate on *Semantics* setting, we replace entities or relations with their original semantic words. For example, we replace the "r1" of fact "r1(e1, e2)" with "parentOf", denoted by *rel_commonsense*; replace "e1" of fact "r1(e1, e2)" with "Amy", namely *ent_words*. Descriptions of other replacements are in Appendix T. The results in Tab. 4 indicate that whatever replacing entities and relation with, FT-models perform well, especially in *rel_commonsense*, *enl_words* and *all_other_symbols*, achieving 100% accuracy.

In fact, the good performance is not from its really performing the formal reasoning process but from template matching. For example, when the model memorizes the question-answer pair "given r1(e1, e2) fact and rule r1(A, B) → r2(B, A), we have r2(e2, e1)", when prompted with "given parentOf(e1, e2) fact and rule parentOf(A, B) → childOf(B, A)", the model can first match the template by mapping "r1, r2" to "parentOf, childOf", and then leverage the memorized pattern "given r1(e1, e2) fact and rule r1(A, B) → r2(B, A), we have r2(e2, e1)" to derive "childOf(e2, e1)". Notably, *rel_counter* is a little worse than other replacements, suggesting that even a fine-tuned model is still distracted by prior knowledge. More interestingly, when the question is represented as natural language text (*Symbols (natural language))* instead of logical form, the performance dramatically declines, indicating that when it is not easy to leverage template matching, the performance can drop to random guessing. In fact, LLMs cannot comprehend that "r1(a,b)" and "a is r1 of b" are equivalent. This observation also questions whether supervised fine-tuning can enable *generalizable logic reaosning* (Q2). To further examine it, we proceed with another evaluation as below.

### 3.2.3 FT-LMs is difficult to generalize to novel logic rules

In this section, we evaluate the fine-tuned Llama2 (*Symbols* version of Symbolic Tree as fine-tuning data) on other datasets: ProofWriter, RuDaS and FOLIO. For ProofWriter, we use its *Symbols* version (*e.g.*, *pf-sym-d1*). To ensure that LLMs are not grappling with unfamiliar knowledge, we eliminate questions that contain "not" (*e.g.*, *pf-sym-d1 w/o negation*). We also evaluate on the *Semantic* version (*e.g.*, *pf-sem-d1*).

The results are presented in Tab. 5. We first compare *Symbols* version of these datasets (*RuDaS*, *pf-sym-d1*, *pf-sym-d2*, *pf-sym-d1 w/o negation*, *pf-sym-d2 w/o negation*) and observe that FT-Llama2 struggles to perform well, even producing results worse than random guess. This indicates that fine-tuned Llama2 using *Symbols* datasets is difficult to generalize to novel symbolic rules. Besides, when FT-Llama2 perform *Semantic* version tasks (*pf-sem-d1*, *pf-sem-d2* and FOLIO), the performance is not good. Although the results of FOLIO surpass ramdom,

Table 6: The results of RuDaS and FO-LIO (Llama2-13B-chat). Results are in %.

|  | Llama2 | FT-Llama2 |
|---|---|---|
| RuDaS | 52.3 | 49.4 |
| FOLIO | 72.4 | 70.4 |

its performance is inferior to the non-fine-tuned LLaMA2-13B-chat (Tab. 6). Moreover, we also try to extend training time and scale up training data to further improve the logical reasoning abilities of FT-Llama2 (both using *Symbols* version of Symbolic Tree as fine-tuning data). Refer to Fig. 5: The training accuracy still maintains 100% while the testing accuracy on RuDaS is still near random. Similarly, we scale up the fine-tuning data and test on RuDaS, *pf-sym-d1* and FOLIO (Fig. 6), yet find there is hardly any improvement. These additional evidences highlight that fine-tuning LLMs struggle to achieve generalizable logic reasoning (Q2).

## 4 Conclusion

Our paper presents a comprehensive investigation of generalizable logical reasoning in Large Language Models. First, we systematically study the in-context abilities of LLMs by decoupling semantics from in-context prompts. Secondly, we further fine-tune LLMs on pure symbolic reasoning tasks to improve logic reasoning abilities. The extensive experiments reveal two paradoxes: (1) LLMs tend to perform significantly worse when semantics are decoupled, indicating that LLMs might rely on semantic associations for prediction instead of really performing the formal reasoning process; (2) although fine-tuning achieves shallow generalization, it fails to generalize to novel logic rules. In summary, LLMs still fails to really master human's generalizable logical reasoning abilities. These findings inspire further research on unveiling the magic existing within the black-box LLMs.

**Limitation and Future Work** Despite our thorough experiments to gauge the generalizable logical reasoning of LLMs, it may not be precisely reflected due to several factors, including the limitedness of the size and diversity of symbolic reasoning datasets for fine-tuning, a lack of adequate training duration, and the potential for more efficiently prompts to further stimulate the model's abilities.

## Reproducibility Statement

For the empirical results, our synthetic datasets are clearly described in Appendix J. Architecture, implementations and hyperparameters are in Appendix H. We intend to release our code as open source prior to publication.

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

# Table of Contents

# A  PROMPTS

## A.1  DEDUCTIVE REASONING

### A.1.1  ZERO-SHOT

```
system: You are a helpful assistant with deductive reasoning abilities.
user: I will provide a set of logical rules L1 to L{number of rules} and facts F1
    to F{number of basic facts}. Please select one single logical rule from L1 to
    L{number of rules} and a few facts from F1 to F{number of basic facts} to
    predict True/False of the unknown fact using deductive reasoning.
Logical rules: {logical rules}
Facts: {basic facts}
Unknown fact: {statement}
The answer (True or False) is:
```

### A.1.2  ZERO-SHOT-CoT

```
system: You are a helpful assistant with deductive reasoning abilities. Please
    select one single logical rule and a few facts to predict True/False of the
    following statement.
user: I will provide a set of logical rules L1 to L{number of rules} and facts F1
    to F{number of basic facts}. Please select one single logical rule from L1 to
    L{number of rules} and a few facts from F1 to F{number of basic facts} to
    predict True/False of the following statement using deductive reasoning.
Logical rules: {logical rules}
Facts: {basic facts}
Statement: {statement}
Answer with True or False? Let's think step by step.
```

### A.1.3  FEW-SHOT-CoT

```
system: You are a helpful assistant with deductive reasoning abilities. Please
    select one single logical rule and a few facts to predict True/False of the
    following statement.
user:I will provide a set of logical rules L1 to L{number of rules} and facts F1
    to F{number of basic facts}.
Logical rules: {logical rules}
Facts: {basic facts}
Please select one single logical rule from L1 to L{number of rules} and a few
    facts from F1 to F{number of basic facts} to predict True/False of the
    following statement using deductive reasoning.
{Demonstrations}
Statement: {statement}
Answer:
```

### A.1.4  EXAMPLES OF FEW-SHOT-CoT DEMONSTRATIONS

Statement: $r14$(Amelie, Jonathan)
Answer: We can use L11: $\forall A, B, C, D, E : r3(B, A) \wedge r3(B, C) \wedge r3(C, D) \wedge r3(D, E) \wedge r2(A) \to r14(A, E)$ to make a deduction. From the facts, we know that $r3$(Amelie, Thomas) and $r3$(Thomas, Jonathan). Therefore, we can apply L11 with A = Amelie, B = Thomas, C = Jonathan, D and E as variables. This gives us: $r3$(Thomas, Jonathan) $\wedge$ $r3$(Amelie, Thomas) $\wedge$ $r3$(Thomas, D) $\wedge$ $r3$(D, E) $\wedge$ $r2$(Amelie) $\to r14$(Amelie, E) Since we know that $r2$(Amelie) is true from F7, we can simplify the above statement to: $r3$(Thomas, Jonathan) $\wedge$ $r3$(Amelie, Thomas) $\wedge$ $r3$(Thomas, D) $\wedge$ $r3$(D, E) $\to r14$(Amelie, E) Now, we can substitute the facts $r3$(Amelie, Thomas) and $r3$(Thomas, Jonathan) to get: $r3$(Thomas, Jonathan) $\wedge$ $r3$(Thomas, D) $\wedge$ $r3$(D, E) $\to r14$(Amelie, E) We can see that this statement is true if we choose D = Alina and E = Jonathan, since we know that $r3$(Thomas, Alina) from F50 and $r3$(Alina, Jonathan) from F56. Therefore, the statement $r14$(Amelie, Jonathan) is true.

Statement: $r31$(Philipp, Nina)
Answer: Let's use L28: $\forall A, B, C, D : r3(B, A) \wedge r3(C, B) \wedge r3(C, D) \wedge r1(A) \to r31(A, D)$ to see if we can prove the statement $r31$(Philipp, Nina). From the facts, we know that $r3$(Luisa, Nina) (F52) and $r3$(Emilia, Philipp) (F62). We can use L28 with $A = Philipp, B = Emilia, C = Luisa$,

and $D = Nina$ to get: $r3$(Emilia, Philipp) $\wedge$ $r3$(Luisa, Emilia) $\wedge$ $r3$(Luisa, Nina) $\wedge$ $r1$(Philipp) $\rightarrow$ $r31$(Philipp, Nina) Since all the conditions are met, we can conclude that $r31$(Philipp, Nina) is True. Therefore, the answer is True.

Statement: $r8$(Leonie, Nico)
Answer: We can use L5 to make a deduction about the statement $r8$(Leonie, Nico). L5 states that if there exist three individuals A, B, and C such that A is related to B, B is related to C, and A has a certain property (r2), then A is related to C in a certain way (r8). Using this rule, we can see that we have the following facts: - $r3$(Leonie, Emily)
- $r3$(Emily, Nico)
Therefore, we can conclude that $r8$(Leonie, Nico) is true.

Statement: $r7$(Patrick, Alina)
Answer: We can use L4: $\forall A, B : r3(A, B) \wedge r1(A) \rightarrow r7(A, B)$ to determine if $r7$(Patrick, Alina) is true or false. From the facts, we know that $r3$(Patrick, Alina) is true (F55). We also know that $r1$(Patrick) is true (F9). Therefore, we can apply L4 to conclude that $r7$(Patrick, Alina) is true. Answer: True.

Statement: $r27$(Jonathan, Leonie)
Answer: Let's use L4: $\forall A, B : r3(A, B) \wedge r1(A) \rightarrow r7(A, B)$ and F56: $r3$(Patrick, Jonathan) to infer that $r7$(Patrick, Jonathan) is true. Then, we can use L24: $\forall A, B, C : r3(B, A) \wedge r3(C, B) \wedge r1(A) \rightarrow r27(A, C)$ and F44: $r3$(Leonie, Emily) to infer that $r27$(Jonathan, Leonie) is false, since there is no fact that supports $r3$(Jonathan, Emily). Therefore, the answer is False.

### A.1.5 ZERO-SHOT OF REMOVING RULES SETTING

```
system: Please answer the question only with True or False.
user: I will provide a set of facts. Please predict True/False of the unknown fact
    based on given facts.
Facts: {facts}
Unknown fact: {statement}
The answer (True or False) is:
```

## A.2 INDUCTIVE REASONING

### A.2.1 ZERO-SHOT

```
system: You are a helpful assistant with inductive reasoning abilities. Please
    generate one single rule to match the template and logically entail the facts.
     Note that the symbol '##' in the template should be filled with either 'r1'
    or 'r45', while the symbol '++' should be filled with either 'r43' or 'r44'.
user: I will give you a set of facts F1 to F{number of basic facts}, facts G1 to G
    {number of inferred fact} and a template for a logical rule. Please generate
    one single rule to match the template and logically entail the facts G1 to G{
    number of inferred fact} based on facts F1 to F{number of basic facts}.
Facts: {facts}
Template: {rule template}
Note that the symbol '##' in the template should be filled with either 'r1' or '
    r45', while the symbol '++' should be filled with either 'r43' or 'r44'.
After filling in the template, the generated rule is:
```

### A.2.2 ZERO-SHOT COT

```
system: You are a helpful assistant with inductive reasoning abilities. Please
    generate one single rule to match the template and logically entail the facts.
     Note that the symbol '##' in the template should be filled with either 'r1'
    or 'r45', while the symbol '++' should be filled with either 'r43' or 'r44'.
user: I will give you a set of facts F1 to F{number of basic facts}, facts G1 to G
    {number of inferred fact} and a template for a logical rule. Please generate
    one single rule to match the template and logically entail the facts G1 to G{
    number of inferred fact} based on facts F1 to F{number of basic facts}.
Facts: {facts}
Template: {rule template}
Note that the symbol '##' in the template should be filled with either 'r1' or '
    r45', while the symbol '++' should be filled with either 'r43' or 'r44'.
After filling in the template, the generated rule is: Let's think step by step.
```

### A.2.3 Zero-Shot of removing facts setting

```
system: Please generate one single rule to match the template. Note that the
    symbol '##' in the template should be filled with either 'parent' or 'child',
    while the symbol '++' should be filled with either 'male' or 'female'.
user: I will give you a template for a logical rule. Please generate one single
    rule to match the template and logically infer the relation sister
Template: If A is ## of B and B is ## of C and A is ++, then A is sister of C.
Note that the symbol '##' in the template should be filled with either 'parent' or
    'child', while the symbol '++' should be filled with either 'male' or 'female
    '.
After filling in the template, the generated rule is:
```

## A.3 Abductive reasoning

### A.3.1 Zero-Shot

```
system: You are a helpful assistant with abductive reasoning abilities. Please
    select one single logical rule and a few facts to explain the following
    statement.
user: I will provide a set of logical rules L1 to L{number of rules} and facts F1
    to F{number of basic facts}. Please select one single logical rule from L1 to
    L{number of rules} and a few facts from F1 to F{number of basic facts} to
    explain the following statement.
Rules: {logical rules}
Facts: {basic facts}
Statement: {statement}
Answer with the numbers of the selected rule and facts. The selected rule and
    facts are:
```

### A.3.2 Zero-Shot-CoT

```
system: You are a helpful assistant with abductive reasoning abilities. Please
    select one single logical rule and a few facts to explain the following
    statement.
user: I will provide a set of logical rules L1 to L{number of rules} and facts F1
    to F{number of basic facts}. Please select one single logical rule from L1 to
    L{number of rules} and a few facts from F1 to F{number of basic facts} to
    explain the following statement.
Rules: {logical rules}
Facts: {basic facts}
Statement: {statement}
Answer with the numbers of the selected rule and facts. The selected rule and
    facts are: Let's think step by step.
```

### A.3.3 Few-Shot-CoT prompt

```
system: You are a helpful assistant with abductive reasoning abilities. Please
    select one single logical rule and a few facts to explain the following
    statement.
user:
  I will provide a set of logical rules L1 to L{number of rules} and facts F1 to
      F{number of basic facts}. Please select one single logical rule from L1 to
      L{number of rules} and a few facts from F1 to F{number of basic facts} to
      explain the following statement.
  Rules: {logical rules}
  Facts: {basic facts}
  {Demonstrations}
  Statement: {statement}
  Answer:
```

### A.3.4 Examples of Few-Shot-CoT Demonstrations

Statement: $r4$(Laura, Fabian)
Answer: To explain the statement $r4$(Laura, Fabian), we can use the rule L3 and fact F1 and F37.
L3: $\forall A, B : r1(A, B) \land r44(A) \to r4(A, B)$

F1: $r1$(Laura,Fabian)
F37: $r44$(Laura)
According to L3, if there is a relationship r1 between A and B, and A has the property r44, then the relationship r4 holds between A and B. In this case, we have a relationship r1 between Laura and Fabian (F1) and Laura has the property r44 (F37). Therefore, the relationship r4 holds between Laura and Fabian, which is the given statement.
The selected rule and facts are: L3, F1, and F37.

Statement: $r5$(Stefan, Emily)
Answer: To explain the statement $r5$(Stefan, Emily), we can use the rule L4 and fact F17. L4: $\forall A, B : r1(A, B) \land r43(A) \to r5(A, B)$
F17: $r1$(Stefan,Emily)
According to rule L4, if there is a relation r1(A, B) and A has the property r43, then the relation r5(A, B) holds. In our case, we have the fact F17, which states that there is a relation r1(Stefan, Emily). Additionally, Stefan has the property r43 (F61). Therefore, we can conclude that the relation r5(Stefan, Emily) holds.
The selected rule and facts are: L4, F17, and F61.

Statement: $r23$(Felix, Elias)
Answer: To explain the statement $r23$(Felix, Elias), we can use the rule L22 and fact F5. L22: $\forall A, B : r1(B, A) \land r43(A) \to r23(A, B)$
F5: $r1$(Elias,Felix)
According to rule L22, if there exists a relationship $r1$(B, A) and A has the property $r43$, then the relationship $r23$(A, B) holds. In our case, we have the fact F5, which states that there is a relationship $r1$(Elias, Felix). Additionally, we know that Felix has the property $r43$ (F59). Therefore, applying rule L22, we can conclude that the relationship $r23$(Felix, Elias) holds.
The selected rule and facts are: L22, F5, and F59.

Statement: $r3$(Felix, Fabian)
Answer: To explain the statement $r3$(Felix, Fabian), we can use the rule L2 and facts F2, F1, and F59.
L2: $\forall A, B, C : r1(B, A) \land r1(B, C) \land r43(A) \to r3(A, C)$
F1: $r1$(Laura,Fabian)
F2: $r1$(Laura,Felix)
F59: $r43$(Felix)
By substituting the values into L2, we get:
$r1$(Laura, Felix) $\land$ $r1$(Laura, Fabian) $\land$ $r43$(Felix) $\to r3$(Felix, Fabian)
Since all the premises are true according to the given facts, the conclusion $r3$(Felix, Fabian) is also true.
The selected rule and facts are: L2, F1, F2, and F59.

Statement: $r7$(Fabian, Helga)
Answer: To explain the statement $r7$(Fabian, Helga), we can use the rule L6 and facts F13, F21, and F39.
L6: $\forall A, B, C : r1(A, B) \land r1(B, C) \land r43(A) \to r7(A, C)$
F13: $r1$(Fabian,Thomas)
F21: $r1$(Thomas,Helga)
F39: $r43$(Fabian)
According to L6, if there exist A, B, and C such that $r1(A, B), r1(B, C)$, and $r43(A)$, then $r7(A, C)$ holds. In our case, A = Fabian, B = Thomas, and C = Helga. Since we have $r1$(Fabian,Thomas) from F13, $r1$(Thomas,Helga) from F21, and $r43$(Fabian) from F39, we can conclude that $r7$(Fabian, Helga) is true.
So, the selected rule and facts are: L6, F13, F21, and F39.

### A.3.5 FEW-SHOT COT OF REMOVING RULES SETTING

```
system: You are a helpful assistant with abductive reasoning abilities. Please
    select a few facts to explain the following statement.
user: I will provide a set of facts F1 to F63. Please select a few facts from F1
    to F63 to explain the following statement.
Facts: {facts}
```

```
Statement: Laura is mother of Felix.
Answer: To explain the statement "Laura is mother of Felix", we can use Facts:
Fact F2 states: Laura is parent of Felix.
Fact F37 states: Laura is female.
Using F2 and F37, we can conclude that "Laura is mother of Felix" holds.
Therefore, the selected rule and facts are F2, F37.

Statement: Samuel is brother of Alina.
Answer: To infer the statement "Samuel is brother of Alina", we have:
F27: Patrick is parent of Samuel.
F28: Patrick is parent of Alina.
F47: Samuel is male.
Based on these facts, we can infer "Samuel is brother of Alina":
Therefore, the selected rule and facts are F27, F28, F47.

Statement: Patrick is grandfather of David.
Answer: To explain the statement "Patrick is grandfather of David", we have:
F28: Patrick is parent of Alina.
F7: Alina is parent of David.
F45: Patrick is male.
Based on these facts, we can infer "Patrick is grandfather of David":
Therefore, the selected rule and facts are F28, F7, F45.

Statement: Amelie is daughter of Elena.
Answer: To explain the statement "Amelie is daughter of Elena", we have:
F20: Elena is parent of Amelie.
F43: Amelie is female.
Based on these facts, we can infer "Amelie is daughter of Elena".
Therefore, the selected rule and facts are F20, F43.

Statement: Claudia is sister of Felix
Answer: To prove the statement "Claudia is sister of Felix", we can use facts:
F3: Laura is parent of Claudia.
F2: Laura is parent of Felix.
F40: Claudia is female.
Based on these facts, we can infer "Claudia is sister of Felix".
Therefore, the selected rule and facts are F3, F2, F40.

Statement: Laura is mother of Fabian.
Answer:
```

# B  DEDUCTION EXAMPLES OF SYMBOLIC TREE DATASETS

In this section, we provide examples of deduction experiments conducted on the Symbolic Tree datasets. We present examples for both the *Semantics* and *Symbols* settings, represented in both natural language text and logic language

## B.1  SEMANTICS

### B.1.1  LOGIC LANGUAGE REPRESENTATIONS

```
Logical rules:
L1: $\forall A,B,C: parentOf(B, A) \land parentOf(B, C) \land female(A) \
    rightarrow sisterOf(A,C)$
L2: $\forall A,B,C: parentOf(B, A) \land parentOf(B, C) \land male(A) \rightarrow
    brotherOf(A,C)$
L3: $\forall A,B: parentOf(A, B) \land female(A) \rightarrow motherOf(A,B)$
L4: $\forall A,B: parentOf(A, B) \land male(A) \rightarrow fatherOf(A,B)$
L5: $\forall A,B,C: parentOf(A, B) \land parentOf(B, C) \land female(A) \
    rightarrow grandmotherOf(A,C)$
L6: $\forall A,B,C: parentOf(A, B) \land parentOf(B, C) \land male(A) \rightarrow
    grandfatherOf(A,C)$
L7: $\forall A,B,C,D: parentOf(A, B) \land parentOf(B, C) \land parentOf(C, D) \
    land female(A) \rightarrow greatGrandmotherOf(A,D)$
L8: $\forall A,B,C,D: parentOf(A, B) \land parentOf(B, C) \land parentOf(C, D) \
    land male(A) \rightarrow greatGrandfatherOf(A,D)$
L9: $\forall A,B,C,D: parentOf(B, A) \land parentOf(B, C) \land parentOf(C, D) \
    land female(A) \rightarrow auntOf(A,D)$
```

```
L10: $\forall A,B,C,D: parentOf(B, A) \land parentOf(B, C) \land parentOf(C, D) \
    land male(A) \rightarrow uncleOf(A,D)$
L11: $\forall A,B,C,D,E: parentOf(B, A) \land parentOf(B, C) \land parentOf(C, D)
    \land parentOf(D, E) \land female(A) \rightarrow greatAuntOf(A,E)$
L12: $\forall A,B,C,D,E: parentOf(B, A) \land parentOf(B, C) \land parentOf(C, D)
    \land parentOf(D, E) \land male(A) \rightarrow greatUncleOf(A,E)$
L13: $\forall A,B,C,D,E,F: parentOf(B, A) \land parentOf(C, B) \land parentOf(C, D
    ) \land parentOf(D, E) \land parentOf(E, F) \land female(A) \rightarrow
    secondAuntOf(A,F)$
L14: $\forall A,B,C,D,E,F: parentOf(B, A) \land parentOf(C, B) \land parentOf(C, D
    ) \land parentOf(D, E) \land parentOf(E, F) \land male(A) \rightarrow
    secondUncleOf(A,F)$
L15: $\forall A,B,C,D,E: parentOf(B, A) \land parentOf(C, B) \land parentOf(C, D)
    \land parentOf(D, E) \land female(A) \rightarrow girlCousinOf(A,E)$
L16: $\forall A,B,C,D,E: parentOf(B, A) \land parentOf(C, B) \land parentOf(C, D)
    \land parentOf(D, E) \land male(A) \rightarrow boyCousinOf(A,E)$
L17: $\forall A,B,C,D,E,F,G: parentOf(B, A) \land parentOf(C, B) \land parentOf(D,
    C) \land parentOf(D, E) \land parentOf(E, F) \land parentOf(F, G) \land
    female(A) \rightarrow girlSecondCousinOf(A,G)$
L18: $\forall A,B,C,D,E,F,G: parentOf(B, A) \land parentOf(C, B) \land parentOf(D,
    C) \land parentOf(D, E) \land parentOf(E, F) \land parentOf(F, G) \land male(
    A) \rightarrow boySecondCousinOf(A,G)$
L19: $\forall A,B,C,D,E,F: parentOf(B, A) \land parentOf(C, B) \land parentOf(D, C
    ) \land parentOf(D, E) \land parentOf(E, F) \land female(A) \rightarrow
    girlFirstCousinOnceRemovedOf(A,F)$
L20: $\forall A,B,C,D,E,F: parentOf(B, A) \land parentOf(C, B) \land parentOf(D, C
    ) \land parentOf(D, E) \land parentOf(E, F) \land male(A) \rightarrow
    boyFirstCousinOnceRemovedOf(A,F)$
L21: $\forall A,B: parentOf(B, A) \land female(A) \rightarrow daughterOf(A,B)$
L22: $\forall A,B: parentOf(B, A) \land male(A) \rightarrow sonOf(A,B)$
L23: $\forall A,B,C: parentOf(B, A) \land parentOf(C, B) \land female(A) \
    rightarrow granddaughterOf(A,C)$
L24: $\forall A,B,C: parentOf(B, A) \land parentOf(C, B) \land male(A) \rightarrow
    grandsonOf(A,C)$
L25: $\forall A,B,C,D: parentOf(B, A) \land parentOf(C, B) \land parentOf(D, C) \
    land female(A) \rightarrow greatGranddaughterOf(A,D)$
L26: $\forall A,B,C,D: parentOf(B, A) \land parentOf(C, B) \land parentOf(D, C) \
    land male(A) \rightarrow greatGrandsonOf(A,D)$
L27: $\forall A,B,C,D: parentOf(B, A) \land parentOf(C, B) \land parentOf(C, D) \
    land female(A) \rightarrow nieceOf(A,D)$
L28: $\forall A,B,C,D: parentOf(B, A) \land parentOf(C, B) \land parentOf(C, D) \
    land male(A) \rightarrow nephewOf(A,D)$

Facts:
F1: female(Laura)
F2: male(Elias)
F3: male(Fabian)
F4: female(Claudia)
F5: female(Elena)
F6: male(Thomas)
F7: female(Amelie)
F8: female(Luisa)
F9: male(Patrick)
F10: female(Emilia)
F11: male(Samuel)
F12: female(Alina)
F13: male(Jonathan)
F14: male(Philipp)
F15: male(Nico)
F16: male(David)
F17: female(Emily)
F18: male(Konstantin)
F19: male(Florian)
F20: female(Helga)
F21: female(Nina)
F22: female(Lea)
F23: male(Felix)
F24: female(Leonie)
F25: male(Stefan)
F26: male(Gabriel)
F27: male(Tobias)
F28: parentOf(Laura, Fabian)
```

```
F29: parentOf(Laura, Felix)
F30: parentOf(Laura, Claudia)
F31: parentOf(Elias, Fabian)
F32: parentOf(Elias, Felix)
F33: parentOf(Elias, Claudia)
F34: parentOf(Alina, David)
F35: parentOf(Alina, Lea)
F36: parentOf(Nico, David)
F37: parentOf(Nico, Lea)
F38: parentOf(Emily, Nico)
F39: parentOf(Konstantin, Nico)
F40: parentOf(Fabian, Thomas)
F41: parentOf(Fabian, Amelie)
F42: parentOf(Nina, Tobias)
F43: parentOf(Leonie, Emily)
F44: parentOf(Stefan, Emily)
F45: parentOf(Gabriel, Tobias)
F46: parentOf(Elena, Thomas)
F47: parentOf(Elena, Amelie)
F48: parentOf(Thomas, Helga)
F49: parentOf(Thomas, Nina)
F50: parentOf(Thomas, Patrick)
F51: parentOf(Luisa, Helga)
F52: parentOf(Luisa, Nina)
F53: parentOf(Luisa, Patrick)
F54: parentOf(Patrick, Samuel)
F55: parentOf(Patrick, Alina)
F56: parentOf(Patrick, Jonathan)
F57: parentOf(Patrick, Philipp)
F58: parentOf(Patrick, Florian)
F59: parentOf(Emilia, Samuel)
F60: parentOf(Emilia, Alina)
F61: parentOf(Emilia, Jonathan)
F62: parentOf(Emilia, Philipp)
F63: parentOf(Emilia, Florian)

Unknown fact: boyCousinOf(Tobias, David)
```

### B.1.2 NATURAL LANGUAGE REPRESENTATIONS

```
Logical rules:
L1: If B is parent of A and B is parent of C and A is female, then A is sister of
    D.
L2: If B is parent of A and B is parent of C and A is male, then A is brother of D
    .
L3: If A is parent of B and A is female, then A is mother of C.
L4: If A is parent of B and A is male, then A is father of C.
L5: If A is parent of B and B is parent of C and A is female, then A is
    grandmother of D.
L6: If A is parent of B and B is parent of C and A is male, then A is grandfather
    of D.
L7: If A is parent of B and B is parent of C and C is parent of D and A is female,
     then A is greatGrandmother of E.
L8: If A is parent of B and B is parent of C and C is parent of D and A is male,
    then A is greatGrandfather of E.
L9: If B is parent of A and B is parent of C and C is parent of D and A is female,
     then A is aunt of E.
L10: If B is parent of A and B is parent of C and C is parent of D and A is male,
    then A is uncle of E.
L11: If B is parent of A and B is parent of C and C is parent of D and D is parent
     of E and A is female, then A is greatAunt of F.
L12: If B is parent of A and B is parent of C and C is parent of D and D is parent
     of E and A is male, then A is greatUncle of F.
L13: If B is parent of A and C is parent of B and C is parent of D and D is parent
     of E and E is parent of F and A is female, then A is secondAunt of G.
L14: If B is parent of A and C is parent of B and C is parent of D and D is parent
     of E and E is parent of F and A is male, then A is secondUncle of G.
L15: If B is parent of A and C is parent of B and C is parent of D and D is parent
     of E and A is female, then A is girlCousin of F.
L16: If B is parent of A and C is parent of B and C is parent of D and D is parent
     of E and A is male, then A is boyCousin of F.
```

```
L17: If B is parent of A and C is parent of B and D is parent of C and D is parent
     of E and E is parent of F and F is parent of G and A is female, then A is
     girlSecondCousin of H.
L18: If B is parent of A and C is parent of B and D is parent of C and D is parent
     of E and E is parent of F and F is parent of G and A is male, then A is
     boySecondCousin of H.
L19: If B is parent of A and C is parent of B and D is parent of C and D is parent
     of E and E is parent of F and A is female, then A is
     girlFirstCousinOnceRemoved of G.
L20: If B is parent of A and C is parent of B and D is parent of C and D is parent
     of E and E is parent of F and A is male, then A is boyFirstCousinOnceRemoved
     of G.
L21: If B is parent of A and A is female, then A is daughter of C.
L22: If B is parent of A and A is male, then A is son of C.
L23: If B is parent of A and C is parent of B and A is female, then A is
     granddaughter of D.
L24: If B is parent of A and C is parent of B and A is male, then A is grandson of
     D.
L25: If B is parent of A and C is parent of B and D is parent of C and A is female
     , then A is greatGranddaughter of E.
L26: If B is parent of A and C is parent of B and D is parent of C and A is male,
     then A is greatGrandson of E.
L27: If B is parent of A and C is parent of B and C is parent of D and A is female
     , then A is niece of E.
L28: If B is parent of A and C is parent of B and C is parent of D and A is male,
     then A is nephew of E.

Facts:
F1: Laura is female.
F2: Elias is male.
F3: Fabian is male.
F4: Claudia is female.
F5: Elena is female.
F6: Thomas is male.
F7: Amelie is female.
F8: Luisa is female.
F9: Patrick is male.
F10: Emilia is female.
F11: Samuel is male.
F12: Alina is female.
F13: Jonathan is male.
F14: Philipp is male.
F15: Nico is male.
F16: David is male.
F17: Emily is female.
F18: Konstantin is male.
F19: Florian is male.
F20: Helga is female.
F21: Nina is female.
F22: Lea is female.
F23: Felix is male.
F24: Leonie is female.
F25: Stefan is male.
F26: Gabriel is male.
F27: Tobias is male.
F28: Laura is parent of Fabian.
F29: Laura is parent of Felix.
F30: Laura is parent of Claudia.
F31: Elias is parent of Fabian.
F32: Elias is parent of Felix.
F33: Elias is parent of Claudia.
F34: Alina is parent of David.
F35: Alina is parent of Lea.
F36: Nico is parent of David.
F37: Nico is parent of Lea.
F38: Emily is parent of Nico.
F39: Konstantin is parent of Nico.
F40: Fabian is parent of Thomas.
F41: Fabian is parent of Amelie.
F42: Nina is parent of Tobias.
F43: Leonie is parent of Emily.
F44: Stefan is parent of Emily.
```

```
F45: Gabriel is parent of Tobias.
F46: Elena is parent of Thomas.
F47: Elena is parent of Amelie.
F48: Thomas is parent of Helga.
F49: Thomas is parent of Nina.
F50: Thomas is parent of Patrick.
F51: Luisa is parent of Helga.
F52: Luisa is parent of Nina.
F53: Luisa is parent of Patrick.
F54: Patrick is parent of Samuel.
F55: Patrick is parent of Alina.
F56: Patrick is parent of Jonathan.
F57: Patrick is parent of Philipp.
F58: Patrick is parent of Florian.
F59: Emilia is parent of Samuel.
F60: Emilia is parent of Alina.
F61: Emilia is parent of Jonathan.
F62: Emilia is parent of Philipp.
F63: Emilia is parent of Florian.

Unknown fact: Gabriel is uncle of Lea.
```

## B.2 SYMBOLIZATION

### B.2.1 LOGIC LANGUAGE REPRESENTATIONS

```
Logical rules:
L1: $\forall A,B,C: r3(B, A) \land r3(B, C) \land r2(A) \rightarrow r4(A, C)$
L2: $\forall A,B,C: r3(B, A) \land r3(B, C) \land r1(A) \rightarrow r5(A, C)$
L3: $\forall A,B: r3(A, B) \land r2(A) \rightarrow r6(A, B)$
L4: $\forall A,B: r3(A, B) \land r1(A) \rightarrow r7(A, B)$
L5: $\forall A,B,C: r3(A, B) \land r3(B, C) \land r2(A) \rightarrow r8(A, C)$
L6: $\forall A,B,C: r3(A, B) \land r3(B, C) \land r1(A) \rightarrow r9(A, C)$
L7: $\forall A,B,C,D: r3(A, B) \land r3(B, C) \land r3(C, D) \land r2(A) \
    rightarrow r10(A, D)$
L8: $\forall A,B,C,D: r3(A, B) \land r3(B, C) \land r3(C, D) \land r1(A) \
    rightarrow r11(A, D)$
L9: $\forall A,B,C,D: r3(B, A) \land r3(B, C) \land r3(C, D) \land r2(A) \
    rightarrow r12(A, D)$
L10: $\forall A,B,C,D: r3(B, A) \land r3(B, C) \land r3(C, D) \land r1(A) \
    rightarrow r13(A, D)$
L11: $\forall A,B,C,D,E: r3(B, A) \land r3(B, C) \land r3(C, D) \land r3(D, E) \
    land r2(A) \rightarrow r14(A, E)$
L12: $\forall A,B,C,D,E: r3(B, A) \land r3(B, C) \land r3(C, D) \land r3(D, E) \
    land r1(A) \rightarrow r15(A, E)$
L13: $\forall A,B,C,D,E,F: r3(B, A) \land r3(C, B) \land r3(C, D) \land r3(D, E) \
    land r3(E, F) \land r2(A) \rightarrow r16(A, F)$
L14: $\forall A,B,C,D,E,F: r3(B, A) \land r3(C, B) \land r3(C, D) \land r3(D, E) \
    land r3(E, F) \land r1(A) \rightarrow r17(A, F)$
L15: $\forall A,B,C,D,E: r3(B, A) \land r3(C, B) \land r3(C, D) \land r3(D, E) \
    land r2(A) \rightarrow r18(A, E)$
L16: $\forall A,B,C,D,E: r3(B, A) \land r3(C, B) \land r3(C, D) \land r3(D, E) \
    land r1(A) \rightarrow r19(A, E)$
L17: $\forall A,B,C,D,E,F,G: r3(B, A) \land r3(C, B) \land r3(D, C) \land r3(D, E)
     \land r3(E, F) \land r3(F, G) \land r2(A) \rightarrow r20(A, G)$
L18: $\forall A,B,C,D,E,F,G: r3(B, A) \land r3(C, B) \land r3(D, C) \land r3(D, E)
     \land r3(E, F) \land r3(F, G) \land r1(A) \rightarrow r21(A, G)$
L19: $\forall A,B,C,D,E,F: r3(B, A) \land r3(C, B) \land r3(D, C) \land r3(D, E) \
    land r3(E, F) \land r2(A) \rightarrow r22(A, F)$
L20: $\forall A,B,C,D,E,F: r3(B, A) \land r3(C, B) \land r3(D, C) \land r3(D, E) \
    land r3(E, F) \land r1(A) \rightarrow r23(A, F)$
L21: $\forall A,B: r3(B, A) \land r2(A) \rightarrow r24(A, B)$
L22: $\forall A,B: r3(B, A) \land r1(A) \rightarrow r25(A, B)$
L23: $\forall A,B,C: r3(B, A) \land r3(C, B) \land r2(A) \rightarrow r26(A, C)$
L24: $\forall A,B,C: r3(B, A) \land r3(C, B) \land r1(A) \rightarrow r27(A, C)$
L25: $\forall A,B,C,D: r3(B, A) \land r3(C, B) \land r3(D, C) \land r2(A) \
    rightarrow r28(A, D)$
L26: $\forall A,B,C,D: r3(B, A) \land r3(C, B) \land r3(D, C) \land r1(A) \
    rightarrow r29(A, D)$
L27: $\forall A,B,C,D: r3(B, A) \land r3(C, B) \land r3(C, D) \land r2(A) \
    rightarrow r30(A, D)$
```

```
L28: $\forall A,B,C,D: r3(B, A) \land r3(C, B) \land r3(C, D) \land r1(A) \
    rightarrow r31(A, D)$

Facts:
F1: $r2$(Laura)
F2: $r1$(Elias)
F3: $r1$(Fabian)
F4: $r2$(Claudia)
F5: $r2$(Elena)
F6: $r1$(Thomas)
F7: $r2$(Amelie)
F8: $r2$(Luisa)
F9: $r1$(Patrick)
F10: $r2$(Emilia)
F11: $r1$(Samuel)
F12: $r2$(Alina)
F13: $r1$(Jonathan)
F14: $r1$(Philipp)
F15: $r1$(Nico)
F16: $r1$(David)
F17: $r2$(Emily)
F18: $r1$(Konstantin)
F19: $r1$(Florian)
F20: $r2$(Helga)
F21: $r2$(Nina)
F22: $r2$(Lea)
F23: $r1$(Felix)
F24: $r2$(Leonie)
F25: $r1$(Stefan)
F26: $r1$(Gabriel)
F27: $r1$(Tobias)
F28: $r3$(Laura, Fabian)
F29: $r3$(Laura, Felix)
F30: $r3$(Laura, Claudia)
F31: $r3$(Elias, Fabian)
F32: $r3$(Elias, Felix)
F33: $r3$(Elias, Claudia)
F34: $r3$(Alina, David)
F35: $r3$(Alina, Lea)
F36: $r3$(Nico, David)
F37: $r3$(Nico, Lea)
F38: $r3$(Emily, Nico)
F39: $r3$(Konstantin, Nico)
F40: $r3$(Fabian, Thomas)
F41: $r3$(Fabian, Amelie)
F42: $r3$(Nina, Tobias)
F43: $r3$(Leonie, Emily)
F44: $r3$(Stefan, Emily)
F45: $r3$(Gabriel, Tobias)
F46: $r3$(Elena, Thomas)
F47: $r3$(Elena, Amelie)
F48: $r3$(Thomas, Helga)
F49: $r3$(Thomas, Nina)
F50: $r3$(Thomas, Patrick)
F51: $r3$(Luisa, Helga)
F52: $r3$(Luisa, Nina)
F53: $r3$(Luisa, Patrick)
F54: $r3$(Patrick, Samuel)
F55: $r3$(Patrick, Alina)
F56: $r3$(Patrick, Jonathan)
F57: $r3$(Patrick, Philipp)
F58: $r3$(Patrick, Florian)
F59: $r3$(Emilia, Samuel)
F60: $r3$(Emilia, Alina)
F61: $r3$(Emilia, Jonathan)
F62: $r3$(Emilia, Philipp)
F63: $r3$(Emilia, Florian)

Unknown fact: $r9$(Thomas, Claudia)
```

### B.2.2 NATURAL LANGUAGE REPRESENTATIONS:

```
Logical rules:
L1: If B is $r3$ of A and B is $r3$ of C and A is $r2$, then A is $r4$ of D.
L2: If B is $r3$ of A and B is $r3$ of C and A is $r1$, then A is $r5$ of D.
L3: If A is $r3$ of B and A is $r2$, then A is $r6$ of C.
L4: If A is $r3$ of B and A is $r1$, then A is $r7$ of C.
L5: If A is $r3$ of B and B is $r3$ of C and A is $r2$, then A is $r8$ of D.
L6: If A is $r3$ of B and B is $r3$ of C and A is $r1$, then A is $r9$ of D.
L7: If A is $r3$ of B and B is $r3$ of C and C is $r3$ of D and A is $r2$, then A
    is $r10$ of E.
L8: If A is $r3$ of B and B is $r3$ of C and C is $r3$ of D and A is $r1$, then A
    is $r11$ of E.
L9: If B is $r3$ of A and B is $r3$ of C and C is $r3$ of D and A is $r2$, then A
    is $r12$ of E.
L10: If B is $r3$ of A and B is $r3$ of C and C is $r3$ of D and A is $r1$, then A
    is $r13$ of E.
L11: If B is $r3$ of A and B is $r3$ of C and C is $r3$ of D and D is $r3$ of E
    and A is $r2$, then A is $r14$ of F.
L12: If B is $r3$ of A and B is $r3$ of C and C is $r3$ of D and D is $r3$ of E
    and A is $r1$, then A is $r15$ of F.
L13: If B is $r3$ of A and C is $r3$ of B and C is $r3$ of D and D is $r3$ of E
    and E is $r3$ of F and A is $r2$, then A is $r16$ of G.
L14: If B is $r3$ of A and C is $r3$ of B and C is $r3$ of D and D is $r3$ of E
    and E is $r3$ of F and A is $r1$, then A is $r17$ of G.
L15: If B is $r3$ of A and C is $r3$ of B and C is $r3$ of D and D is $r3$ of E
    and A is $r2$, then A is $r18$ of F.
L16: If B is $r3$ of A and C is $r3$ of B and C is $r3$ of D and D is $r3$ of E
    and A is $r1$, then A is $r19$ of F.
L17: If B is $r3$ of A and C is $r3$ of B and D is $r3$ of C and D is $r3$ of E
    and E is $r3$ of F and F is $r3$ of G and A is $r2$, then A is $r20$ of H.
L18: If B is $r3$ of A and C is $r3$ of B and D is $r3$ of C and D is $r3$ of E
    and E is $r3$ of F and F is $r3$ of G and A is $r1$, then A is $r21$ of H.
L19: If B is $r3$ of A and C is $r3$ of B and D is $r3$ of C and D is $r3$ of E
    and E is $r3$ of F and A is $r2$, then A is $r22$ of G.
L20: If B is $r3$ of A and C is $r3$ of B and D is $r3$ of C and D is $r3$ of E
    and E is $r3$ of F and A is $r1$, then A is $r23$ of G.
L21: If B is $r3$ of A and A is $r2$, then A is $r24$ of C.
L22: If B is $r3$ of A and A is $r1$, then A is $r25$ of C.
L23: If B is $r3$ of A and C is $r3$ of B and A is $r2$, then A is $r26$ of D.
L24: If B is $r3$ of A and C is $r3$ of B and A is $r1$, then A is $r27$ of D.
L25: If B is $r3$ of A and C is $r3$ of B and D is $r3$ of C and A is $r2$, then A
    is $r28$ of E.
L26: If B is $r3$ of A and C is $r3$ of B and D is $r3$ of C and A is $r1$, then A
    is $r29$ of E.
L27: If B is $r3$ of A and C is $r3$ of B and C is $r3$ of D and A is $r2$, then A
    is $r30$ of E.
L28: If B is $r3$ of A and C is $r3$ of B and C is $r3$ of D and A is $r1$, then A
    is $r31$ of E.

Facts:
F1: Laura is $r2$.
F2: Elias is $r1$.
F3: Fabian is $r1$.
F4: Claudia is $r2$.
F5: Elena is $r2$.
F6: Thomas is $r1$.
F7: Amelie is $r2$.
F8: Luisa is $r2$.
F9: Patrick is $r1$.
F10: Emilia is $r2$.
F11: Samuel is $r1$.
F12: Alina is $r2$.
F13: Jonathan is $r1$.
F14: Philipp is $r1$.
F15: Nico is $r1$.
F16: David is $r1$.
F17: Emily is $r2$.
F18: Konstantin is $r1$.
F19: Florian is $r1$.
F20: Helga is $r2$.
F21: Nina is $r2$.
```

```
F22: Lea is $r2$.
F23: Felix is $r1$.
F24: Leonie is $r2$.
F25: Stefan is $r1$.
F26: Gabriel is $r1$.
F27: Tobias is $r1$.
F28: Laura is $r3$ of Fabian.
F29: Laura is $r3$ of Felix.
F30: Laura is $r3$ of Claudia.
F31: Elias is $r3$ of Fabian.
F32: Elias is $r3$ of Felix.
F33: Elias is $r3$ of Claudia.
F34: Alina is $r3$ of David.
F35: Alina is $r3$ of Lea.
F36: Nico is $r3$ of David.
F37: Nico is $r3$ of Lea.
F38: Emily is $r3$ of Nico.
F39: Konstantin is $r3$ of Nico.
F40: Fabian is $r3$ of Thomas.
F41: Fabian is $r3$ of Amelie.
F42: Nina is $r3$ of Tobias.
F43: Leonie is $r3$ of Emily.
F44: Stefan is $r3$ of Emily.
F45: Gabriel is $r3$ of Tobias.
F46: Elena is $r3$ of Thomas.
F47: Elena is $r3$ of Amelie.
F48: Thomas is $r3$ of Helga.
F49: Thomas is $r3$ of Nina.
F50: Thomas is $r3$ of Patrick.
F51: Luisa is $r3$ of Helga.
F52: Luisa is $r3$ of Nina.
F53: Luisa is $r3$ of Patrick.
F54: Patrick is $r3$ of Samuel.
F55: Patrick is $r3$ of Alina.
F56: Patrick is $r3$ of Jonathan.
F57: Patrick is $r3$ of Philipp.
F58: Patrick is $r3$ of Florian.
F59: Emilia is $r3$ of Samuel.
F60: Emilia is $r3$ of Alina.
F61: Emilia is $r3$ of Jonathan.
F62: Emilia is $r3$ of Philipp.
F63: Emilia is $r3$ of Florian.

Unknown fact: Nico is $r27$ of Stefan.
```

## B.3 SEMANTICS OF REMOVING RULE SETTING

```
I will provide a set of facts. Please predict True/False of the unknown fact based
    on given facts.
Facts:
F1: Laura is female.
F2: Elias is male.
F3: Fabian is male.
F4: Claudia is female.
F5: Elena is female.
F6: Thomas is male.
F7: Amelie is female.
F8: Luisa is female.
F9: Patrick is male.
F10: Emilia is female.
F11: Samuel is male.
F12: Alina is female.
F13: Jonathan is male.
F14: Philipp is male.
F15: Nico is male.
F16: David is male.
F17: Emily is female.
F18: Konstantin is male.
F19: Florian is male.
F20: Helga is female.
F21: Nina is female.
```

```
F22: Lea is female.
F23: Felix is male.
F24: Leonie is female.
F25: Stefan is male.
F26: Gabriel is male.
F27: Tobias is male.
F28: Laura is parent of Fabian.
F29: Laura is parent of Felix.
F30: Laura is parent of Claudia.
F31: Elias is parent of Fabian.
F32: Elias is parent of Felix.
F33: Elias is parent of Claudia.
F34: Alina is parent of David.
F35: Alina is parent of Lea.
F36: Nico is parent of David.
F37: Nico is parent of Lea.
F38: Emily is parent of Nico.
F39: Konstantin is parent of Nico.
F40: Fabian is parent of Thomas.
F41: Fabian is parent of Amelie.
F42: Nina is parent of Tobias.
F43: Leonie is parent of Emily.
F44: Stefan is parent of Emily.
F45: Gabriel is parent of Tobias.
F46: Elena is parent of Thomas.
F47: Elena is parent of Amelie.
F48: Thomas is parent of Helga.
F49: Thomas is parent of Nina.
F50: Thomas is parent of Patrick.
F51: Luisa is parent of Helga.
F52: Luisa is parent of Nina.
F53: Luisa is parent of Patrick.
F54: Patrick is parent of Samuel.
F55: Patrick is parent of Alina.
F56: Patrick is parent of Jonathan.
F57: Patrick is parent of Philipp.
F58: Patrick is parent of Florian.
F59: Emilia is parent of Samuel.
F60: Emilia is parent of Alina.
F61: Emilia is parent of Jonathan.
F62: Emilia is parent of Philipp.
F63: Emilia is parent of Florian.

Unknown fact: Jonathan is aunt of Thomas.
The answer (True or False) is:
```

## C    EXAMPLES OF PROOFWRITER

In this section, we provide examples of deduction experiments conducted on the ProofWriter Depth-1 dataset. We present examples for both the *Semantics* and *Symbols* settings.

### C.1    SEMANTICS

```
The bear likes the dog.
The cow is round.
The cow likes the bear.
The cow needs the bear.
The dog needs the squirrel.
The dog sees the cow.
The squirrel needs the dog.
If someone is round then they like the squirrel.
If the bear is round and the bear likes the squirrel then the squirrel needs the
    bear.
If the cow needs the dog then the cow is cold.
Does it imply that the statement "The cow likes the squirrel." is True?

The bear likes the dog.
The cow is round.
```

```
The cow likes the bear.
The cow needs the bear.
The dog needs the squirrel.
The dog sees the cow.
The squirrel needs the dog.
If someone is round then they like the squirrel.
If the bear is round and the bear likes the squirrel then the squirrel needs the
    bear.
If the cow needs the dog then the cow is cold.
Does it imply that the statement "The cow does not like the squirrel." is True?
```

```
Bob is blue.
Erin is quiet.
Fiona is cold.
Harry is cold.
All quiet things are blue.
If Harry is blue then Harry is not young.
Blue things are young.
Blue, round things are cold.
If something is blue and not red then it is round.
If something is young then it is white.
If Erin is red and Erin is not round then Erin is young.
If Erin is red and Erin is not cold then Erin is white.
Does it imply that the statement "Erin is white" is True?
Answer with only True or False. The answer is:
```

```
The bear likes the dog.
The cow is round.
The cow likes the bear.
The cow needs the bear.
The dog needs the squirrel.
The dog sees the cow.
The squirrel needs the dog.
If someone is round then they like the squirrel.
If the bear is round and the bear likes the squirrel then the squirrel needs the
    bear.
If the cow needs the dog then the cow is cold.
Does it imply that the statement "The cow likes the squirrel." is True?
```

## C.2 SYMBOLS

```
The e4 likes the e5.
The e14 is e2.
The e14 likes the e4.
The e14 needs the e4.
The e5 needs the e26.
The e5 sees the e14.
The e26 needs the e5.
If someone is e2 then they like the e26.
If the e4 is e2 and the e4 likes the e26 then the e26 needs the e4.
If the e14 needs the e5 then the e14 is e1.
Does it imply that the statement "The e14 likes the e26." is True?
```

```
The e27 is e7.
The e27 is e15.
The e30 does not chase the e27.
The e30 eats the e27.
The e30 is e1.
The e30 is e15.
The e30 visits the e27.
If something visits the e27 then the e27 does not visit the e30.
If something is e1 and e15 then it visits the e30.
Does it imply that the statement "The e30 visits the e30." is True?
```

```
The e27 is e7.
The e27 is e15.
The e30 does not chase the e27.
The e30 eats the e27.
The e30 is e1.
```

```
The e30 is e15.
The e30 visits the e27.
If something visits the e27 then the e27 does not visit the e30.
If something is e1 and e15 then it visits the e30.
Does it imply that the statement "The e30 visits the e30." is True?
```

# D  SPECIFIC FINE-TUNING EXAMPLES

## D.1  EXAMPLE 1

```
Q: Given a set of rules and facts, you have to reason whether a statement is true
   or false. Here are some facts and rules:
F1: $p14$(c26,c16).
F2: $p12$(c11,c20).
F3: $p14$(c18,c3).
F4: $p7$(c27,c1).
F5: $p12$(c23,c29).
F6: $p3$(c29,c20).
F7: $p7$(c19,c6).
F8: $p7$(c18,c7).
F9: $p7$(c25,c13).
F10: $p12$(c15,c19).
F11: $p3$(c0,c10).
F12: $p11$(c11,c11).
F13: $p12$(c19,c18).
F14: $p1$(c0,c2).
F15: $p1$(c5,c17).
F16: $p3$(c24,c13).
F17: $p3$(c8,c30).
F18: $p7$(c24,c19).
F19: $p7$(c12,c15).
F20: $p9$(c1,c21).
F21: $p12$(c29,c3).
F22: $p14$(c7,c2).
F23: $p7$(c27,c8).
F24: $p7$(c20,c23).
F25: $p3$(c27,c23).
F26: $p3$(c19,c31).
F27: $p9$(c13,c13).
F28: $p11$(c18,c18).
F29: $p3$(c15,c24).
F30: $p9$(c2,c27).
F31: $p1$(c2,c4).
F32: $p9$(c26,c18).
F33: $p1$(c15,c18).
F34: $p3$(c1,c0).
F35: $p14$(c9,c23).
F36: $p11$(c27,c27).
F37: $p1$(c31,c11).
F38: $p9$(c17,c5).
F39: $p14$(c24,c21).
F40: $p3$(c10,c29).
F41: $p11$(c20,c20).
F42: $p9$(c27,c9).
F43: $p11$(c17,c17).
F44: $p11$(c2,c2).
F45: $p11$(c0,c0).
F46: $p12$(c16,c25).
F47: $p7$(c5,c22).
F48: $p1$(c24,c29).
F49: $p11$(c29,c29).
F50: $p14$(c30,c4).
F51: $p1$(c20,c5).
F52: $p12$(c6,c12).
F53: $p9$(c0,c4).
F54: $p3$(c8,c10).
F55: $p9$(c26,c12).
F56: $p7$(c9,c29).
F57: $p14$(c10,c15).
```

```
F58: $p1$(c10,c20).
F59: $p11$(c1,c1).
F60: $p7$(c15,c16).
F61: $p12$(c6,c4).
F62: $p12$(c8,c0).
F63: $p12$(c13,c17).
F64: $p14$(c8,c10).
F65: $p7$(c6,c11).
F66: $p9$(c31,c30).
F67: $p11$(c21,c21).
F68: $p12$(c7,c1).
F69: $p3$(c27,c27).
F70: $p1$(c7,c0).
F71: $p9$(c10,c31).
F72: $p7$(c10,c20).
F73: $p14$(c14,c3).
F74: $p9$(c29,c26).
F75: $p7$(c30,c2).
F76: $p12$(c16,c16).
F77: $p9$(c28,c10).
F78: $p3$(c21,c31).
F79: $p12$(c22,c26).
F80: $p7$(c7,c6).
F81: $p9$(c10,c6).
F82: $p14$(c22,c29).
F83: $p11$(c31,c31).
F84: $p3$(c27,c16).
F85: $p11$(c5,c5).
F86: $p9$(c4,c18).
F87: $p3$(c0,c11).
F88: $p14$(c15,c5).
F89: $p14$(c26,c8).
F90: $p14$(c26,c11).
F91: $p12$(c1,c30).
F92: $p11$(c4,c4).
F93: $p7$(c11,c16).
F94: $p1$(c21,c1).
F95: $p1$(c8,c31).
F96: $p12$(c2,c22).
F97: $p12$(c20,c26).
F98: $p1$(c22,c21).
F99: $p1$(c25,c27).
F100: $p14$(c8,c15).
F101: $p9$(c9,c10).

L1: $\forall X0,X1: p14(X1,X0) \rightarrow p6(X0,X1)$.
L2: $\forall X0,X1: p9(X1,X0) \rightarrow p10(X0,X1)$.
L3: $\forall X0,X1,X2: p12(X0,X2) \land p7(X1,X0) \rightarrow p0(X0,X1)$.
L4: $\forall X0,X1: p3(X1,X0) \rightarrow p5(X0,X1)$.
L5: $\forall X0,X1: p1(X0,X1) \land p11(X1,X1) \rightarrow p13(X0,X1)$.

Does it imply that the statement "$p13$(c2,c4)." is True?

A: True
```

## D.2 EXAMPLE 2

```
Q: Given a set of rules and facts, you have to reason whether a statement is true
   or false. Here are some facts and rules:
F1: r4($e116$).
F2: r4($e186$).
F3: r4($e84$).
F4: r2($e36$, $e32$).
F5: r2($e71$, $e56$).
F6: r2($e145$, $e186$).
F7: r4($e173$).
F8: r2($e108$, $e168$).
F9: r2($e21$, $e168$).
F10: r2($e139$, $e77$).
F11: r2($e31$, $e152$).
F12: r2($e74$, $e25$).
```

```
F13: r4($e29$).
F14: r2($e180$, $e50$).
F15: r2($e90$, $e1$).
F16: r4($e42$).
F17: r2($e14$, $e51$).
F18: r4($e35$).
F19: r2($e80$, $e146$).
F20: r4($e94$).
F21: r4($e83$).
F22: r2($e87$, $e186$).
F23: r2($e142$, $e158$).
F24: r2($e1$, $e140$).
F25: r4($e89$).
F26: r4($e127$).
F27: r4($e103$).
F28: r2($e46$, $e3$).
F29: r2($e58$, $e146$).
F30: r4($e22$).
F31: r2($e16$, $e7$).
F32: r2($e37$, $e24$).
F33: r2($e146$, $e152$).
F34: r4($e99$).
F35: r4($e51$).
F36: r2($e12$, $e173$).
F37: r4($e141$).
F38: r4($e111$).
F39: r2($e156$, $e103$).
F40: r4($e181$).
F41: r4($e55$).
F42: r4($e170$).
F43: r2($e59$, $e6$).
F44: r4($e45$).
F45: r4($e40$).
F46: r4($e161$).
F47: r4($e12$).
F48: r2($e178$, $e27$).
F49: r2($e176$, $e36$).
F50: r4($e139$).
F51: r2($e91$, $e32$).
F52: r2($e110$, $e65$).
F53: r2($e24$, $e161$).
F54: r2($e159$, $e76$).
F55: r4($e58$).
F56: r2($e22$, $e99$).
F57: r2($e75$, $e173$).
F58: r4($e120$).
F59: r4($e74$).
F60: r4($e39$).
F61: r4($e158$).
F62: r2($e163$, $e50$).
L1: $\forall A,B: r4(C) \land r2(B, A) \rightarrow r3(A, B)$
Does it imply that the statement "r3($e76$, $e1$)." is True?

A: False
```

## E   FAILURE CASES OF GPT-4 IN INDUCTION AND ABDUCTION

### E.1   EXAMPLE 1 OF ABDUCTIVE REASONING

```
system: You are a helpful assistant with abductive reasoning abilities. Please
    select one single logical rule and a few facts to explain the following
    statement.
user: I will provide a set of logical rules L1 to L28 and facts F1 to F63. Please
    select one single logical rule from L1 to L28 and a few facts from F1 to F63
    to explain the following statement.
Rules:
L1: $\forall A,B,C: r1(B, A) \land r1(B, C) \land r44(A) \rightarrow r2(A, C)$
L2: $\forall A,B,C: r1(B, A) \land r1(B, C) \land r43(A) \rightarrow r3(A, C)$
L3: $\forall A,B: r1(A, B) \land r44(A) \rightarrow r4(A, B)$
```

```
L4: $\forall A,B: r1(A, B) \land r43(A) \rightarrow r5(A, B)$
L5: $\forall A,B,C: r1(A, B) \land r1(B, C) \land r44(A) \rightarrow r6(A, C)$
L6: $\forall A,B,C: r1(A, B) \land r1(B, C) \land r43(A) \rightarrow r7(A, C)$
L7: $\forall A,B,C,D: r1(A, B) \land r1(B, C) \land r1(C, D) \land r44(A) \
    rightarrow r8(A, D)$
L8: $\forall A,B,C,D: r1(A, B) \land r1(B, C) \land r1(C, D) \land r43(A) \
    rightarrow r9(A, D)$
L9: $\forall A,B,C,D: r1(B, A) \land r1(B, C) \land r1(C, D) \land r44(A) \
    rightarrow r10(A, D)$
L10: $\forall A,B,C,D: r1(B, A) \land r1(B, C) \land r1(C, D) \land r43(A) \
    rightarrow r11(A, D)$
L11: $\forall A,B,C,D,E: r1(B, A) \land r1(B, C) \land r1(C, D) \land r1(D, E) \
    land r44(A) \rightarrow r12(A, E)$
L12: $\forall A,B,C,D,E: r1(B, A) \land r1(B, C) \land r1(C, D) \land r1(D, E) \
    land r43(A) \rightarrow r13(A, E)$
L13: $\forall A,B,C,D,E,F: r1(B, A) \land r1(C, B) \land r1(C, D) \land r1(D, E) \
    land r1(E, F) \land r44(A) \rightarrow r14(A, F)$
L14: $\forall A,B,C,D,E,F: r1(B, A) \land r1(C, B) \land r1(C, D) \land r1(D, E) \
    land r1(E, F) \land r43(A) \rightarrow r15(A, F)$
L15: $\forall A,B,C,D,E: r1(B, A) \land r1(C, B) \land r1(C, D) \land r1(D, E) \
    land r44(A) \rightarrow r16(A, E)$
L16: $\forall A,B,C,D,E: r1(B, A) \land r1(C, B) \land r1(C, D) \land r1(D, E) \
    land r43(A) \rightarrow r17(A, E)$
L17: $\forall A,B,C,D,E,F,G: r1(B, A) \land r1(C, B) \land r1(D, C) \land r1(D, E)
    \land r1(E, F) \land r1(F, G) \land r44(A) \rightarrow r18(A, G)$
L18: $\forall A,B,C,D,E,F,G: r1(B, A) \land r1(C, B) \land r1(D, C) \land r1(D, E)
    \land r1(E, F) \land r1(F, G) \land r43(A) \rightarrow r19(A, G)$
L19: $\forall A,B,C,D,E,F: r1(B, A) \land r1(C, B) \land r1(D, C) \land r1(D, E) \
    land r1(E, F) \land r44(A) \rightarrow r20(A, F)$
L20: $\forall A,B,C,D,E,F: r1(B, A) \land r1(C, B) \land r1(D, C) \land r1(D, E) \
    land r1(E, F) \land r43(A) \rightarrow r21(A, F)$
L21: $\forall A,B: r1(B, A) \land r44(A) \rightarrow r22(A, B)$
L22: $\forall A,B: r1(B, A) \land r43(A) \rightarrow r23(A, B)$
L23: $\forall A,B,C: r1(B, A) \land r1(C, B) \land r44(A) \rightarrow r24(A, C)$
L24: $\forall A,B,C: r1(B, A) \land r1(C, B) \land r43(A) \rightarrow r25(A, C)$
L25: $\forall A,B,C,D: r1(B, A) \land r1(C, B) \land r1(D, C) \land r44(A) \
    rightarrow r26(A, D)$
L26: $\forall A,B,C,D: r1(B, A) \land r1(C, B) \land r1(D, C) \land r43(A) \
    rightarrow r27(A, D)$
L27: $\forall A,B,C,D: r1(B, A) \land r1(C, B) \land r1(C, D) \land r44(A) \
    rightarrow r28(A, D)$
L28: $\forall A,B,C,D: r1(B, A) \land r1(C, B) \land r1(C, D) \land r43(A) \
    rightarrow r29(A, D)$

Facts:
F1: $r1$(Laura,Fabian)
F2: $r1$(Laura,Felix)
F3: $r1$(Laura,Claudia)
F4: $r1$(Elias,Fabian)
F5: $r1$(Elias,Felix)
F6: $r1$(Elias,Claudia)
F7: $r1$(Alina,David)
F8: $r1$(Alina,Lea)
F9: $r1$(Nico,David)
F10: $r1$(Nico,Lea)
F11: $r1$(Emily,Nico)
F12: $r1$(Konstantin,Nico)
F13: $r1$(Fabian,Thomas)
F14: $r1$(Fabian,Amelie)
F15: $r1$(Nina,Tobias)
F16: $r1$(Leonie,Emily)
F17: $r1$(Stefan,Emily)
F18: $r1$(Gabriel,Tobias)
F19: $r1$(Elena,Thomas)
F20: $r1$(Elena,Amelie)
F21: $r1$(Thomas,Helga)
F22: $r1$(Thomas,Nina)
F23: $r1$(Thomas,Patrick)
F24: $r1$(Luisa,Helga)
F25: $r1$(Luisa,Nina)
F26: $r1$(Luisa,Patrick)
F27: $r1$(Patrick,Samuel)
```

```
F28: $r1$(Patrick,Alina)
F29: $r1$(Patrick,Jonathan)
F30: $r1$(Patrick,Philipp)
F31: $r1$(Patrick,Florian)
F32: $r1$(Emilia,Samuel)
F33: $r1$(Emilia,Alina)
F34: $r1$(Emilia,Jonathan)
F35: $r1$(Emilia,Philipp)
F36: $r1$(Emilia,Florian)
F37: $r44$(Laura)
F38: $r43$(Elias)
F39: $r43$(Fabian)
F40: $r44$(Claudia)
F41: $r44$(Elena)
F42: $r43$(Thomas)
F43: $r44$(Amelie)
F44: $r44$(Luisa)
F45: $r43$(Patrick)
F46: $r44$(Emilia)
F47: $r43$(Samuel)
F48: $r44$(Alina)
F49: $r43$(Jonathan)
F50: $r43$(Philipp)
F51: $r43$(Nico)
F52: $r43$(David)
F53: $r44$(Emily)
F54: $r43$(Konstantin)
F55: $r43$(Florian)
F56: $r44$(Helga)
F57: $r44$(Nina)
F58: $r44$(Lea)
F59: $r43$(Felix)
F60: $r44$(Leonie)
F61: $r43$(Stefan)
F62: $r43$(Gabriel)
F63: $r43$(Tobias)

Statement: $r9$(Elias, Nina)
Answer with the numbers of the selected rule and facts. The selected rule and
    facts are: Let's think step by step.

GPT4: To prove the statement $r9$(Elias, Nina), we can use the rule L9 and facts
    F5, F22, and F38.

L9: $\forall A,B,C,D: r1(B, A) \land r1(B, C) \land r1(C, D) \land r44(A) \
    rightarrow r9(A, D)$

F5: $r1$(Elias,Felix)
F22: $r1$(Thomas,Nina)
F38: $r44$(Elias)

According to L9, we need to find a relation between Elias and Nina. We have the
    following relations:

1. Elias is related to Felix (F5).
2. Thomas is related to Nina (F22).
3. Elias has property r44 (F38).

Now, we can apply L9:

$r1$(Elias, Felix) $\land$ $r1$(Elias, Thomas) $\land$ $r1$(Thomas, Nina) $\land$
    $r44$(Elias) $\rightarrow$ $r9$(Elias, Nina)

Since we have all the required relations and properties, we can conclude that $r9$
    (Elias, Nina) is true.

The selected rule and facts are: L9, F5, F22, and F38.
```

From the failure example, we can observe that the model was expected to identify rule L8 for reasoning, but it incorrectly formed rule L9, which indicates that it might have been influenced by

irrelevant information in the long context. Moreover, the model made up facts, such as $r1$(Elias, Thomas), demonstrating that it experienced a hallucination.

## E.2 EXAMPLE 2 OF ABDUCTIVE REASONING

**system**: You are a helpful assistant with abductive reasoning abilities. Please
    select one single logical rule and a few facts to explain the following
    statement.
**user**: I will provide a set of logical rules L1 to L28 and facts F1 to F63. Please
    select one single logical rule from L1 to L28 and a few facts from F1 to F63
    to explain the following statement.
Rules:
L1: $\forall A,B,C: r1(B, A) \land r1(B, C) \land r44(A) \rightarrow r2(A, C)$
L2: $\forall A,B,C: r1(B, A) \land r1(B, C) \land r43(A) \rightarrow r3(A, C)$
L3: $\forall A,B: r1(A, B) \land r44(A) \rightarrow r4(A, B)$
L4: $\forall A,B: r1(A, B) \land r43(A) \rightarrow r5(A, B)$
L5: $\forall A,B,C: r1(A, B) \land r1(B, C) \land r44(A) \rightarrow r6(A, C)$
L6: $\forall A,B,C: r1(A, B) \land r1(B, C) \land r43(A) \rightarrow r7(A, C)$
L7: $\forall A,B,C,D: r1(A, B) \land r1(B, C) \land r1(C, D) \land r44(A) \
    rightarrow r8(A, D)$
L8: $\forall A,B,C,D: r1(A, B) \land r1(B, C) \land r1(C, D) \land r43(A) \
    rightarrow r9(A, D)$
L9: $\forall A,B,C,D: r1(B, A) \land r1(B, C) \land r1(C, D) \land r44(A) \
    rightarrow r10(A, D)$
L10: $\forall A,B,C,D: r1(B, A) \land r1(B, C) \land r1(C, D) \land r43(A) \
    rightarrow r11(A, D)$
L11: $\forall A,B,C,D,E: r1(B, A) \land r1(B, C) \land r1(C, D) \land r1(D, E) \
    land r44(A) \rightarrow r12(A, E)$
L12: $\forall A,B,C,D,E: r1(B, A) \land r1(B, C) \land r1(C, D) \land r1(D, E) \
    land r43(A) \rightarrow r13(A, E)$
L13: $\forall A,B,C,D,E,F: r1(B, A) \land r1(C, B) \land r1(C, D) \land r1(D, E) \
    land r1(E, F) \land r44(A) \rightarrow r14(A, F)$
L14: $\forall A,B,C,D,E,F: r1(B, A) \land r1(C, B) \land r1(C, D) \land r1(D, E) \
    land r1(E, F) \land r43(A) \rightarrow r15(A, F)$
L15: $\forall A,B,C,D,E: r1(B, A) \land r1(C, B) \land r1(C, D) \land r1(D, E) \
    land r44(A) \rightarrow r16(A, E)$
L16: $\forall A,B,C,D,E: r1(B, A) \land r1(C, B) \land r1(C, D) \land r1(D, E) \
    land r43(A) \rightarrow r17(A, E)$
L17: $\forall A,B,C,D,E,F,G: r1(B, A) \land r1(C, B) \land r1(D, C) \land r1(D, E)
     \land r1(E, F) \land r1(F, G) \land r44(A) \rightarrow r18(A, G)$
L18: $\forall A,B,C,D,E,F,G: r1(B, A) \land r1(C, B) \land r1(D, C) \land r1(D, E)
     \land r1(E, F) \land r1(F, G) \land r43(A) \rightarrow r19(A, G)$
L19: $\forall A,B,C,D,E,F: r1(B, A) \land r1(C, B) \land r1(D, C) \land r1(D, E) \
    land r1(E, F) \land r44(A) \rightarrow r20(A, F)$
L20: $\forall A,B,C,D,E,F: r1(B, A) \land r1(C, B) \land r1(D, C) \land r1(D, E) \
    land r1(E, F) \land r43(A) \rightarrow r21(A, F)$
L21: $\forall A,B: r1(B, A) \land r44(A) \rightarrow r22(A, B)$
L22: $\forall A,B: r1(B, A) \land r43(A) \rightarrow r23(A, B)$
L23: $\forall A,B,C: r1(B, A) \land r1(C, B) \land r44(A) \rightarrow r24(A, C)$
L24: $\forall A,B,C: r1(B, A) \land r1(C, B) \land r43(A) \rightarrow r25(A, C)$
L25: $\forall A,B,C,D: r1(B, A) \land r1(C, B) \land r1(D, C) \land r44(A) \
    rightarrow r26(A, D)$
L26: $\forall A,B,C,D: r1(B, A) \land r1(C, B) \land r1(D, C) \land r43(A) \
    rightarrow r27(A, D)$
L27: $\forall A,B,C,D: r1(B, A) \land r1(C, B) \land r1(C, D) \land r44(A) \
    rightarrow r28(A, D)$
L28: $\forall A,B,C,D: r1(B, A) \land r1(C, B) \land r1(C, D) \land r43(A) \
    rightarrow r29(A, D)$

Facts:
F1: $r1$(Laura,Fabian)
F2: $r1$(Laura,Felix)
F3: $r1$(Laura,Claudia)
F4: $r1$(Elias,Fabian)
F5: $r1$(Elias,Felix)
F6: $r1$(Elias,Claudia)
F7: $r1$(Alina,David)
F8: $r1$(Alina,Lea)
F9: $r1$(Nico,David)
F10: $r1$(Nico,Lea)
F11: $r1$(Emily,Nico)

```
F12: $r1$(Konstantin,Nico)
F13: $r1$(Fabian,Thomas)
F14: $r1$(Fabian,Amelie)
F15: $r1$(Nina,Tobias)
F16: $r1$(Leonie,Emily)
F17: $r1$(Stefan,Emily)
F18: $r1$(Gabriel,Tobias)
F19: $r1$(Elena,Thomas)
F20: $r1$(Elena,Amelie)
F21: $r1$(Thomas,Helga)
F22: $r1$(Thomas,Nina)
F23: $r1$(Thomas,Patrick)
F24: $r1$(Luisa,Helga)
F25: $r1$(Luisa,Nina)
F26: $r1$(Luisa,Patrick)
F27: $r1$(Patrick,Samuel)
F28: $r1$(Patrick,Alina)
F29: $r1$(Patrick,Jonathan)
F30: $r1$(Patrick,Philipp)
F31: $r1$(Patrick,Florian)
F32: $r1$(Emilia,Samuel)
F33: $r1$(Emilia,Alina)
F34: $r1$(Emilia,Jonathan)
F35: $r1$(Emilia,Philipp)
F36: $r1$(Emilia,Florian)
F37: $r44$(Laura)
F38: $r43$(Elias)
F39: $r43$(Fabian)
F40: $r44$(Claudia)
F41: $r44$(Elena)
F42: $r43$(Thomas)
F43: $r44$(Amelie)
F44: $r44$(Luisa)
F45: $r43$(Patrick)
F46: $r44$(Emilia)
F47: $r43$(Samuel)
F48: $r44$(Alina)
F49: $r43$(Jonathan)
F50: $r43$(Philipp)
F51: $r43$(Nico)
F52: $r43$(David)
F53: $r44$(Emily)
F54: $r43$(Konstantin)
F55: $r43$(Florian)
F56: $r44$(Helga)
F57: $r44$(Nina)
F58: $r44$(Lea)
F59: $r43$(Felix)
F60: $r44$(Leonie)
F61: $r43$(Stefan)
F62: $r43$(Gabriel)
F63: $r43$(Tobias)

Statement: $r11$(Samuel, Lea)
Answer with the numbers of the selected rule and facts. The selected rule and
    facts are: Let's think step by step.
```

**GPT4**: First, we need to find a rule that has the conclusion $r11(A, D)$. The only
    rule that has this conclusion is L10.

L10: $\forall A,B,C,D: r1(B, A) \land r1(B, C) \land r1(C, D) \land r43(A) \rightarrow r11(A, D)$

Now, we need to find facts that match the premises of L10. We are looking for
    facts where $r1(B, A)$, $r1(B, C)$, $r1(C, D)$, and $r43(A)$ are true.

From the statement, we know that A = Samuel and D = Lea. So, we need to find facts
    where $r1(B, Samuel)$, $r1(B, C)$, $r1(C, Lea)$, and $r43(Samuel)$ are true.

From the facts, we have:

```
F47: $r43$(Samuel)

Now we need to find B and C such that $r1(B, Samuel)$, $r1(B, C)$, and $r1(C, Lea)
    $ are true.

From the facts, we have:

F28: $r1$(Patrick, Samuel)

Now we need to find C such that $r1$(Patrick, C) and $r1(C, Lea)$ are true.

From the facts, we have:

F30: $r1$(Patrick, Philipp)
F52: $r1$(David, Lea)

Now we have all the facts needed to match the premises of L10:

L10: $\forall A,B,C,D: r1(B, A) \land r1(B, C) \land r1(C, D) \land r43(A) \
    rightarrow r11(A, D)$

F28: $r1$(Patrick, Samuel)
F30: $r1$(Patrick, Philipp)
F52: $r1$(David, Lea)
F47: $r43$(Samuel)

The selected rule and facts are: L10, F28, F30, F52, and F47.
```

In this case, although GPT-4 was able to identify the correct reasoning rules in this example, it made errors during the variable grounding process, which ultimately led to failed predictions.

### E.3  EXAMPLE 1 OF INDUCTIVE REASONING

```
system: You are a helpful assistant with inductive reasoning abilities. Please
    generate one single rule to match the template and logically entail the facts.
     Note that the symbol '##' in the template should be filled with either 'r1'
    or 'r45', while the symbol '++' should be filled with either 'r43' or 'r44'.
user: I will give you a set of facts F1 to F99, facts G1 to G12 and a template for
     a logical rule. Please fill in the template so that the generated rule can
    logically entail the facts G1 to G12 based on facts F1 to F99.
Facts:
F1: $r44$(Laura)
F2: $r43$(Elias)
F3: $r43$(Fabian)
F4: $r44$(Claudia)
F5: $r44$(Elena)
F6: $r43$(Thomas)
F7: $r44$(Amelie)
F8: $r44$(Luisa)
F9: $r43$(Patrick)
F10: $r44$(Emilia)
F11: $r43$(Samuel)
F12: $r44$(Alina)
F13: $r43$(Jonathan)
F14: $r43$(Philipp)
F15: $r43$(Nico)
F16: $r43$(David)
F17: $r44$(Emily)
F18: $r43$(Konstantin)
F19: $r43$(Florian)
F20: $r44$(Helga)
F21: $r44$(Nina)
F22: $r44$(Lea)
F23: $r43$(Felix)
F24: $r44$(Leonie)
F25: $r43$(Stefan)
F26: $r43$(Gabriel)
F27: $r43$(Tobias)
F28: $r1$(Laura, Fabian)
F29: $r45$(Fabian, Laura)
```

```
F30: $r1$(Laura, Felix)
F31: $r45$(Felix, Laura)
F32: $r1$(Laura, Claudia)
F33: $r45$(Claudia, Laura)
F34: $r1$(Elias, Fabian)
F35: $r45$(Fabian, Elias)
F36: $r1$(Elias, Felix)
F37: $r45$(Felix, Elias)
F38: $r1$(Elias, Claudia)
F39: $r45$(Claudia, Elias)
F40: $r1$(Alina, David)
F41: $r45$(David, Alina)
F42: $r1$(Alina, Lea)
F43: $r45$(Lea, Alina)
F44: $r1$(Nico, David)
F45: $r45$(David, Nico)
F46: $r1$(Nico, Lea)
F47: $r45$(Lea, Nico)
F48: $r1$(Emily, Nico)
F49: $r45$(Nico, Emily)
F50: $r1$(Konstantin, Nico)
F51: $r45$(Nico, Konstantin)
F52: $r1$(Fabian, Thomas)
F53: $r45$(Thomas, Fabian)
F54: $r1$(Fabian, Amelie)
F55: $r45$(Amelie, Fabian)
F56: $r1$(Nina, Tobias)
F57: $r45$(Tobias, Nina)
F58: $r1$(Leonie, Emily)
F59: $r45$(Emily, Leonie)
F60: $r1$(Stefan, Emily)
F61: $r45$(Emily, Stefan)
F62: $r1$(Gabriel, Tobias)
F63: $r45$(Tobias, Gabriel)
F64: $r1$(Elena, Thomas)
F65: $r45$(Thomas, Elena)
F66: $r1$(Elena, Amelie)
F67: $r45$(Amelie, Elena)
F68: $r1$(Thomas, Helga)
F69: $r45$(Helga, Thomas)
F70: $r1$(Thomas, Nina)
F71: $r45$(Nina, Thomas)
F72: $r1$(Thomas, Patrick)
F73: $r45$(Patrick, Thomas)
F74: $r1$(Luisa, Helga)
F75: $r45$(Helga, Luisa)
F76: $r1$(Luisa, Nina)
F77: $r45$(Nina, Luisa)
F78: $r1$(Luisa, Patrick)
F79: $r45$(Patrick, Luisa)
F80: $r1$(Patrick, Samuel)
F81: $r45$(Samuel, Patrick)
F82: $r1$(Patrick, Alina)
F83: $r45$(Alina, Patrick)
F84: $r1$(Patrick, Jonathan)
F85: $r45$(Jonathan, Patrick)
F86: $r1$(Patrick, Philipp)
F87: $r45$(Philipp, Patrick)
F88: $r1$(Patrick, Florian)
F89: $r45$(Florian, Patrick)
F90: $r1$(Emilia, Samuel)
F91: $r45$(Samuel, Emilia)
F92: $r1$(Emilia, Alina)
F93: $r45$(Alina, Emilia)
F94: $r1$(Emilia, Jonathan)
F95: $r45$(Jonathan, Emilia)
F96: $r1$(Emilia, Philipp)
F97: $r45$(Philipp, Emilia)
F98: $r1$(Emilia, Florian)
F99: $r45$(Florian, Emilia)
G1: $r2$(Alina, Samuel)
G2: $r2$(Alina, Jonathan)
```

```
G3: $r2$(Alina, Philipp)
G4: $r2$(Alina, Florian)
G5: $r2$(Helga, Nina)
G6: $r2$(Helga, Patrick)
G7: $r2$(Nina, Helga)
G8: $r2$(Nina, Patrick)
G9: $r2$(Lea, David)
G10: $r2$(Claudia, Fabian)
G11: $r2$(Claudia, Felix)
G12: $r2$(Amelie, Thomas)

Template: $\forall A, B, C: ##(A, B) \land ##(B, C) \land ++(A) \rightarrow r2(A,
    C)$
Note that the symbol '##' in the template should be filled with either 'r1' or '
    r45', while the symbol '++' should be filled with either 'r43' or 'r44'.
After filling in the template, the generated rule is: Let's think step by step.

GPT4: $\forall A, B, C: r1(A, B) \land r45(B, C) \land r44(A) \rightarrow r2(A, C)
    $

GroundTruth: $\forall A, B, C: r45(A, B) \land r1(B, C) \land r44(A) \rightarrow
    r2(A, C)$
```

In the inductive failure example, GPT-4's failure might be attributed to its confusion between "r1(A, B)" and "r45(B, A)" to some extent. Such phenomena can also be observed in other work, as reported by (Berglund et al., 2023) in their study on reversal curse.

## F    DIFFERENT ZERO-SHOT PROMPTING

We try different Zero-Shot prompts:

(1)

```
I will provide a set of logical rules L1 to L{number of rules} and facts F1 to F{
    number of basic facts}. Please select one single logical rule from L1 to L{
    number of rules} and a few facts from F1 to F{number of basic facts} to
    predict True/False of the unknown fact using deductive reasoning.
Logical rules: {rules}
Facts: {basic facts}
Unknown fact: {unknown fact}
The answer (True or False) is:
```

(2)

```
I will provide a set of logical rules L1 to L{number of rules} and facts F1 to F{
    number of basic facts}. Please predict True/False of the unknown fact using
    deductive reasoning.
Logical rules: {rules}
Facts: {basic facts}
Unknown fact: {unknown fact}
The answer (True or False) is:
```

(3)

```
Given a set of rules and facts, you have to reason whether a statement is True or
    False.
Here are some rules: {rules}
Here are some facts: {basic facts}
Does it imply that the statement "{unknown fact}" is True?
The answer (YES or NO) is:
```

The results of the three prompts in the Zero-Shot setting are presented in Table 7. Among the three prompts, we select the one that achieves the best performance as our Zero-Shot prompt.

## G    TASK DEFINITIONS

We define a few tasks to evaluate LLMs' abilities of three kinds of reasoning and memorization.

|          | prompt1 | prompt2 | prompt3 |
|----------|---------|---------|---------|
| Tree$_1$ | 54.5    | 51.5    | 53.8    |

Table 7: Different Zero-Shot Prompts of deductive reasoning. Results are in %.

- *deductive reasoning:* we use *hypothesis classification*, *i.e.*, predict the *correctness* of the *hypothesis* given the *theory* where *theory* consists of basic facts and logical rules, *correctness* can be true or false, and *hypothesis* is a predicted fact, which is one of the inferred facts or negative samples. The accuracy is the proportion of correct predictions.

- *inductive reasoning:* we perform the *rule generation* task. Given multiple facts with similar patterns and a rule template, the goal is to induce a rule that entails these facts. Specifically, for each relation $r$, we use basic facts and those inferred facts that contain only relation $r$ as provided facts. The induced rule is generated after filling in the rule template. We test the generated rules against the ground truth rules. If the generated rule matches the ground truth rule exactly, we predict the rule to be correct; otherwise, we predict the rule to be incorrect. The precision is the proportion of correct predictions. Note that considering logical rules maybe not all chain rules (e.g., $r_1(y, x) \wedge r_2(y, z) \rightarrow r_3(x, z)$), we add inverse relation for each relation in order to transform them into chain rules and simplify the rule template (e.g., $r_1^{-1}(x, y) \wedge r_2(y, z) \rightarrow r_3(x, z)$). Furthermore, we provide a rule template for each relation. Take $auntOf$ as example, its rule template can be $\forall x, y, z : \#\#(x, y) \wedge \#\#(y, z) \wedge ++(x) \rightarrow auntOf(x, z)$ or "If x is ## of y and y is ## of z and x is ++, then x is aunt of z.", where ## can be $parent$ or $inverse\_parent$, ++ can be $female$ or $male$.

  Besides, a single rule can be equivalent to multiple rules. For example, the rule $\forall x, y, z :$ parentOf$(x, y) \wedge$ parentOf$(y, z) \wedge$ gender$(x,$ female$) \rightarrow$ GrandmotherOf$(x, z)$ can be represented as $\forall x, y, z :$ parentOf$(x, y) \wedge$ parentOf$(y, z) \rightarrow$ GrandparentOf$(x, z)$, GrandparentOf$(x, z) \wedge$ gender$(x,$ female$) \rightarrow$ GrandmotherOf$(x, z)$. We conduct the experiments with both rule representations and find single-longer rules perform better than multiple-short rules. Results are presented in Appendix S. Based on these observations and considering the simplicity of induction evaluation, we rewrite all logical rules by including only the $parentOf$ and $gender$ relations in the rule body. This also ensures that each inferred relation is implied by a single logical rule, referred to as *grounding truth rule*.

- *abductive reasoning:* We use *explanation generation* to evaluate abductive reasoning abilities. Given a *theory* including basic facts and all logical rules, the task is to select specific facts and a logical rule to explain the *observation*. The *observation* is chosen from inferred facts. We use Proof Accuracy (PA) as an evaluation metric, i.e., the fraction of examples where the generated proof matches exactly any of the gold proofs.

# H  IMPLEMENTATION

## H.1  HUMAN STUDY

For the human study, we recruited 11 participants from diverse science and engineering backgrounds, including computer science, electronics, artificial intelligence, and automation. Although they have basic understanding of simple logic concepts, they are not experts in logical reasoning. Therefore, we provided them with task instructions that explained the concepts of deduction, induction, and abduction, aligned with the illustrations and definitions of logical reasoning presented in Section 3 of our paper.

We then presented them with 18 specific tasks, including six tasks for each deductive, inductive, and abductive reasoning type. Each task closely resembled the zero-shot prompts given to LLMs. We refer to this setting as "zero-shot" because we did not provide any further specific examples to help participants understand the tasks, and there were no time limits for completion. The examples can be found in Appendix I.

## H.2 Fine-tuning

For fine-tuning, we select five sampled Symbolic Trees for fine-tuning and another three for testing. We utilized 4 A100 80G GPUs with batch size 2 for finetuning. The training process involved 1 epochs (by default), employing a cosine learning rate schedule with an initial learning rate of 2e-5.

# I Examples of Human study

## I.1 Deduction

```
Given a set of rules and facts, you have to reason whether a statement is true or
    false. Here are some facts and rules.
F1: $r1$(maximilian, nina).
F2: $r1$(maximilian, david).
F3: $r1$(maximilian, lukas).
F4: $r1$(lina, maximilian).
F5: $r1$(lina, marie).
F6: $r1$(clara, claudia).
F7: $r1$(claudia, lea).
F8: $r1$(sarah, paula).
F9: $r1$(sarah, emma).
F10: $r1$(angelina, victoria).
F11: $r1$(adam, victoria).
F12: $r1$(raphael, paula).
F13: $r1$(raphael, emma).
F14: $r1$(luca, maximilian).
F15: $r1$(luca, marie).
F16: $r1$(emma, julian).
F17: $r1$(emma, leon).
F18: $r1$(jonas, lea).
F19: $r1$(daniel, julian).
F20: $r1$(daniel, leon).
F21: $r1$(olivia, nina).
F22: $r1$(olivia, david).
F23: $r1$(olivia, lukas).
F24: $r1$(david, sarah).
F25: $r1$(david, valentina).
F26: $r1$(lukas, vincent).
F27: $r1$(lukas, paul).
F28: $r1$(victoria, sarah).
F29: $r1$(victoria, valentina).
F30: $r1$(valerie, vincent).
F31: $r1$(valerie, paul).
F32: $r1$(paul, claudia).
F33: $r43$(maximilian).
F34: $r44$(lina).
F35: $r43$(luca).
F36: $r44$(olivia).
F37: $r43$(david).
F38: $r43$(lukas).
F39: $r44$(victoria).
F40: $r44$(valentina).
F41: $r44$(valerie).
F42: $r43$(paul).
F43: $r44$(clara).
F44: $r44$(claudia).
F45: $r44$(sarah).
F46: $r44$(angelina).
F47: $r43$(adam).
F48: $r44$(marie).
F49: $r43$(vincent).
F50: $r44$(nina).
F51: $r43$(raphael).
F52: $r44$(paula).
F53: $r44$(emma).
F54: $r43$(jonas).
F55: $r44$(lea).
F56: $r43$(daniel).
F57: $r43$(julian).
```

```
F58: $r43$(leon).
L1: $\forall A,B,C: r1(B, A) \land r1(B, C) \land r44(A) \rightarrow r2(A, C).
L2: $\forall A,B,C: r1(B, A) \land r1(B, C) \land r43(A) \rightarrow r3(A, C).
L3: $\forall A,B: r1(A, B) \land r44(A) \rightarrow r4(A, B).
L4: $\forall A,B: r1(A, B) \land r43(A) \rightarrow r5(A, B).
L5: $\forall A,B,C: r1(A, B) \land r1(B, C) \land r44(A) \rightarrow r6(A, C).
L6: $\forall A,B,C: r1(A, B) \land r1(B, C) \land r43(A) \rightarrow r7(A, C).
L7: $\forall A,B,C,D: r1(A, B) \land r1(B, C) \land r1(C, D) \land r44(A) \
    rightarrow r8(A, D).
L8: $\forall A,B,C,D: r1(A, B) \land r1(B, C) \land r1(C, D) \land r43(A) \
    rightarrow r9(A, D).
L9: $\forall A,B,C,D: r1(B, A) \land r1(B, C) \land r1(C, D) \land r44(A) \
    rightarrow r10(A, D).
L10: $\forall A,B,C,D: r1(B, A) \land r1(B, C) \land r1(C, D) \land r43(A) \
    rightarrow r11(A, D).
L11: $\forall A,B,C,D,E: r1(B, A) \land r1(B, C) \land r1(C, D) \land r1(D, E) \
    land r44(A) \rightarrow r12(A, E).
L12: $\forall A,B,C,D,E: r1(B, A) \land r1(B, C) \land r1(C, D) \land r1(D, E) \
    land r43(A) \rightarrow r13(A, E).
L13: $\forall A,B,C,D,E,F: r1(B, A) \land r1(C, B) \land r1(C, D) \land r1(D, E) \
    land r1(E, F) \land r44(A) \rightarrow r14(A, F).
L14: $\forall A,B,C,D,E,F: r1(B, A) \land r1(C, B) \land r1(C, D) \land r1(D, E) \
    land r1(E, F) \land r43(A) \rightarrow r15(A, F).
L15: $\forall A,B,C,D,E: r1(B, A) \land r1(C, B) \land r1(C, D) \land r1(D, E) \
    land r44(A) \rightarrow r16(A, E).
L16: $\forall A,B,C,D,E: r1(B, A) \land r1(C, B) \land r1(C, D) \land r1(D, E) \
    land r43(A) \rightarrow r17(A, E).
L17: $\forall A,B,C,D,E,F,G: r1(B, A) \land r1(C, B) \land r1(D, C) \land r1(D, E)
     \land r1(E, F) \land r1(F, G) \land r44(A) \rightarrow r18(A, G).
L18: $\forall A,B,C,D,E,F,G: r1(B, A) \land r1(C, B) \land r1(D, C) \land r1(D, E)
     \land r1(E, F) \land r1(F, G) \land r43(A) \rightarrow r19(A, G).
L19: $\forall A,B,C,D,E,F: r1(B, A) \land r1(C, B) \land r1(D, C) \land r1(D, E) \
    land r1(E, F) \land r44(A) \rightarrow r20(A, F).
L20: $\forall A,B,C,D,E,F: r1(B, A) \land r1(C, B) \land r1(D, C) \land r1(D, E) \
    land r1(E, F) \land r43(A) \rightarrow r21(A, F).
L21: $\forall A,B: r1(B, A) \land r44(A) \rightarrow r22(A, B).
L22: $\forall A,B: r1(B, A) \land r43(A) \rightarrow r23(A, B).
L23: $\forall A,B,C: r1(B, A) \land r1(C, B) \land r44(A) \rightarrow r24(A, C).
L24: $\forall A,B,C: r1(B, A) \land r1(C, B) \land r43(A) \rightarrow r25(A, C).
L25: $\forall A,B,C,D: r1(B, A) \land r1(C, B) \land r1(D, C) \land r44(A) \
    rightarrow r26(A, D).
L26: $\forall A,B,C,D: r1(B, A) \land r1(C, B) \land r1(D, C) \land r43(A) \
    rightarrow r27(A, D).
L27: $\forall A,B,C,D: r1(B, A) \land r1(C, B) \land r1(C, D) \land r44(A) \
    rightarrow r28(A, D).
L28: $\forall A,B,C,D: r1(B, A) \land r1(C, B) \land r1(C, D) \land r43(A) \
    rightarrow r29(A, D).
Does it imply that the statement "$r23$(vincent, lukas)." is True? If the
    statement is True, please answer with "True". Otherwise, please answer with "
    False".
```

## I.2 INDUCTION

```
I will give you a set of facts F1 to F94, facts G1 to G5 and a template for a
    logical rule. Please generate one single rule to match the template and
    logically entail the facts G1 to G5 based on facts F1 to F94.
F1: $r1$(moritz, natalie).
F2: $r45$(natalie, moritz).
F3: $r1$(moritz, sophie).
F4: $r45$(sophie, moritz).
F5: $r1$(valerie, natalie).
F6: $r45$(natalie, valerie).
F7: $r1$(valerie, sophie).
F8: $r45$(sophie, valerie).
F9: $r1$(katharina, victoria).
F10: $r45$(victoria, katharina).
F11: $r1$(katharina, benjamin).
F12: $r45$(benjamin, katharina).
F13: $r1$(david, theodor).
F14: $r45$(theodor, david).
F15: $r1$(david, helga).
```

```
F16: $r45$(helga, david).
F17: $r1$(david, patrick).
F18: $r45$(patrick, david).
F19: $r1$(theodor, fabian).
F20: $r45$(fabian, theodor).
F21: $r1$(patrick, tobias).
F22: $r45$(tobias, patrick).
F23: $r1$(emily, fabian).
F24: $r45$(fabian, emily).
F25: $r1$(vanessa, tobias).
F26: $r45$(tobias, vanessa).
F27: $r1$(natalie, theodor).
F28: $r45$(theodor, natalie).
F29: $r1$(natalie, helga).
F30: $r45$(helga, natalie).
F31: $r1$(natalie, patrick).
F32: $r45$(patrick, natalie).
F33: $r1$(noah, victoria).
F34: $r45$(victoria, noah).
F35: $r1$(noah, benjamin).
F36: $r45$(benjamin, noah).
F37: $r1$(olivia, moritz).
F38: $r45$(moritz, olivia).
F39: $r1$(stefan, moritz).
F40: $r45$(moritz, stefan).
F41: $r1$(sophie, marie).
F42: $r45$(marie, sophie).
F43: $r1$(sophie, jonas).
F44: $r45$(jonas, sophie).
F45: $r1$(oliver, marie).
F46: $r45$(marie, oliver).
F47: $r1$(oliver, jonas).
F48: $r45$(jonas, oliver).
F49: $r1$(jonas, katharina).
F50: $r45$(katharina, jonas).
F51: $r1$(jonas, vincent).
F52: $r45$(vincent, jonas).
F53: $r1$(jonas, amelie).
F54: $r45$(amelie, jonas).
F55: $r1$(jonas, larissa).
F56: $r45$(larissa, jonas).
F57: $r1$(jonas, sebastian).
F58: $r45$(sebastian, jonas).
F59: $r1$(emilia, katharina).
F60: $r45$(katharina, emilia).
F61: $r1$(emilia, vincent).
F62: $r45$(vincent, emilia).
F63: $r1$(emilia, amelie).
F64: $r45$(amelie, emilia).
F65: $r1$(emilia, larissa).
F66: $r45$(larissa, emilia).
F67: $r1$(emilia, sebastian).
F68: $r45$(sebastian, emilia).
F69: $r43$(moritz).
F70: $r44$(valerie).
F71: $r44$(natalie).
F72: $r44$(olivia).
F73: $r43$(stefan).
F74: $r44$(sophie).
F75: $r43$(oliver).
F76: $r43$(jonas).
F77: $r44$(emilia).
F78: $r43$(sebastian).
F79: $r44$(katharina).
F80: $r43$(vincent).
F81: $r43$(david).
F82: $r43$(theodor).
F83: $r44$(helga).
F84: $r43$(patrick).
F85: $r44$(emily).
F86: $r43$(fabian).
F87: $r44$(vanessa).
```

```
F88: $r43$(tobias).
F89: $r43$(noah).
F90: $r44$(victoria).
F91: $r43$(benjamin).
F92: $r44$(marie).
F93: $r44$(amelie).
F94: $r44$(larissa).
G1: $r16$(helga, marie.
G2: $r16$(helga, jonas.
G3: $r16$(marie, theodor.
G4: $r16$(marie, helga.
G5: $r16$(marie, patrick.
Template: $\forall A, B, C, D, E: ##(A, B) \land ##(B, C) \land ##(C, D) \land ##(
    D, E) \land ++(A) \rightarrow r16(A, E).
Note that the symbol '##' in the template should be filled with either 'r1' or '
    r45', while the symbol '++' should be filled with either 'r43' or 'r44'.After
    filling in the template, the generated rule is:
```

## I.3  ABDUCTION

```
I will provide a set of logical rules L1 to L28 and facts F1 to F58. Please select
    one single logical rule from L1 to L28 and a few facts from F1 to F58 to
    explain the following statement.
Rules:
L1: $\forall A,B,C: r1(B, A) \land r1(B, C) \land r44(A) \rightarrow r2(A, C)$
L2: $\forall A,B,C: r1(B, A) \land r1(B, C) \land r43(A) \rightarrow r3(A, C)$
L3: $\forall A,B: r1(A, B) \land r44(A) \rightarrow r4(A, B)$
L4: $\forall A,B: r1(A, B) \land r43(A) \rightarrow r5(A, B)$
L5: $\forall A,B,C: r1(A, B) \land r1(B, C) \land r44(A) \rightarrow r6(A, C)$
L6: $\forall A,B,C: r1(A, B) \land r1(B, C) \land r43(A) \rightarrow r7(A, C)$
L7: $\forall A,B,C,D: r1(A, B) \land r1(B, C) \land r1(C, D) \land r44(A) \
    rightarrow r8(A, D)$
L8: $\forall A,B,C,D: r1(A, B) \land r1(B, C) \land r1(C, D) \land r43(A) \
    rightarrow r9(A, D)$
L9: $\forall A,B,C,D: r1(B, A) \land r1(B, C) \land r1(C, D) \land r44(A) \
    rightarrow r10(A, D)$
L10: $\forall A,B,C,D: r1(B, A) \land r1(B, C) \land r1(C, D) \land r43(A) \
    rightarrow r11(A, D)$
L11: $\forall A,B,C,D,E: r1(B, A) \land r1(B, C) \land r1(C, D) \land r1(D, E) \
    land r44(A) \rightarrow r12(A, E)$
L12: $\forall A,B,C,D,E: r1(B, A) \land r1(B, C) \land r1(C, D) \land r1(D, E) \
    land r43(A) \rightarrow r13(A, E)$
L13: $\forall A,B,C,D,E,F: r1(B, A) \land r1(C, B) \land r1(C, D) \land r1(D, E) \
    land r1(E, F) \land r44(A) \rightarrow r14(A, F)$
L14: $\forall A,B,C,D,E,F: r1(B, A) \land r1(C, B) \land r1(C, D) \land r1(D, E) \
    land r1(E, F) \land r43(A) \rightarrow r15(A, F)$
L15: $\forall A,B,C,D,E: r1(B, A) \land r1(C, B) \land r1(C, D) \land r1(D, E) \
    land r44(A) \rightarrow r16(A, E)$
L16: $\forall A,B,C,D,E: r1(B, A) \land r1(C, B) \land r1(C, D) \land r1(D, E) \
    land r43(A) \rightarrow r17(A, E)$
L17: $\forall A,B,C,D,E,F,G: r1(B, A) \land r1(C, B) \land r1(D, C) \land r1(D, E)
     \land r1(E, F) \land r1(F, G) \land r44(A) \rightarrow r18(A, G)$
L18: $\forall A,B,C,D,E,F,G: r1(B, A) \land r1(C, B) \land r1(D, C) \land r1(D, E)
     \land r1(E, F) \land r1(F, G) \land r43(A) \rightarrow r19(A, G)$
L19: $\forall A,B,C,D,E,F: r1(B, A) \land r1(C, B) \land r1(D, C) \land r1(D, E) \
    land r1(E, F) \land r44(A) \rightarrow r20(A, F)$
L20: $\forall A,B,C,D,E,F: r1(B, A) \land r1(C, B) \land r1(D, C) \land r1(D, E) \
    land r1(E, F) \land r43(A) \rightarrow r21(A, F)$
L21: $\forall A,B: r1(B, A) \land r44(A) \rightarrow r22(A, B)$
L22: $\forall A,B: r1(B, A) \land r43(A) \rightarrow r23(A, B)$
L23: $\forall A,B,C: r1(B, A) \land r1(C, B) \land r44(A) \rightarrow r24(A, C)$
L24: $\forall A,B,C: r1(B, A) \land r1(C, B) \land r43(A) \rightarrow r25(A, C)$
L25: $\forall A,B,C,D: r1(B, A) \land r1(C, B) \land r1(D, C) \land r44(A) \
    rightarrow r26(A, D)$
L26: $\forall A,B,C,D: r1(B, A) \land r1(C, B) \land r1(D, C) \land r43(A) \
    rightarrow r27(A, D)$
L27: $\forall A,B,C,D: r1(B, A) \land r1(C, B) \land r1(C, D) \land r44(A) \
    rightarrow r28(A, D)$
L28: $\forall A,B,C,D: r1(B, A) \land r1(C, B) \land r1(C, D) \land r43(A) \
    rightarrow r29(A, D)$
```

```
Facts:
F1: $r1$(nico,tobias)
F2: $r1$(nico,dominik)
F3: $r1$(elena,tobias)
F4: $r1$(elena,dominik)
F5: $r1$(emily,angelina)
F6: $r1$(florian,clara)
F7: $r1$(florian,valentin)
F8: $r1$(isabella,valentina)
F9: $r1$(stefan,valentina)
F10: $r1$(lea,clara)
F11: $r1$(lea,valentin)
F12: $r1$(sebastian,angelina)
F13: $r1$(tobias,marlene)
F14: $r1$(tobias,johanna)
F15: $r1$(sarah,stefan)
F16: $r1$(noah,stefan)
F17: $r1$(charlotte,luca)
F18: $r1$(dominik,luca)
F19: $r1$(valentina,marlene)
F20: $r1$(valentina,johanna)
F21: $r1$(valerie,nico)
F22: $r1$(valerie,raphael)
F23: $r1$(valerie,adrian)
F24: $r1$(valerie,marie)
F25: $r1$(elias,nico)
F26: $r1$(elias,raphael)
F27: $r1$(elias,adrian)
F28: $r1$(elias,marie)
F29: $r1$(marie,emily)
F30: $r1$(marie,florian)
F31: $r1$(leo,emily)
F32: $r1$(leo,florian)
F33: $r43$(nico)
F34: $r44$(elena)
F35: $r43$(tobias)
F36: $r43$(dominik)
F37: $r44$(valentina)
F38: $r44$(johanna)
F39: $r44$(valerie)
F40: $r43$(elias)
F41: $r44$(marie)
F42: $r43$(leo)
F43: $r44$(emily)
F44: $r43$(florian)
F45: $r44$(marlene)
F46: $r44$(isabella)
F47: $r43$(stefan)
F48: $r44$(lea)
F49: $r44$(clara)
F50: $r43$(raphael)
F51: $r43$(sebastian)
F52: $r44$(angelina)
F53: $r44$(sarah)
F54: $r43$(noah)
F55: $r44$(charlotte)
F56: $r43$(luca)
F57: $r43$(valentin)
F58: $r43$(adrian)
Statement: $r5$(tobias, marlene)
Answer with the numbers of the selected rule and facts. The selected rule and
    facts are (There may be multiple explanations for the statement, please
    provide one) :
```

## J    DESCRIPTION OF DATASETS

The Symbolic Tree dataset is an artificially close-world and noise-free symbolic dataset generated with complex logical rules. The dataset consists of randomly sampled "*basic facts*", which include gender information and "parentOf" relations among individuals. With the given logical rules, the

dataset allows for reasoning about 28 different types of family relations, ranging from easy inferences (*e.g.*, fatherhood), to more elaborate ones (*e.g.*, a daughter of someone's cousin). *Facts* consist of *basic facts* (in-context knowledge) and *inferred facts* (what to reason). Note that Symbolic Tree is a close-world dataset, which means that any facts not presented in the dataset are assumed to be false. Thus, we construct the false facts by replacing the head entity or tail entity with a random entity as negative examples in *inferred facts*. Considering the context window size limitation, we restrict each tree's depth to 5 to generate the dataset. We experiment with 10 sampled Symbolic Trees; each has 30 kinds of relations (28 inferred relations, gender and parentOf relation), 26 entities, about 35 basic facts, 300 inferred facts and 300 false ones.

To decouple the semantics within the dataset, we replace the relation names (such as "parent") with hand-crafted symbols (*e.g.*, "r1", "r2", ...), so that LLMs cannot leverage the semantics of the predicates in reasoning but must resort to the given new knowledge (presented as in-context facts and rules). We also experiment with replacing entity names (such as "Alice") with "e1", "e2", ..., but find that it has little impact on performance (more details are provided in Appendix R). During the symbol generation process, we also try to randomly sample some letters as relation names (*e.g.*, "lnqgv" instead of "r1"), but we observe that LLMs struggle to understand garbled characters, which may negatively affect performance (further discussion is provided in Appendix O).

ProofWriter (Tafjord et al., 2020) tasks provide artificial facts and rules expressed in natural language. For our experiments, we use a subset of the ProofWriter Open World Assumption (OWA) dataset with a depth of 1, 2, 3 and 5 (there is no depth 4 task), which contains many small rulebases of facts and rules, expressed in English and do not exist in LLMs' knowledge base. Each rulebase has a set of questions (English statements) that can be proven true, false or "Unknown". Note that if we want to prove something Unknown, it is necessary to enumerate all possible facts and check their true/false. Thus, we remove all the Unknowns and replace the subject and object with entity IDs. This dataset is simpler than Symbolic Tree. Considering most of the predicates in the sentences are unmeaningful verbs like "is" and "can", we only replace the entities with entity IDs to decouple semantics. Take "Anne is kind." as an example. We substitute subject (Anne) and object (kind) with "e1" and "e2", respectively, resulting in "e1 is e2".

## K    DIFFERENT FEW-SHOT-COT EXAMPLES

We use three different kinds of demonstrations generation:

(1) leverage LLMs with the "Let's think step by step" prompt to generate reasoning chains for demonstrations.

(2) manually designed each demonstration with a question and a reasoning chain. A reasoning chain is composed of a rationale (a series of intermediate reasoning steps) and an expected answer.

(3) leverage LLMs with the "Let's think step by step" prompt to generate reasoning chains and filter with those generated chains coming with mistakes for demonstrations.

The results of these approaches are provided in Table 8. From the results, we observe that the *Auto-Randomly* approach outperforms both the *Auto-Filter* and *Manually-Designed* approaches. This suggests that randomly generating reasoning chains is more suitable for LLMs and can generate a diverse range of solutions, resulting in better reasoning performance.

### K.1    FEW-SHOT-COT EXAMPLES WITH AUTO-FILTER

Statement: $r8$(Elena, Nina)
Answer: We can use logical rule L5: $\forall A, B, C : r3(A, B) \land r3(B, C) \land r2(A) \rightarrow r8(A, C)$ to deduce whether the statement $r8$(Elena, Nina) is true or false. From the facts, we know that $r3$(Elena, Thomas) and $r3$(Thomas, Nina). Therefore, we can use L5 with $A$ =Elena, $B$ =Thomas, and $C$ =Nina to see if $r8$(Elena, Nina) is true or false. L5 states that if $r3$(A, B) and $r3$(B, C) and $r2$(A) are true, then $r8$(A, C) is true. In this case, we have $r3$(Elena, Thomas) and $r3$(Thomas, Nina) as true facts, and we also know that $r2$(Elena) is true from F5. Therefore, we can apply L5 to get:
$r8$(Elena, Nina) is true.
Therefore, the answer is True.

Statement: $r29$(Philipp, Elena)
Answer: We can use logical rule L26: $\forall A, B, C, D : r3(B, A) \wedge r3(C, B) \wedge r3(D, C) \wedge r1(A) \rightarrow r29(A, D)$ to infer that if $r3(B, A)$, $r3(C, B)$, $r3(D, C)$, and $r1(A)$ are true, then $r29(A, D)$ is also true.
From the given facts, we know that $r3$(Elena, Thomas) (F46) and $r3$(Thomas, Amelie) (F47). Therefore, we can use L26 with $A$ = Philipp, $B$ = Thomas, $C$ = Amelie, and $D$ = Elena to get:
$r3$(Thomas, Philipp) (from F62 and $r3(B, A)$)
$r3$(Amelie, Thomas) (from F47 and $r3(C, B)$)
$r3$(Elena, Amelie) (from F47 and $r3(D, C)$)
$r1$(Philipp) (from F14 and $r1(A)$)
Therefore, we can conclude that $r29$(Philipp, Elena) is True.

Statement: $r6$(Luisa, Helga)
Answer: We can use logical rule L3: $\forall A, B : r3(A, B) \wedge r2(A) \rightarrow r6(A, B)$ to deduce the truth value of the statement $r6$(Luisa, Helga).
From the facts, we know that $r3$(Luisa, Helga) and $r2$(Luisa) are true. Therefore, we can apply L3 to conclude that $r6$(Luisa, Helga) is true.
Thus, the answer is True.

Statement: $r14$(Amelie, Alina)
Answer: We can use logical rule L11: $\forall A, B, C, D, E : r3(B, A) \wedge r3(B, C) \wedge r3(C, D) \wedge r3(D, E) \wedge r2(A) \rightarrow r14(A, E)$ to infer $r14$(Amelie, Alina) if we can find facts that satisfy the antecedent of the rule.
From the facts, we can see that $r3$(Amelie, Thomas) and $r3$(Thomas, Helga) and $r3$(Helga, David) and $r3$(David, Alina) are true. Therefore, we can apply L11 with $A$ =Amelie, $B$ =Thomas, $C$ =Helga, $D$ =David, and $E$ =Alina to get $r14$(Amelie, Alina) is True.
Therefore, the answer is True.

Statement: $r26$(Alina, Luisa)
Answer: We can use logical rule L23: $\forall A, B, C : r3(B, A) \wedge r3(C, B) \wedge r2(A) \rightarrow r26(A, C)$ to determine the truth value of the statement $r26$(Alina, Luisa).
From the facts, we know that $r3$(Alina, David) and $r3$(David, Luisa) (F34 and F52). Therefore, we can substitute A = Alina, B = David, and C = Luisa into L23 to get:
$r3$(David, Alina) $\wedge$ $r3$(Luisa, David) $\wedge$ $r2$(Alina) $\rightarrow r26$(Alina, Luisa)
$r3$(Alina, David) is true (F34) and $r2$(Alina) is also true (F12). Therefore, we can simplify the above expression to:
$r3$(Luisa, David) $\rightarrow r26$(Alina, Luisa)
$r3$(Luisa, David) is true (F52), so we can conclude that $r26$(Alina, Luisa) is true.
Therefore, the answer is True.

## K.2 FEW-SHOT-COT EXAMPLES WITH MANUALLY-DESIGNED

Statement: $r14$(Amelie, Jonathan)
Answer: We can use logical rule L11: $\forall A, B, C, D, E : r3(B, A) \wedge r3(B, C) \wedge r3(C, D) \wedge r3(D, E) \wedge r2(A) \rightarrow r14(A, E)$. We can substitute A = Amelie and E = Jonathan to get: $r3(B, Amelie) \wedge r3(B, C) \wedge r3(C, D) \wedge r3(D, Jonathan) \wedge r2(Amelie) \rightarrow r14(Amelie, Jonathan)$. Now, we need to find all facts that match the conditions $r3$(B, Amelie), $r3$(B, C), $r3$(C, D), $r3$(D, Jonathan) and $r2$(Amelie). We have:
F41: $r3$(Fabian, Amelie)
F47: $r3$(Elena, Amelie)
F56: $r3$(Patrick, Jonathan)
F61: $r3$(Emilia, Jonathan)
F7: $r2$(Amelie)
We can see that F41 and F47 match the condition $r3$(B, Amelie) with $B$ being Fabian and Elena, respectively. F56 and F61 match the condition $r3$(D, Jonathan) with $D$ being Patrick and Emilia, respectively. F7 matches the condition $r2$(Amelie). Therefore, we have two possible ways of assigning values to B, C, and D:
(1) B = Fabian, D = Patrick.
(2) B = Elena, D = Emilia.
(3) B = Fabian, D = Emilia.

Table 8: The deductive reasoning results of one Symbolic Tree KG with ChatGPT. Results are in %.

|  | Auto-Randomly | Auto-Filter | Manually-Designed |
|---|---|---|---|
| $KG_1$ | 55.6 | 53.6 | 40.7 |

(4) B = Elena, D = Patrick.
Using the values of B and D from the possible assignments above, we can find the value of C that satisfies the conditions $r3$(B, C) and $r3$(C, D). For assignment (1), we have:
F40: $r3$(Fabian, Thomas)
F50: $r3$(Thomas, Patrick)
Since all the conditions are satisfied, we can conclude that the statement $r14$(Amelie, Jonathan) is True if we assign B = Fabian and C = Thomas and D = Patrick. Thus, the output is True.

Statement: $r31$(Philipp, Nina)
Answer: We can use logical rule L28: $\forall A, B, C, D : r3(B, A) \land r3(C, B) \land r3(C, D) \land r1(A) \rightarrow r31(A, D)$. We can substitute A = Philipp and D = Nina to get: $r3(B, Philipp) \land r3(C, B) \land r3(C, Nina) \land r1(Philipp) \rightarrow r31(Philipp, Nina)$. Now, we need to find all facts that match the conditions $r3$(B, Philipp), $r3$(C, B), $r3$(C, Nina), and $r1$(Philipp). We have:
F57: $r3$(Patrick, Philipp)
F62: $r3$(Emilia, Philipp)
F49: $r3$(Thomas, Nina)
F52: $r3$(Luisa, Nina)
We can see that F57 and F62 match the condition $r3$(B, Philipp) with $B$ being Patrick and Emilia, respectively. F49 and F52 match the condition $r3$(C, Nina) with $C$ being Thomas and Luisa, respectively. Therefore, there is only one single way of assigning values to B and C.
B = Patrick and C = Thomas. We can substitute B and C to get: $r3(Patrick, Philipp) \land r3(Thomas, Patrick) \land r3(Thomas, Nina) \land r1(Philipp) \rightarrow r31(Philipp, Nina)$. Since all the conditions are satisfied, we can conclude that the statement $r31$ (Philipp, Nina) is True if we assign B = Patrick and C = Thomas. Thus, the output is True.

Statement: $r8$(Leonie, Nico)
Answer: We can use logical rule L5: $\forall A, B, C : r3(A, B) \land r3(B, C) \land r2(A) \rightarrow r8(A, C)$. We can substitute A = Leonie and C = Nico to get: $r3(Leonie, B) \land r3(B, Nico) \land r2(Leonie) \rightarrow r8(Leonie, Nico)$. Now, we need to find all facts that match the conditions $r3$(Leonie, B), $r3$(B, Nico), and $r2$(Leonie). We have:
F43: $r3$(Leonie, Emily)
F38: $r3$(Emily, Nico)
F39: $r3$(Konstantin, Nico)
F24: $r2$(Leonie)
We can see that F43 matches the condition $r3$(Leonie, B) with $B$ being Emily. and F38 matches the condition $r3$(B, Nico) with $B$ being Emily. Therefore, we can assign B = Emily. Also, F24 matches the condition $r2$(Leonie). Therefore, we can conclude that the statement $r8$(Leonie, Nico) is True if we assign B = Emily. Therefore, the output is True.

Statement: $r7$(Patrick, Alina)
Answer: We can use logical rule L4: $\forall A, B : r3(A, B) \land r1(A) \rightarrow r7(A, B)$. We can substitute A = Patrick and B = Alina to get: $r3(Patrick, Alina) \land r1(Patrick) \rightarrow r7(Patrick, Alina)$. Now, we need to find all facts that match the conditions $r3$(Patrick, Alina) and $r1$(Patrick). We have:
F55: $r3$(Patrick, Alina)
F9: $r1$(Patrick)
We can see that F55 matches the condition $r3$(Patrick, Alina), and F9 matches the condition $r1$(Patrick). Therefore, we can conclude that the statement $r7$(Patrick, Alina) is True based on L4. Thus, the output is True.

## L    REASONING RESULTS OF AFTER HUMAN SELECTION

In this section, we perform all experiments after human selection, providing the LLMs with only the relevant facts related to the predicted fact.

For example, we have facts:

```
F1: Lea is female.
F2: Philipp is male.
F3: Leonie is female.
F4: Patrick is parent of Alina.
F5: Patrick is parent of Philipp.
F6: Nico is parent of David.
F7: Nico is parent of Lea.
F8: Laura is parent of Fabian.
F9: Laura is parent of Felix.
```

If we predict the unknown fact "Philipp is the brother of Alina", we filter out the irrelevant facts and retain only the related facts:

```
F1: Philipp is male.
F2: Patrick is parent of Alina.
F3: Patrick is parent of Philipp.
```

## L.1 COMPARISON OF REMOVING RULES AND SEMANTICS

To explore the ability of LLMs to utilize internal and external knowledge, we conduct an additional experiment where we provided LLMs with only the relevant facts related to the predicted fact. We compared the performance of *Removing rules* (leveraging internal knowledge) with *Semantics* (providing external logical rules). Surprisingly, from Table 9. we find that *Removing rules* performed better than *Semantics*. This suggests that LLMs possess the necessary internal knowledge to support answering questions and reasoning tasks, and leveraging internal knowledge is more effective for reasoning than relying on external logical rules.

|           | Remove rules | Provide rules |
|-----------|--------------|---------------|
| Zero-Shot | 80.5         | 77.1          |

Table 9: Comparison of Removing and providing rules of deductive reasoning. Results are in %.

LLMs' output examples are provided below:

**Semantics**: According to the given logical rule, for A to be a granddaughter of D, B must be the parent of A and C must be the parent of B. However, there is no direct connection between Emilia and Lea in the given facts. Although Emilia is the parent of Alina, who is the parent of Lea, Emilia is not the parent of Lea's parent. Therefore, we cannot conclude that Lea is Emilia's granddaughter based on the given facts and logical rule.

**Removing rules**: Based on the given facts, Emilia is the parent of Alina, who is the parent of Lea. Therefore, Emilia is the grandmother of Lea, making Lea Emilia's granddaughter.

From the example, we can observe that when relying on external logical rules, LLMs need to strictly adhere to the reasoning process, which can be more challenging for LLMs to predict unknown answers compared to utilizing the commonsense knowledge already contained within LLMs. This suggests that leveraging the internal knowledge of LLMs can be more effective for reasoning tasks.

## L.2 REASONING RESULTS AFTER HUMAN SELECTION

We conduct deductive and inductive reasoning experiments to examine the performance of LLMs when only provided with the relevant facts related to the predicted fact. The results are presented in Table 10. They demonstrate that after selecting useful information, LLMs perform reasoning tasks more effectively. This finding suggests that LLMs face challenges when processing excessively long in-context information. Selecting relevant facts helps to reduce the memorization load on LLMs and enables them to focus on the most relevant information for reasoning, leading to improved performance.

|  |  | Zero-Shot | Zero-Shot-CoT |
|---|---|---|---|
| **Deductive** | standard | 52.6 | 56.1 |
|  | removing irr | 55.7 | 63.0 |
| **Inductive** | standard | 7.14 | 7.14 |
|  | removing irr | 67.9 | 67.9 |

Table 10: Reasoning results after removing irrelevant information. Results are %.

Table 11: The reasoning results of Symbolic Tree (ChatGPT). Results are in %.

| Category | Baseline | deduction | induction | abduction |
|---|---|---|---|---|
| **Logic language** | Zero-Shot | 52.6 | 7.14 | 1.95 |
|  | Zero-Shot-CoT | 56.1 | 7.14 | 3.57 |
|  | Few-Shot-CoT | 53.7 | - | 13.3 |
| **Natural language** | Zero-Shot | 50.6 | 3.57 | 3.90 |
|  | Zero-Shot-CoT | 50.2 | 7.14 | 1.95 |
|  | Few-Shot-CoT | 51.9 | - | 8.13 |

## M  REASONING WITH NATURAL LANGUAGE

In this section, we conducted experiments using the *Symbols* setting with deduction, induction, and abduction on a Symbolic Tree dataset expressed in natural language. The results are presented in Table 11. We observed that, in general, LLMs performed better when using logical language compared to natural language.

## N  REASONING RESULTS OF TWO REPRESENTATIONS

For the Symbolic Tree dataset, facts and rules can be represented as logic language and natural language text as the input of LLMs. For example, the fact "motherOf(Alice, Bob)" can be represented as "Alice is Bob's mother"; the fact "r1(Alice, Bob) can be represented as "Alice is r1 of Bob"; the rule "$\forall x, y : \text{parentOf}(x, y) \rightarrow \text{childOf}(y, x)$" can be represented as "If x is parent of y, then y is parent of x.". Through numerous trials, we find that for the *Symbols* or *Counter-CS* setting, LLMs tend to perform better when using logic language representations. Conversely, for the *Semantics* setting, LLMs tend to perform better when using natural language text. The results are presented in Table 12. These observations suggest that natural language representations better stimulate the semantic understanding capabilities of LLMs, while logical language representations are more conducive to symbolic reasoning.

|  |  | Zero-Shot | Zero-Shot-CoT |
|---|---|---|---|
| **Symbols** | logic | 52.6 | 56.1 |
|  | natural language | 49.0 | 51.1 |
| **Semantics** | logic | 61.4 | 61.9 |
|  | natural language | 69.3 | 64.3 |
| **Counter-CS** | logic | 52.6 | 54.4 |
|  | natural language | 48.7 | 48.3 |

Table 12: Deductive reasoning results in different representations. Results are %.

## O  REASONING WITH GARBLED SYMBOLS

In this section, we randomly sample 4-8 letters to construct a garbled symbols word as each relation label. However, because LLMs process text by tokens, common sequences of characters found in

Table 13: The deductive reasoning results of one Symbolic Tree KG with ChatGPT. Results are in %.

|  | Zero-Shot | Zero-Shot-CoT | Few-Shot-CoT | Zero-Plus-Few-Shot-CoT |
|---|---|---|---|---|
| garbled symbols | 52.4 | 54.4 | 55.5 | 52.9 |
| single-token symbols | 52.5 | 53.2 | 53.9 | 52.3 |
| ID symbols | 54.5 | 57.0 | 57.2 | 52.3 |
| short rules | 52.1 | 51.1 | 52.9 | 54.6 |
| long rules | 54.5 | 57.0 | 57.2 | 52.3 |

the text, this random label is often tokenized as a single letter by LLMs. This leads to difficulty for LLMs to comprehend the word-level meaning of the label, harming the performance of reasoning. We also attempted to use single-token garbled symbols as relation labels, but similar challenges were encountered, with the labels often being interpreted as typos by LLMs, further hindering their reasoning abilities. Our experiments and case studies also demonstrate this point.

As shown in Table 13, using ID symbols (*e.g.*, "r1") as relation labels performs better than using garbled symbols or single-token symbols.

Some examples with garbled symbols:

Statement: $ckxovsaq$(Alina, Philipp)
Answer: We can use logical rule L1 to make a deduction about Alina and Philipp. L1 states that if there is a person B who is a friend of both A and C, and A is not outgoing, then A trusts C. We know that Alina is not outgoing (F12: $ntoea$(Alina)) and that Philipp is a friend of Emilia (F62: $yufevh$(Emilia, Philipp)). However, we do not have any information about Alina and Emilia's friendship. Therefore, we cannot use L1 to make a deduction about $ckxovsaq$(Alina, Philipp). The answer is False.

Statement: $nyheg$(Emily, Nico)
Answer: We can use logical rule L3: $\forall A, B : yufevh(A, B) \wedge ntoea(A) \rightarrow nyheg(A, B)$ to determine if the statement is true or false. This rule states that if A is not outgoing and there is a friendship relation between A and B, then A is jealous of B. From the facts, we know that Emily is not outgoing (F17: $ntoea$(Emily)) and she is friends with Nico (F38: $yufevh$(Emily, Nico)). Therefore, we can apply L3 to conclude that $nyheg$(Emily, Nico) is true. Therefore, the answer is True.

Statement: $phnd$(Luisa, Jonathan)
Answer: We can use logical rule L5: $\forall A, B, C : yufevh(A, B) \wedge yufevh(B, C) \wedge ntoea(A) \rightarrow phnd(A, C)$ to determine if the statement is true or false. This rule states that if A is not outgoing and there is a path from A to C through B, then A is phoned by C. From the given facts, we know that Luisa is not outgoing (F8: $ntoea$(Luisa)). We also know that there is a path from Luisa to Jonathan through Patrick (F53: $yufevh$(Luisa, Patrick) and F56: $yufevh$(Patrick, Jonathan)). Therefore, we can apply L5 with A = Luisa, B = Patrick, and C = Jonathan to get: $phnd$(Luisa, Jonathan) is true. Therefore, the answer is True.

Some examples with single-token symbols:

Statement: $iance$(Patrick, Lea)
Answer: We can use logical rule L4: $\forall A, B : inely(A, B) \wedge atis(A) \rightarrow lesai(A, B)$, which states that if A is an artist and B is in A's inner circle, then B is a muse of A. We can also use fact F53: $inely$(Luisa, Patrick), which means that Luisa is in Patrick's inner circle. Therefore, we can conclude that $lesai$(Patrick, Luisa) is true.
Next, we can use logical rule L6: $\forall A, B, C : inely(A, B) \wedge inely(B, C) \wedge atis(A) \rightarrow iance(A, C)$, which states that if A is an artist, B is in A's inner circle, and C is in B's inner circle, then A and C have an artistic alliance. We can use fact F35: $inely$(Alina, Lea), which means that Alina is in Lea's inner circle. Therefore, we can conclude that $iance$(Alina, Lea) is true.
Finally, we can use logical rule L1: $\forall A, B, C : inely(B, A) \wedge inely(B, C) \wedge uitka(A) \rightarrow reib(A, C)$, which states that if A is a writer, B is in A's inner circle, and B is also in C's inner circle, then A and C have a literary rivalry. We can use fact F54: $inely$(Patrick, Samuel), which

means that Samuel is in Patrick's inner circle. We can also use fact F35: $inely$(Alina, Lea), which means that Alina is in Lea's inner circle. Therefore, we can conclude that $reib$(Patrick, Lea) is true. Since Patrick has a literary rivalry with Lea, but not an artistic alliance, we can conclude that the statement $iance$(Patrick, Lea) is False.

Statement: $icers$(Jonathan, David)
Answer: We can use logical rule L10: $\forall A, B, C, D : inely(B, A) \wedge inely(B, C) \wedge inely(C, D) \wedge atis(A) \rightarrow icers(A, D)$, which states that if A is an artist and B, C, and D are in a line with B in the middle, then A influences D.
We can use the following facts: - $inely$(Jonathan, Alina) (F55)
- $inely$(Alina, David) (F34)
From these facts, we can conclude that Jonathan, Alina, and David are in a line with Alina in the middle. Also, we know that Alina is an artist. Therefore, we can apply L10 to conclude that Alina influences David.
Since Alina influences David, we can use L4: $\forall A, B : inely(A, B) \wedge atis(A) \rightarrow lesai(A, B)$, which states that if A influences B and A is an artist, then B is a musician. Therefore, we can conclude that David is a musician.
Therefore, the statement $icers$(Jonathan, David) is False, since David is not an ice skater.

Statement: $indr$(Florian, Nina)
Answer: We can use logical rule L24: $\forall A, B : inely(B, A) \wedge atis(A) \rightarrow indr(A, B)$, which states that if A is an artist and B is a fan of A, then B is indirectly a fan of all the artists that A is a fan of. From the given facts, we know that $atis$(Florian) and $inely$(Emilia, Nina). Therefore, we can conclude that Nina is indirectly a fan of Florian, since Emilia is a fan of Florian and Nina is a fan of Emilia.
Thus, the answer is True.

|  | Zero-Shot | Zero-Shot-CoT |
|---|---|---|
| replacing none | 69.3 | 66.1 |
| replacing ent | 63.6 | 58.9 |
| replacing rel | 54.5 | 54.5 |
| replacing ent & rel | 57.5 | 55.6 |

Table 14: Comparison of replacing entity labels in deductive reasoning experiment (ChatGPT). Results are in %.

Table 15: The results of Symbolic Tree (Llama2-13B-chat). Results are in %.

|  | $r1$($e1$,$e2$)$ | $r1(e1,e2)$ | $r1$(e1,e2) | r1($e1$,$e2$) | r1(e1,e2) |
|---|---|---|---|---|---|
| training | - | - | 100 | 100 | 100 |
| testing | - | - | 100 | 100 | 100 |
| rep_ent_with_words | - | - | 99.7 | 100 | 99.4 |
| rep_rel_with_commonsense | - | - | 79.1 | 100 | 60.5 |
| rep_rel_with_counter | - | - | - | 87.4 | 65.4 |
| rep_all_with_words | - | - | 64.9 | 92.4 | 58.3 |
| FOLIO | - | - | 70.4 | 70.4 | 69.6 |

## P   MORE REASONING RESULTS OF SYMBOLIC TREE

We experiment with 10 sampled trees and report the average results in the main body. In this section, we provide the reasoning results of each sampled Symbolic Tree, presented in Table Tabs. 16 to 18.

## Q   ABDUCTIVE REASONING ON SMALLER DATASETS

We use smaller Symbolic Tree datasets to conduct the abductive reasoning experiment, which contains about 12 entities and 100 facts. The results are provided in Table 19. We compare *Symbols* and

Table 16: The deductive reasoning results of Symbolic Tree datasets. Results are in %.

| Category | Model | Baseline | $S_1$ | $S_2$ | $S_3$ | $S_4$ | $S_5$ | $S_6$ | $S_7$ | $S_8$ | $S_9$ | $S_{10}$ | Avg. |
|---|---|---|---|---|---|---|---|---|---|---|---|---|---|
| Symbols | Random | - | 52.4 | 50.8 | 51.3 | 50.2 | 49.3 | 49.1 | 48.1 | 52.3 | 48.4 | 49.0 | 50.1 |
| | ChatGPT | Zero-Shot | 52.6 | 50.6 | 50.5 | 49.5 | 55.2 | 53.1 | 50.0 | 53.4 | 56.6 | 54.0 | 52.6 |
| | | Zero-Shot-CoT | 56.1 | 57.0 | 55.4 | 57.0 | 54.5 | 56.1 | 55.5 | 56.9 | 50.0 | 58.0 | 55.7 |
| | | Few-Shot-CoT | 53.7 | 56.9 | 55.2 | 54.4 | 55.1 | 52.0 | 54.0 | 55.8 | 56.8 | 54.5 | 54.8 |
| | | Zero-Plus-Few-Shot-CoT | 53.7 | 53.6 | 55.4 | 51.4 | 54.0 | 50.9 | 54.0 | 54.2 | 58.4 | 54.5 | 54.0 |
| Semantics | ChatGPT | Zero-Shot | 70.0 | 64.8 | 70.4 | 65.8 | 61.4 | 63.8 | 65.8 | 67.4 | 63.0 | 68.9 | 66.1 |
| | | Zero-Shot-CoT | 66.7 | 64.8 | 64.6 | 64.1 | 64.4 | 67.2 | 66.5 | 66.7 | 64.6 | 65.4 | 65.5 |
| | | Few-Shot-CoT | 71.8 | 70.4 | 63.9 | 69.2 | 66.7 | 59.3 | 68.7 | 68.3 | 67.9 | 64.4 | 67.1 |
| | | Zero-Plus-Few-Shot-CoT | 71.3 | 67.8 | 66.6 | 69.5 | 65.7 | 60.9 | 68.4 | 68.3 | 66.5 | 66.8 | 67.2 |
| | Logic-based | - | 100 | 100 | 100 | 100 | 100 | 100 | 100 | 100 | 100 | 100 | 100 |

Table 17: The inductive reasoning results of Symbolic Tree datasets. Results are in %.

| Category | Model | Baseline | $S_1$ | $S_2$ | $S_3$ | $S_4$ | $S_5$ | $S_6$ | $S_7$ | $S_8$ | $S_9$ | $S_{10}$ | Avg. |
|---|---|---|---|---|---|---|---|---|---|---|---|---|---|
| Symbols | ChatGPT | Zero-Shot | 7.14 | 9.09 | 3.57 | 7.14 | 4.54 | 14.3 | 4.54 | 7.14 | 3.57 | 0.0 | 6.10 |
| | | Zero-Shot-CoT | 7.14 | 7.14 | 3.57 | 14.3 | 14.3 | 7.14 | 3.57 | 0.0 | 14.3 | 7.14 | 7.86 |
| | GPT-4 | Zero-Shot | 14.3 | 10.7 | 10.7 | 10.7 | 7.14 | 7.14 | 10.7 | 7.14 | 7.14 | 7.14 | 9.28 |
| | | Zero-Shot-CoT | 21.4 | 7.14 | 17.9 | 7.14 | 3.57 | 7.14 | 7.14 | 7.14 | 7.14 | 3.57 | 8.93 |
| Semantics | ChatGPT | Zero-Shot | 25.0 | 32.1 | 39.3 | 39.3 | 42.9 | 39.3 | 35.7 | 32.1 | 35.7 | 42.9 | 36.4 |
| | | Zero-Shot-CoT | 25.0 | 28.6 | 35.7 | 28.6 | 35.7 | 35.7 | 28.6 | 35.7 | 39.3 | 28.6 | 32.2 |
| | GPT-4 | Zero-Shot | 53.6 | 53.6 | 50.0 | 53.6 | 50.0 | 53.6 | 50.0 | 57.1 | 53.6 | 50.0 | 52.5 |
| | | Zero-Shot-CoT | 53.6 | 57.1 | 53.6 | 53.6 | 57.1 | 53.6 | 50.0 | 53.6 | 57.1 | 50.0 | 53.9 |
| | Rule-based | - | 64.3 | 60.7 | 60.7 | 46.4 | 67.9 | 50.0 | 64.3 | 57.1 | 53.6 | 46.4 | 57.1 |

*Semantics* and find that the *Semantics* setting still outperforms the *Symbols* setting. This reinforces the hypothesis that preserving semantics enhances the reasoning capabilities of LLMs.

Additionally, abductive reasoning in a shorter context yielded better performance compared to a longer context. This suggests that the length of the context has an impact on reasoning performance. Shorter contexts make selecting relevant and useful information easier while minimizing the influence of unrelated content.

## R    REPLACING ENTITY LABELS

In this section, we conducted experiments to investigate the effects of replacing entity names (such as "Alice") with entity IDs (*e.g.*, "e1") in the context of reasoning tasks. The results are provided in Table 14. Comparing the performance of replacing relation names with replacing both entity and relation names, we observe that replacing entity names after replacing relation names had little impact on the overall performance.

Furthermore, we consider the scenario of only replacing entity names. Compared to the case of not replacing any labels, the results indicate that although replacing entity labels retains some level of semantics, it has a detrimental effect on reasoning performance. Additionally, we observed that the negative impact of decoupling the semantics of relations was more significant than that of decoupling the semantics of entities. These findings indicate a substantial portion of the semantic information is concentrated in the relation names.

## S    MULTI-SHORT RULES

Besides, a single rule can be equivalent to multiple rules. For example, the rule $\forall x, y, z$ : parentOf$(x, y) \wedge$ parentOf$(y, z) \wedge$ gender$(x, \text{female}) \rightarrow$ GrandmotherOf$(x, z)$ can be represented as $\forall x, y, z$ : parentOf$(x, y) \wedge$ parentOf$(y, z) \rightarrow$ GrandparentOf$(x, z)$, GrandparentOf$(x, z) \wedge$ gender$(x, \text{female}) \rightarrow$ GrandmotherOf$(x, z)$. We conduct the experiments with both rule representations and find single-longer rules perform better than multiple-short rules. Results are presented in Table 13.

Table 18: The abductive reasoning results of Symbolic Tree KGs. Results are in %.

| Category | Model | Baseline | $S_1$ | $S_2$ | $S_3$ | $S_4$ | $S_5$ | $S_6$ | $S_7$ | $S_8$ | $S_9$ | $S_{10}$ | Avg. |
|---|---|---|---|---|---|---|---|---|---|---|---|---|---|
| **Symbols** | **ChatGPT** | Zero-Shot | 1.95 | 0.31 | 1.07 | 1.52 | 2.36 | 1.45 | 1.06 | 0.75 | 3.1 | 1.39 | 1.50 |
| | | Zero-Shot-CoT | 3.57 | 4.08 | 5.00 | 3.03 | 3.70 | 3.77 | 5.28 | 7.55 | 7.78 | 5.21 | 4.90 |
| | | Zero-Plus-Few-Shot-CoT | 22.7 | 16.7 | 15.0 | 11.5 | 19.9 | 12.6 | 12.7 | 25.3 | 15.2 | 16.3 | 16.8 |
| **Semantics** | **ChatGPT** | Zero-Shot | 1.95 | 3.14 | 3.57 | 1.52 | 2.69 | 2.32 | 3.87 | 3.02 | 3.89 | 3.47 | 2.94 |
| | | Zero-Shot-CoT | 4.22 | 5.34 | 4.64 | 3.63 | 2.69 | 2.90 | 4.23 | 1.89 | 3.11 | 1.39 | 3.40 |
| | | Zero-Plus-Few-Shot-CoT | 17.5 | 25.2 | 22.1 | 16.7 | 16.5 | 18.0 | 22.2 | 27.2 | 22.6 | 21.5 | 20.9 |
| | **Rule-based** | - | 100 | 100 | 100 | 100 | 100 | 100 | 100 | 100 | 100 | 100 | 100 |

Table 19: The abductive reasoning results of a smaller Symbolic Tree. Results are in %.

| Category | Baseline | short context | long context |
|---|---|---|---|
| **Symbols** | ChatGPT: Zero-Shot-CoT | 9.78 | 3.57 |
| | GPT-4: Zero-Shot-CoT | 46.7 | 32.1 |
| **Semantics** | ChatGPT: Zero-Shot-CoT | 5.43 | 4.22 |
| | GPT-4: Zero-Shot-CoT | 59.8 | 31.8 |

# T OTHER REPLACEMENT

To evaluate on *Semantics* setting, we replace entities or relations with their original semantic words:

(1)replace the "r1" of fact "r1(e1, e2)" with "parentOf", denoted by *rep_rel_with_commonsense*;

(2)replace "e1" of fact "r1(e1, e2)" with "Amy", namely *rep_ent_with_words*;

(3) replace the "r1" of fact "r1(e1, e2)" with "granddaugterOf", denoted by *rep_rel_with_counter*;

(4) replace the "r1" of fact "r1(e1, e2)" with "p1" and replace the "e1" of fact "r1(e1, e2)" with "c1", denoted by *rep_all_with_other_symbols*;

(5) replace the "r1" of fact "r1(e1, e2)" with "parentOf" and replace the "e1" of fact "r1(e1, e2)" with "Amy", denoted by *rep_all_with_words*;

# U PERFORMANCE OF FT-MODEL ON DIFFERENT REPRESENTATIONS

We fine-tune the Llama2-13B-chat model using various symbolic representations of the Symbolic Tree, specifically including the addition of the '$' symbol around entities or relations, considering that most symbols appear in latex code. The datasets maintain the same size of training data. Due to training resource limitations, each dataset is trained for only one epoch. In fact, as concluded in section 3.2.2, a proficient learner given a sufficient volume of data should be able to fit the training data within one epoch. The results are presented in Table 15.

The results indicate a notable variance in the performance across different symbolic representations. Unlike human-like robust reasoning, FT-Llama2 is sensitive to query representations. According to the definitions of Section 1, the observations indicate that FT-Llama2 is not a generalizable logic reasoning reasoner.

