# OpenReview forum: "On the Paradox of Generalizable Logical Reasoning in Large Language Models"
_ICLR.cc/2024/Conference — Submitted to ICLR 2024_

### Official Review · Reviewer_qhHL · 2023-10-18

**Soundness:** 2 fair
**Presentation:** 2 fair
**Contribution:** 2 fair
**Rating:** 3
**Confidence:** 2

**Summary:**

I appreciated the related works section. However, I am not sure that the experimental design is up to the standards of a top venue like ICLR.

**Strengths:**

I appreciated the references to the literature. The questions you want to answer are interesting.

**Weaknesses:**

An experiment should try to isolate the hypothesis being tested and removing confounding factors. I am also suspicious your  conclusions; e.g., you claim "In other words, LLMs show significantly worse performance when semantics are decoupled " but without error bars, the Symbols and Semantic of Table 2 look the same to me. (I don't think that you should generalize from a single example (Symbol tree), where the results don't hold for other example (ProofWriter).

Much deductive reasoning is combinatorially difficult, and is difficult even for humans. I'm surprised humans can do the examples B.2 well. (I thought that the psychology literature results are that humans are not good at logical reasoning -- but I am not a psychologist).

You have a strange definition of abduction. It looks like "find a proof" (from the only example given in Figure 1 and on page 4), where the sisterOf and motherOf are given as facts. Abduction in logic means to find a plausible explanation:
e.g. why did someone cough? An explanation is that they have asthma. Another explanation is they have a cold. The system does not know wether they have asthma or a cold. It is difficult to judge the correctness of an answer.

(see also questions)

**Questions:**

Why only "zero-shot Symbols" for humans? Who are the humans used? Are they trained in logic? (Appendix F1 doesn't provide much details being being diverse college/graduate students). This is not up the standards of a good human-experiment to make any conclusions. Were the humans all the same? Why isn't there a range? The examples you gave on p 36 for humans were for the semantics case (unless I misunderstood appendix I). I wish your appendices gave a few complete examples rather than more abstract examples; what was the actual input and what was the actual output for the computers and the humans?

For the Symbolic Tree dataset, is the LLM/human told it is a closed-world dataset? What is a false fact? None of the examples seem to rely on negative facts, and none of the examples in appendix B have any. Are there negative examples as well as positive examples?

How hard are the induction tasks? What bias do you assume for the correct answer?  Why should we believe the "gold proofs" are correct?

Can you explain why In Table 1, for ChatGPT Zero-Shot-CoT is better than Zero-Shot for deduction, but in Table 2, it is worse for all depths? Does CoT help?

For "Paradox 1" and "Paradox 2" - why are they paradoxes? Maybe use "hypothesis"?

---

> ### Author Response · Authors · 2023-11-17
> **Reply to Reviewer qhHL (part 1/3)**
>
> Thank the reviewer for their comments.
>
> > W1. concern about the experimental results and conclusion
>
> We have added error bars to the ChatGPT results in the revised version. Due to the cost of GPT-4, we do not provide its error bars. As seen in Table 1, the semantic performance under error bars is still significantly worse than symbolic performance.
>
> As for Table 2, we observed that the symbol version performs comparably to the semantic version, primarily because the semantics in ProofWriter are irrelevant to commonsense. Essentially, tasks which require less intricate semantic understanding or minimal dependence on common sense knowledge are not significantly impacted by the presence or absence of semantics in the language models' performance, as highlighted in blue.
>
> To further validate our finding that Language Learning Models serve as in-context semantic reasoners rather than symbolic reasoners, we introduced a counter-commonsense setting and conducted more ablation studies, as shown in Table 3. In addition to deduction tasks, we executed comprehensive experiments in induction and abduction, to provide a holistic illustration of our conclusions.
>
>
> > W2. concern about the results of human study
>
>
> Piaget's theory of formal operations (https://en.wikipedia.org/wiki/Piaget' s_theory_of_cognitive_development) points out that individuals in early to middle adolescence are capable of hypothetical and deductive reasoning that involves assumptions with no necessary relation to reality, also including counterfactual thinking. While not all individuals use formal reasoning in their daily lives [1], most humans possess this skill to some degree. And they possess the cognitive ability to engage in formal reasoning when required. In addition to Piaget's theories, it has been suggested by citations [1][2][3] that humans possess symbolic reasoning abilities. To rigorously verify this claim, we will conduct a human study to measure human performance in these tasks.
>
> [1] https://cbmm.mit.edu/research/2013-2018-research/exploring-future-directions/learning-and-reasoning-symbolic-domains
> [2] https://www.frontiersin.org/articles/10.3389/fpsyg.2014.00275/full
> [3] https://tigerprints.clemson.edu/all_dissertations/2932/
>
> > W3. definition of abduction
>
> Abduction in logic traditionally involves generating plausible explanations for a given observation. Indeed, our approach to abductive reasoning aligns with this definition, particularly within the framework of first-order logic. Based on the definition of Logic-based abduction (https://en.wikipedia.org/wiki/Abductive_reasoning), a proof-theoretical abduction method for first-order logic have been proposed for finding explanations. In first-order logic contexts, abduction typically involves reasoning that is more definitive and less speculative than in commonsense reasoning scenarios. Most abductive reasoning in this domain tends to yield plausible conclusions.
>
> Moreover, Regarding the specific task settings, our methodology follow the approach "Explanation Generation with Theory" described in the paper https://arxiv.org/pdf/2303.12023.pdf. This approach involves generating a new hypothetical fact h such that, when combined with a given theory C, can prove an observation O, i.e., $C \cup {h} |=O$. We also reference "Explanation Classification," where the task is to choose the most plausible hypothesis from given alternatives to explain sequential observations. Considering the complexity of generating new facts and the necessity of a feasible evaluation framework, we adapted our task to involve selecting the most plausible facts from existing ones. This adaptation allows us to maintain the rigors of first-order logic abduction while ensuring that our tasks are feasible and effectively evaluative.
>
> > Q1. the details of human study
>
> We apologize for any confusion regarding Appendix I, which was designed to examine the influence of irrelevant information on logical reasoning. We will provide more details about the human study in the revised paper.
>
> We recruited 11 participants from diverse science and engineering backgrounds, including computer science, electronics, artificial intelligence, and automation. Although they have basic understanding of simple logic concepts, they are not experts in logical reasoning. Therefore, we provided them with task instructions that explained the concepts of deduction, induction, and abduction, aligned with the illustrations and definitions of logical reasoning presented in Section 3 of our paper.
>
> We then presented them with 18 specific tasks, including six tasks for each deductive, inductive, and abductive reasoning type. Each task closely resembled the zero-shot prompts given to LLMs. We refer to this setting as "zero-shot" because we did not provide any further specific examples to help participants understand the tasks. The examples can be found in Appendix I and detailed materials have been uploaded in supplementary.

---

> ### Author Response · Authors · 2023-11-17
> **Reply to Reviewer qhHL (part 2/3)**
>
> > Q2. concern about closed-world setting
> In our experimental setting, Symbolic Tree is a closed-world dataset, meaning that any facts not presented or inferable in the dataset are assumed to be false. In other words, when the language model (LLM) outputs an "Unknown" answer, we interpret it as "False".
>
> Consider the following false case example:
> ```
> Statement: $r27$(patrick, marko)
> Answer: We can use logical rule L24, which states that if there is a relationship of type 3 between B and A, and a relationship of type 3 between C and B, and A has property 1, then A has relationship 27 with C.
> From the facts, we know that there is a relationship of type 3 between Patrick and Theodor (F43), and a relationship of type 3 between Theodor and Marko (F44). However, we do not have any direct information about a relationship of type 1 involving Patrick. Therefore, we cannot use L24 to prove $r27$(patrick, marko).
> Thus, we cannot determine whether the statement $r27$(patrick, marko) is True or False using the given logical rules and facts.
> ```
>
> > Q3. the complexity and "gold proof" of induction tasks
>
> In our induction tasks, we utilized the Symbolic Tree dataset, which consists of approximately 99 basic facts including gender information ("r43" and "r44"), "parentOf" ("r1"), and "inverse_parentOf" ("r45") relationships. Additionally, there are about 5-15 inferred facts related to the unknown (to be predicted) rule.
>
> Note that a single rule can be equivalent to multiple rules. For example, the rule $\forall x,y,z: \text{parentOf}(x, y) \land \text{parentOf}(y, z) \land \text{gender}(x, \text{female}) \rightarrow \text{GrandmotherOf}(x,z)$ can be represented as two separate rules: $\forall x,y,z: \text{parentOf}(x, y) \land \text{parentOf}(y, z) \rightarrow \text{GrandparentOf}(x,z)$ and $\forall x,y,z: \text{GrandparentOf}(x,z) \land \text{gender}(x, \text{female}) \rightarrow \text{GrandmotherOf}(x,z)$. To simplify the induction evaluation process, we rewrite all logical rules by including only the "parentOf" and "gender" relations in the rule body. Doing so ensures that each inferred relation is implied by a single logical rule. We provide a rule template to define the rule search space, as detailed in Appendix A.2, and specific examples can be found in Appendix E.3.
>
>
> > Q4. Can you explain why In Table 1, for ChatGPT Zero-Shot-CoT is better than Zero-Shot for deduction, but in Table 2, it is worse for all depths? Does CoT help?
>
> Indeed, in Table 1, we observe that the performance of ChatGPT in the Zero-Shot-CoT setting does not show a marked improvement over the Zero-Shot setting, especially when considering error bars. This suggests that the addition of CoT does not significantly enhance the model's ability to solve deduction tasks.
>
> Table 2 presents an even more pronounced discrepancy, where CoT settings perform worse than the Zero-Shot setting. We hypothesize two main reasons for this observed trend:
>
> 1. The CoT settings perform worse than the zero-shot setting with a higher frequency of "Cannot be determined" answers. We speculate that this may arise from the LLMs struggling to generate explicit reasoning traces effectively.
> 2. Utilizing a step-by-step explicit reasoning trace might lead to error propagation, resulting in worse performance than directly obtaining the answer. In addition, this effect seems to be more pronounced in tasks with semantic contexts, as seen in Table 2. The discrepancy between Zero-Shot-CoT and CoT is greater. This observation might indicate that step-by-step reasoning could amplify the distracting effect of weird semantics, such as "The squirrel needs the dog."
>
> Consequently, CoT might not always be helpful for reasoning tasks that involve in-context new knowledge for models with weaker reasoning abilities like ChatGPT.

---

> > ### Author Response · Authors · 2023-11-17
> > **Reply to Reviewer qhHL (part 3/3)**
> >
> > > Q5. For "Paradox 1" and "Paradox 2" - why are they paradoxes? Maybe use "hypothesis"?
> >
> >
> > We would like to explore and highlight two key paradoxical observations in the capabilities of Large Language Models (LLMs) with respect to logical reasoning.
> >
> > **Paradox 1** revolves around the unexpected discrepancy in LLMs' performance in logical reasoning tasks. While LLMs like GPT-4 show high accuracy (91.1%) in tasks rich in semantics, their performance significantly declines when semantic cues are replaced with symbolic representations. This observation is paradoxical because, according to the formal definition of logical reasoning, LLMs' reasoning ability should rely purely on the form or syntax of premises and conclusions, independent of semantic content. Yet, our findings indicate that LLMs heavily rely on semantics, suggesting that current logical reasoning tasks may not fully reflect the true extent of LLMs' formal reasoning capabilities.
> >
> > **Paradox 2** emerges from our fine-tuning experiments on symbolic reasoning tasks. While supervised fine-tuning narrows the performance gap between semantic and symbolic settings, enabling LLMs to perform logical reasoning seemingly agnostic to semantics, their ability to generalize to unseen logical rules remains limited. This indicates that LLMs primarily use template matching mechanisms for reasoning, rather than engaging in genuine formal reasoning processes. This paradox challenges the common perception that fine-tuning effectively enhances LLMs' logical reasoning skills, suggesting that the improvement may be superficial rather than indicative of true reasoning abilities.
> >
> > In summary, our paper identifies two paradoxes: 1) **LLMs' reliance on semantics rather than engaging in formal reasoning, despite seeming logical reasoning proficiency**, and 2) **the superficial nature of improvements in logical reasoning through fine-tuning, which fails to achieve deep generalization to new rules**. These paradoxes question the true logical reasoning capabilities of LLMs and emphasize the need for more nuanced understanding and development in this area.

---

### Official Review · Reviewer_oqtQ · 2023-11-04

**Soundness:** 2 fair
**Presentation:** 3 good
**Contribution:** 3 good
**Rating:** 5
**Confidence:** 3

**Summary:**

This paper provided an in-depth evaluation of the reasoning capability of large language models through the language of first-order logic:

- For reasoning tasks, deductive, inductive, and abductive reasoning are conducted.
- For the representation of language, the pure logic language, natural language, and some corner cases of language inputs with garbled symbols.
- For LLMs, the in-context learning of ChatGPT and GPT4 as well as the fine-tuning of Llama-13B is discussed.

By conducting investigations over logical reasoning, the authors identified two major findings and multiple minor findings from their empirical results. It is suggested that the logical reasoning of the large language model is still a challenging task. The good performance of large language models is either mixed results of the semantics of the language, templating matching of the known knowledge, and eventually, the strict logical reasoning ability.

**Strengths:**

It is praiseworthy that this paper justifies many aspects of logical reasoning.

The highlighted discussions include
- the gap between formal and natural language (or the symbolic or semantics referred to in this paper).
- the impact of in-context knowledge and parameterized knowledge on the commonsense and counter-commonsense settings.

Though there is no innovation from the methodological aspect,  the way of tackling this problem demonstrated by this paper will surely encourage future work.

**Weaknesses:**

Despite the impressive points that the authors intended to make, some facts might undermine the validity of the claims.

1. The first part of the claims are made by direct prompt ChatGPT/GPT4. However, some gaps between the performances are not significant.
2. Some claims are too general to be valid, please check the question part.

**Questions:**

1. For the claim
> The length of the context influences reasoning performance, as shorter contexts make it easier to select relevant and useful information while minimizing the impact of unrelated content.
The effect is also affected by the semantic language and internal knowledge. Are there any results from the symbolic logic language evaluation?

2. For the claim regarding LLM leverages template matching, why do the garbled symbols decrease the performance of deductive reasoning?

3. For the claim regarding learning to reason, why do authors expect that "fine-tuning the LLM on symbolic trees" should lead to good performance in FOLIO and RuDaS?

---

> ### Author Response · Authors · 2023-11-17
> **Reply to Reviewer oqtQ**
>
> Thank you for the reviewer's comments.
>
> > W1. The first part of the claims are made by direct prompt ChatGPT/GPT4. However, some gaps between the performances are not significant.
>
> In Table 1, considering error bars, semantic performance is still notably worse than symbolic performance.
>
> As for Table 2, we observed that the symbol version performs comparably to the semantic version, primarily because the semantics in ProofWriter are irrelevant to commonsense. Essentially, tasks which require less intricate semantic understanding or minimal dependence on common sense knowledge are not significantly impacted by the presence or absence of semantics in the language models' performance, as highlighted in blue.
>
> To further validate our finding that Language Learning Models serve as in-context semantic reasoners rather than symbolic reasoners, we introduced a counter-commonsense setting and conducted more ablation studies, as shown in Table 3. In addition to deduction tasks, we executed comprehensive experiments in induction and abduction, to provide a holistic illustration of our conclusions.
>
>
> > W2. Some claims
>
> Re W2 (1): The results of symbolic reasoning evaluation are provided in Appendix J.2. In this section, we perform deductive and inductive reasoning experiments while providing the models with only relevant facts. The findings demonstrate that LLMs are easily disrupted by irrelevant information, especially when confronted with longer contexts.
>
> Re W2 (2): We suppose the reviewer is concerned about "rep_all_with_counter" setting presented in Table 4. We would like to emphasize that even though fine-tuned LLMs can leverage template matching to make predictions, it is still inevitable that they may become distracted by prior knowledge.
>
> Re W2 (3): Our paper focuses on first-order logical reasoning abilities. Unlike the inference of machine learning which usually generalizes to the same distribution, an agent with “true” logical reasoning abilities can reason with any given information to solve new, unfamiliar problems, regardless of any prior knowledge or different knowledge representations. In other words, it can generalize across novel rules (Sec 1 Line 13). For instance, consider training on $r1(A, B) \land r2(B) \rightarrow r3(B, A)$ and testing on $r1(A, B) \land r2(B, C) \rightarrow r3(C, B)$, as depicted in Fig 3. In our experimental setting, RuDaS is the dataset featuring rules with similar forms as $\text{xx} \land \text{xx} \cdots \rightarrow \text{xx}$. FOLIO is a complex and diverse dataset for logical reasoning expressed in natural language. Through our comprehensive experiment results, we find fine-tuned LLMs on symbolic logic datasets cannot generalize to novel rules, highlighting that fine-tuning LLMs fails to achieve generalizable logic reasoning.

---

### Official Review · Reviewer_VR1b · 2023-11-07

**Soundness:** 3 good
**Presentation:** 3 good
**Contribution:** 2 fair
**Rating:** 6
**Confidence:** 3

**Summary:**

The paper investigates whether large language models (LLMs) like ChatGPT and llama have true logical reasoning abilities that can generalize across facts, rules, domains, and representations.
The authors evaluate LLMs on deductive, inductive, and abductive reasoning tasks. They find that when semantics are removed from the tasks by replacing words with symbols, the performance of LLMs drops significantly. This suggests LLMs rely on semantic associations rather than formal reasoning.
The authors fine-tune an LLM on symbolic reasoning tasks, which improves performance on unseen facts but not novel rules. This indicates the LLM uses template matching rather than truly mastering generalizable reasoning.
Overall, the paper reveals two paradoxes: 1) LLMs rely on semantics rather than formal reasoning, and 2) Fine-tuning enables shallow generalization via template matching but not true generalization to new rules.

**Strengths:**

Authors study the reasoning capabilities of LLMs and find an interesting angle. Authors report extensive negative results for future body of work to tackle.

**Weaknesses:**

Authors can be more specific regarding details. Authors can also perform additional interpretability analyses to help the community understand the failure modes.

**Questions:**

- Can authors clarify which version of GPT4 and ChatGPT they use? There are many timestamped versions with differing context length.
- Can authors provide more study on how GPT4 fails on symbols version of the task?

I read the author response and I am keeping my score.

---

> ### Author Response · Authors · 2023-11-17
> **Reply to Reviewer VR1b**
>
> Thank you for recognizing our contribution.
>
> > additional interpretability analyses about failure cases.
>
> We provided some failure cases and discussions in Appendix E. These failures can likely be attributed to interference from irrelevant information and the inclusion of hallucinated information.
>
> > the version of ChatGPT and GPT-4
>
> We used the gpt-3.5-turbo-0301 and gpt-4-0314 versions for this study. We will provide this detail in the revised paper.

---

### Official Review · Reviewer_np2z · 2023-11-07

**Soundness:** 2 fair
**Presentation:** 2 fair
**Contribution:** 2 fair
**Rating:** 5
**Confidence:** 4

**Summary:**

This paper asks whether LLMs truly understand logical reasoning or is their success on some datasets influenced by linguistic semantics and pattern matching. To this end, they experiment with linguistic logical reasoning datasets both in their original form and in pure symbolic form (e.g., relations r1, r2, ...). The find that there is a substantial performance gap between the two settings for both ChatGPT and GPT-4 in a zero/few-shot setting. Further, fine-tuning closes the gap in the symbolic setting, but there still is a large gap when asked to generalize to rules in a different domain.

**Strengths:**

The basic question that the paper is asking is important to study in order to understand the true capabilities of LLMs, especially when it comes to performing logical reasoning. Their approach of decoupling relation/fact semantics from performing deduction/induction/abduction with logical rules is interesting (though likely not the first, but I cannot pin-point a prior work at this time, so I'll give the authors the benefit of doubt).

The authors conduct a reasonably large study (within the realm of the 2-3 datasets they consider), with many side questions and analyses.

The paper situates itself in the broader NLP / AI research work, citing a LOT of (perhaps too many?) related papers.

**Weaknesses:**

The overall pitch of the paper is not as convincing as it could be. It's written like the community believes (from prior papers) that LLMs have strong logical reasoning skills, and that the current paper questions this belief and provides evidence against it. However, I don't think it's the case that the community believes logical reasoning is solved by LLMs. E.g., for the datasets considered here, even the baseline performance (in the original, so-called *Semantics* version of the tasks) is not high enough. This makes the motivation of the study relatively weak.

The pitch is also confusing because of the use of the word "paradox". What paradox exactly is being explored here? Reading the title, I was expecting to see something like: LLMs are great at X, which implies they should be great at Y too, but they fail at Y, raising a conundrum. Or some such internal conflict or conflict with commonsense expectations, that would justify the word paradox. I'm not sure what the authors have in mind for a paradox.

Overall, while I thought the study was useful, I didn't find anything subjectively surprising. It is generally accepted that LLMs --- being **language models** --- rely on many linguistic clues and prior knowledge to perform tasks. Taking away these clues is thus expected to drop performance. Similarly, training directly on the so-called *Symbolic* form should help, which they authors also find to be the case. All this is very much aligned with expectation, which makes it difficult to pin point what the new knowledge this paper would bring to the community.

There are number of additional side experiments in the paper. This, in principle, is nice. However, while reading through those section, I found the large number of questions to be somewhat distracting. At the least, the authors should try to thread the narrative better through these side experiments and analyses, and try to provide a view of them that helps support the overall message of the paper.

In summary, while it's somewhat useful to see the experiments on *Symbolic* forms of the considered datasets done, the results don't really feel different from what one might expect to see.

MINOR comments:

* The use of *Semantics* when referring to relation names but not when referring to logic is confusing. Logic, of course, by design has a very clear and unambiguous semantics. I think what you mean is *linguistic semantics* of predicate names. If so, please be sure to clarify this and emphasize the *linguistic* aspect.

* Your related work section (as well as the introduction) has a **lot** of citations, almost too many to be meaningfully valuable. E.g., near the top of page 3, you have 12+ citations around ICL, without any explanation of the connection between these prior works and what's in your paper. As a general rule, it's more valuable for the related work section to point out the few most related works AND articulate a clear connection of the current work to them, as opposed to dumping a huge list of papers just to cover every possible connection.

* The last sentence of page 2 ("Wei et al propose symbolic tuning, which ....") is very long and hard to parse.

**Questions:**

* What exactly is the paradox (i.e., some form of commonsense contradiction) that you are referring to? Or if *paradox* is not the right word, please replace it with something else.

* Looks like you forgot to discuss Table 2 (results on ProofWriter) in the main paper. What are the main take-aways from this table and how do they support your claims? E.g., it appears the going from the Semantics setting to the Symbolic setting does *not* reduce the performance of the models substantially; in fact, the performance goes up in many cases. How does this align with your claims from Table 1?

---

> ### Author Response · Authors · 2023-11-17
> **Reply to Reviewer np2z (part 1/2)**
>
> Thank the reviewer for their comments.
>
> > W1. The motivation behind the study is weak due to the discrepancy in baseline performance on logical reasoning tasks in question.
>
> Indeed, Large Language Models have achieved impressive performance on a variety of reasoning tasks[1][2], such as arithmetic and commonsense reasoning. However, the extent to which LLMs have mastered formal logical reasoning remains unclear. Previous studies have shed light on LLMs’ reasoning abilities to some extent, but a deeper exploration of their underlying reasoning mechanisms is still necessary. Our work aims to fill this gap by providing a novel perspective, going beyond mere performance metrics to understand the essence of LLMs' reasoning processes.
>
> The performance on the Symbolic Trees semantics version has achieved commendable results (91% accuracy on GPT-4). However, performance on reasoning datasets with semantics does not necessarily reflect a deep understanding or mastery of formal reasoning. When the semantics are decoupled, LLMs tend to perform much worse. Thus, our primary goal is to investigate whether LLMs possess genuine formal reasoning abilities or simply exploit superficial semantic associations between questions and answers to make intuitive predictions. This investigation is of great importance for advancing the development of artificial general intelligence (AGI).
>
> Furthermore, we also experiment with fine-tuning Llama-2 on pure symbolic reasoning tasks to bridge the gap. Although it appears to close the gap, this is achieved through template matching rather than invoking the fundamental formal reasoning abilities. We identify this phenomenon to provide a clear understanding of the boundaries within which LLMs currently solve logical reasoning tasks.
>
> Therefore, the interesting findings aim to motivate further investigation into unveiling the inner workings of black-box LLMs.
>
> [1] Wei, Jason, et al. "Chain-of-thought prompting elicits reasoning in large language models." Advances in Neural Information Processing Systems 35 (2022): 24824-24837.
> [2] OpenAI. Gpt-4 technical report, 2023. 1
>
> > W2. the use of the word "paradox" is confusing and unclear.
>
> We would like to explore and highlight two key paradoxical observations in the capabilities of Large Language Models (LLMs) with respect to logical reasoning.
>
> **Paradox 1** revolves around the unexpected discrepancy in LLMs' performance in logical reasoning tasks. While LLMs like GPT-4 show high accuracy (91.1%) in tasks rich in semantics, their performance significantly declines when semantic cues are replaced with symbolic representations. This observation is paradoxical because, according to the formal definition of logical reasoning, LLMs' reasoning ability should rely purely on the form or syntax of premises and conclusions, independent of semantic content. Yet, our findings indicate that LLMs heavily rely on semantics, suggesting that current logical reasoning tasks may not fully reflect the true extent of LLMs' formal reasoning capabilities.
>
> **Paradox 2** emerges from our fine-tuning experiments on symbolic reasoning tasks. While supervised fine-tuning narrows the performance gap between semantic and symbolic settings, enabling LLMs to perform logical reasoning seemingly agnostic to semantics, their ability to generalize to unseen logical rules remains limited. This indicates that LLMs primarily use template matching mechanisms for reasoning, rather than engaging in genuine formal reasoning processes. This paradox challenges the common perception that fine-tuning effectively enhances LLMs' logical reasoning skills, suggesting that the improvement may be superficial rather than indicative of true reasoning abilities.
>
> In summary, our paper identifies two paradoxes: 1) **LLMs' reliance on semantics rather than engaging in formal reasoning, despite seeming logical reasoning proficiency**, and 2) **the superficial nature of improvements in logical reasoning through fine-tuning, which fails to achieve deep generalization to new rules**. These paradoxes question the true logical reasoning capabilities of LLMs and emphasize the need for more nuanced understanding and development in this area.

---

> > ### Author Response · Authors · 2023-11-17
> > **Reply to np2z (part 2/2)**
> >
> > > W3. didn't find anything subjectively surprising
> >
> > Please refer to response to W1 and W2. Here, we would like to highlight our study's novelty and contributions:
> >
> >
> > We aim to investigate whether LLMs have strict formal reasoning abilities or simply exploit the superficial semantic associations between questions and answers to make intuitive predictions as well as using template matching of the known knowlege, which is crucially important to AGI development. To answer the research question, we decouple semantics from the language reasoning process and force LLMs to perform symbolic reasoning to finish three kinds of reasoning tasks (deduction/induction/abduction) as well as inverstigating the effect of fine-tuning LMs.
> >
> > * We find that when provided with symbolic knowledge in the form of facts and rules, LLMs struggle to perform simple deduction reasoning tasks. These tasks, which are relatively easy for the average human, only require following deductive rules and making logical inferences, as the human performance states shown in Table 1.
> > * We also compare the reasoning abilities of LLMs across induction and abduction tasks and find that they perform notably worse in these types of reasoning tasks compared to deduction, regardless of whether semantics or symbols are used. When semantics are decoupled, the drop in performance is even more significant. These findings highlight the considerable room for improvement in LLMs' reasoning abilities and suggest that relying solely on semantics to achieve symbolic reasoning is challenging.
> > * When fine-tuning on symbolic reasoning data, supervised fine-tuning considerably narrows the performance gap between symbolic and semantic settings, enabling LLMs to perform logical reasoning independent of semantics. However, when generalized to unseen logical rules, the fine-tuned model's performance significantly drops. This indicates that LLMs rely on a template-matching mechanism rather than a genuine formal reasoning process.
> > * In addition to the aforementioned findings, we have conducted numerous fine-grained experiments. These meticulous experimental results provide valuable insights and further strengthen our work.
> >
> > > W4. To many additional side experiments are distrcting.
> >
> > Thank you for your suggestions. While we designed additional side experiments and performed in-depth analyses in order to provide more insights, we understand that they can seem distracting. In the revision, we will re-organize the paper and strive to strengthen the connection between the side experiments and the overall message of the paper, ensuring a more cohesive narrative throughout the manuscript.
> >
> > > W5. MINOR comments:
> >
> > Re W5 (1): Thank you for pointing out the potential confusion related to the use of the term "semantics". In our paper, we indeed refer to linguistic semantics. We appreciate your feedback and will ensure to clarify and emphasize the linguistic aspect of semantics in our revised manuscript.
> >
> > Re W5 (2): We will also cut down on the number of citations and focus on providing focused explanations of the relationships and differences between our work and the selected prior works.
> >
> > Re W5 (3): We revised and restructure the sentence for better clarity and readability in the updated version of the paper.
> >
> >
> > > Q1: What exactly is the paradox?
> >
> > Please refer to our response to W2.
> >
> > > Q2: concern about the results of Table 2.
> >
> > Our discussion regarding Table 2 can be found on Page 7, Paragraph 1 under the subheading "When are semantics irrelevant to prior knowledge?". To reiterate, we observed that the symbol version performs comparably to the semantic version because the semantics in ProofWriter are irrelevant to commonsense. Essentially, tasks which require less intricate semantic understanding or minimal dependence on common sense knowledge are not significantly impacted by the presence or absence of semantics in the language models' performance.

---

### Official Review · Reviewer_nbkm · 2023-11-09

**Soundness:** 1 poor
**Presentation:** 3 good
**Contribution:** 1 poor
**Rating:** 3
**Confidence:** 4

**Summary:**

This paper provides an experimental evaluation of logical reasoning abilities of large language models. The authors first evaluate pre-trained models (GPT-4, GPT-3.5 Turbo) on logical reasoning tasks (deduction, induction, abduction) on both problems expressed with symbols and with words. They observe a large gap in some of the tasks, with even GPT-4 performing generally very poorly on induction with symbols, but much better with words that carry commonsense semantics. The authors then try fine-tuning LLaMA 2 on these tasks, observing that while it is able to match the training rules very well, it still cannot generalize to novel logical rules at inference time.

**Strengths:**

The main motivating question here is interesting, of whether there are fundamental limitations for logical reasoning in LLMs.

The authors run human comparisons, which is good to sanity check the feasibility of the tasks.

The paper tries to do a very through experimental evaluation, considering both prompting and fine-tuning, and on a range of tasks -- synthetic and existing ones from prior work.

The paper is generally easy to follow.

**Weaknesses:**

The paper has two main sets of results: with prompting and with fine-tuning. I'll give separate comments on those.

## Results with prompting

The main result here was that models performed significantly worse when symbols were used to described the rules, instead of meaningful words. In a broad sense, this question has been studied before in both papers that observed content effects in LLM reasoning that the authors cite (PrOntoQA and Dasgupta et al). In those papers, they used made-up words, whereas here the authors used short symbols (A, B, etc), but I believe the insight is the same. So, if the authors believe this says something that hasn't been said before, I don't think it came across in the paper.

Finally, the gap in these results is significantly larger in the induction and abduction tasks. The results of GPT-4 in induction (< 10% with symbols) do make me wonder whether this was due to the specific way the task was set up, or whether these are honest failures. It would be interesting if this was the latter case, since induction and abduction haven't really gotten as much attention from prior work. However, the paper has little detail about these tasks besides the general description (I have some specific questions below). It would have helped to have seen many examples of the problems and of GPT-4 responses, to make sure that the task was set up properly and that this is actually due to GPT-4 having a surprisingly bad performance. I tried to find such examples in the Appendix, but couldn't (it's possible I just missed them because there's a lot there! In that case, please point me to it).

## Results with fine-tuning

For fine-tuning, the main result was that models can internalize rules seen during training, but fail to generalize to novel rules. But if I understand, the total number of rules in training and testing was extremely small (5 in training, 3 in testing). Indeed, we would not expect to see generalization from these many examples. There are many other works showing that you do need a certain minimal level of task diversity in the training to get in-context learning in LMs [1,2]. In order to draw this strong conclusion that Transformers might have fundamental limitations to generalizing to unseen logical rules, you would have to train with a much larger number of training rules (e.g. hundreds of thousands) to make this argument convincing. If _even then_ you see a large gap, then it starts to look more like scaling the data is not leading to significant improvements, suggesting that such limitation might be more fundamental. But, at the current scale, the negative result is to be expected, and does not lead to insights into the broader motivating question.

[1] Pretraining task diversity and the emergence of non-Bayesian in-context learning for regression. Allan Raventós, Mansheej Paul, Feng Chen, Surya Ganguli, 2023
[2] Data Distributional Properties Drive Emergent In-Context Learning in Transformers. Chan et al, 2022.

**Questions:**

- Can you point to specific examples of GPT-4 failures in induction and abduction?
- Generally, your few-shot numbers seem worse than zero-shot. Why would that be the case? That might point to not giving good examples of reasoning.
-- In particular, looking at some of the examples of the appendix, I don't think they contain valid reasoning. For example, this one in Appendix H:
```
Statement: r8(Elena, Nina)
Answer: We can use logical rule L5: ∀A, B, C : r3(A, B) ∧ r3(B, C) ∧ r2(A) → r8(A, C) to deduce whether the statement r8(Elena, Nina) is true or false. [...]
```
This is not the complete problem, but I don't think this is correct. Rule L5 might only be used to prove that r8(A, C) is true (which in this case it does), but if its premises are not satisfied it does not say anything about r8(A, C) being false. Thus, this example is misleading - this reasoning template does not generalize. In fact, all of the other examples below this one proceed like this, and conclude "true". Do you also give examples of "false" cases?
- Why are there missing entries in induction in Table 1?
- What do you think are the novel insights in your experiments with words <--> symbols compared to results in prior works around content effects in LLM reasoning?
- For induction and abduction, what was the complexity of the held-out premises or rules? How did you make sure the answer was unique, since this is logically non-trivial? (in fact impossible, since formally there will be an infinite set of hypothesis in first-order logic that could be used to derive any given conclusion)
- For fine-tuning, would you be able to provide specific fine-tuning examples, besides just the prompts?

---

> ### Author Response · Authors · 2023-11-17
> **Reply to Reviewer nbkm (part 1/2)**
>
> Thank the reviewer for their comments.
> ### Results with prompting
>
> > W1. similar question about content effect in LLMs' reasoning has been studied before.
>
> While they conclude that "Language models show human-like content effects on reasoning", this is only a subordinate conclusion that suggests LLMs are in-context semantic reasoners influenced by semantic associations. However, their conclusion does not directly imply that "LLMs are not symbolic reasoners," as suggested by Paradox 1: "LLMs are in-context semantic reasoners rather than symbolic reasoners".
>
>
> According to the definition of deductive reasoning from Wikipedia (https://en.wikipedia.org/wiki/Deductive_reasoning), logical reasoning should be formal and depends only on the form or the syntax of the premises and the conclusion, irrespective of the specific contents of this argument. To this end, our task decouples semantics and focuses on pure symbolic reasoning (only provided syntax). When we feed these to LLMs, they show significantly worse performance compared to when normal semantic words are fed. This phenomenon indicates that LLMs fail to invoke the basic formal reasoning abilities of humans but instead rely on shallow semantic associations for prediction.
>
> Moreover, we conduct comprehensive experiments on various logical reasoning, including deduction, induction and abduction tasks and find that they perform notably worse in these types of reasoning tasks compared to deduction, regardless of whether semantics or symbols are used. When semantics are decoupled, the drop in performance is even more significant. These findings highlight the considerable room for improvement in LLMs' reasoning abilities and suggest that relying solely on semantics to achieve symbolic reasoning is challenging.
>
> > W2. specific examples of GPT-4 failures in induction and abduction
>
> We provided some failure cases and discussions in Appendix E. These failures can likely be attributed to interference from irrelevant information and the inclusion of hallucinated information.
>
>
>
> ### Results with fine-tuning
>
> > W3. the main result found models can internalize rules during training but fail to generalize to novel rules, with a limited number of rules tested. To convincingly conclude fundamental limitations in Transformers, a much larger number of training rules should be used.
>
> We appreciate the reviewer's insightful comment on the necessity of task diversity and the scale of training data for effective in-context learning in language models. However, we would like to emphasize that our Symbolic Tree dataset included 28 logical rules (not "5 in training, 3 in testing" as the reviewer mentioned). Moreover, in our extended dataset, RuDaS, we significantly increased the diversity with up to 1,000 different rules. Despite this expansion in rule variety, the models still struggled to generalize to novel rules.
>
> Additionally, we attempted to scale up the training data and extend training time to enhance the models' logical reasoning capabilities. As shown in Figures 5 and 6 of our paper, these efforts did not yield substantial improvements in performance. This suggests that simply scaling data and training duration may not be sufficient to overcome certain inherent limitations in transformers regarding logical rule generalization.
>
> In the "Limitations and Future Work" section of our paper, we discuss these points, acknowledging that while our findings strive to address the limitations of scalability and diversity to obtain the most convincing results possible, they might not fully realize the potential of training with large-scale and highly diverse data.
>
> With these considerations in mind, we will carefully refine and moderate our statements to ensure the precision and reliability of our paper.

---

> > ### Author Response · Authors · 2023-11-17
> > **Reply to Reviewer nbkm (part 2/2)**
> >
> > > Q1: Generally, your few-shot numbers seem worse than zero-shot. Why would that be the case? That might point to not giving good examples of reasoning. Do you also give examples of "false" cases?
> >
> > In our experimental setting, Symbolic Tree is a closed-world dataset, meaning that any facts not presented or inferable in the dataset are assumed to be false. In other words, when the language model (LLM) outputs an "Unknown" answer, we interpret it as "False". Moreover, we selected our final few-shot demonstrations where LLMs perform better to be those generated by LLMs prompted with "Let's think step by step," which include examples of both true and false cases.
> >
> > Consider the following false case example:
> > ```
> > Statement: $r27$(patrick, marko)
> > Answer: We can use logical rule L24, which states that if there is a relationship of type 3 between B and A, and a relationship of type 3 between C and B, and A has property 1, then A has relationship 27 with C.
> > From the facts, we know that there is a relationship of type 3 between Patrick and Theodor (F43), and a relationship of type 3 between Theodor and Marko (F44). However, we do not have any direct information about a relationship of type 1 involving Patrick. Therefore, we cannot use L24 to prove $r27$(patrick, marko).
> > Thus, we cannot determine whether the statement $r27$(patrick, marko) is True or False using the given logical rules and facts.
> > ```
> > Indeed, considering error bars, the results of Few-Shot-CoT are comparable to Zero-Shot-CoT. This finding suggests that the zero-shot capabilities of current large language models are approaching their few-shot learning abilities. A plausible explanation for this development is the enhanced human-alignment and instruction tuning present in current LLMs. These improvements enable the models to better understand the meaning and context of tasks in advance, allowing them to perform well even in zero-shot settings.
> >
> > > Q2: Why are there missing entries in induction in Table 1?
> >
> > This is because we did not perform a few-shot setting. Symbolic Tree dataset has a limited number of rules, and these rules exhibit a degree of interconnectedness. For instance, a rule like 'grandAuntOf' can be derived through the composition of rules like 'sisterOf' and 'grandparentOf'. The potential overlap and dependencies among the rules could skew the assessment of the model's true inductive reasoning capabilities. Consequently, we chose to only conduct a zero-shot setting where we provide facts to infer new rules without exposing the model to similar examples.
> >
> > > Q3: What do you think are the novel insights in your experiments with words <--> symbols compared to results in prior works around content effects in LLM reasoning?
> >
> > Please refer to the reponse to W1 -- similar question about content effect in LLMs' reasoning has been studied before.
> >
> > > Q4: For induction and abduction, what was the complexity of the held-out premises or rules? How did you make sure the answer was unique, since this is logically non-trivial?
> >
> >
> > In our induction tasks, we utilized the Symbolic Tree dataset, which consists of approximately 99 basic facts including gender information ("r43" and "r44"), "parentOf" ("r1"), and "inverse_parentOf" ("r45") relationships. Additionally, there are about 5-15 inferred facts related to the unknown (to be predicted) rule.
> >
> > Note that a single rule can be equivalent to multiple rules. For example, the rule $\forall x,y,z: \text{parentOf}(x, y) \land \text{parentOf}(y, z) \land \text{gender}(x, \text{female}) \rightarrow \text{GrandmotherOf}(x,z)$ can be represented as two separate rules: $\forall x,y,z: \text{parentOf}(x, y) \land \text{parentOf}(y, z) \rightarrow \text{GrandparentOf}(x,z)$ and $\forall x,y,z: \text{GrandparentOf}(x,z) \land \text{gender}(x, \text{female}) \rightarrow \text{GrandmotherOf}(x,z)$. To simplify the induction evaluation process, we rewrite all logical rules by including only the "parentOf" and "gender" relations in the rule body. Doing so ensures that each inferred relation is implied by a single logical rule. We provide a rule template to define the rule search space, as detailed in Appendix A.2, and specific examples can be found in Appendix E.3.
> >
> > For abduction tasks, we created the prompts containing gender information and "parentOf" relationships, alongside logical rules consisting of "male", "female", and "parentOf". The dataset contains about 28 logical rules and 63 facts. We evaluate the abilities by generating explanations. Since there can be multiple valid explanations in abduction, as long as an LLMs output one possible interpretation, the reasoning is considered correct. Specific example are provided in Appendix E.1 and E.2.
> >
> > Further details about task descriptions are included in Appendix G.
> >
> >
> > > Q5: specific fine-tuning examples
> >
> > We have included specific fine-tuning examples in Appendix D of our paper.

---

> > > ### Comment · Reviewer_nbkm · 2023-11-23
> > >
> > > Thank you for the detailed response, and the pointers to the relevant examples in the appendices.
> > >
> > > About comparison with respect to the literature on content effects:
> > >
> > > > While they conclude that "Language models show human-like content effects on reasoning", this is only a subordinate conclusion that suggests LLMs are in-context semantic reasoners influenced by semantic associations. However, their conclusion does not directly imply that "LLMs are not symbolic reasoners," as suggested by Paradox 1: "LLMs are in-context semantic reasoners rather than symbolic reasoners".
> > >
> > > The fact that LLMs show content effects does imply that they are not symbolic reasoners (like people aren't). As you mentioned, symbolic reasoning "depends only on the form or the syntax of the premises and the conclusion, irrespective of the specific contents of this argument". Content effects are the negation of this: the prior work on this shows exactly that LLM reasoning does not "depend only on the form or the syntax of the premises and the conclusion". Therefore, LLMs are not symbolic reasoners. As such, this point (Paradox 1) is implied by prior work.
> > >
> > > After seeing the proper failure examples in Appendix E, I would not expect any LLM to succeed at this task in any case, at least not in this current format. The context is extremely long, and the LLM has to guess in one step what is the relevant rule and the list of facts (when it is predicting the token coming after `To prove the statement ..., we can use the rule `). This is a case where you'd certainly expect to need some trial-and-error and possible backtracking, including for a trained human. To argue that LLMs really cannot do this task at all, it would be more compelling to find much simpler examples where they still fail at it, not one that spans 2 pages.
> > >
> > > The clarifications alleviate my concerns about the novel rule generalization experiments to some extent (I was confused by the "5 training Symbolic Trees / 3 for testing" in Page 7). However, I do think the negative result from Figure 6 is missing some positive control to be convincing. From what we have observed in all the literature on training Transformers, we'd expect that you should see _some_ generalization to novel rules with enough training tasks and diversity, and if enough information is given in context, even if not nearly as strong as a symbolic reasoner. Having just one experiment where you fail to see any generalization (3.2.3 is quite short in detail) still leaves the doubt on whether the experiment makes sense. I strongly encourage the authors to find a positive control first, where varying a specific axis of the setup (e.g., perhaps the maximum depth of the logical rules) makes generalization harder. In such a case, we'd be more confident that we're seeing the influence of a particular variable on the results.
> > >
> > > For these reasons, I'd like to keep my original score.

---

### Official Review · Reviewer_Zwsu · 2023-11-10

**Soundness:** 3 good
**Presentation:** 3 good
**Contribution:** 2 fair
**Rating:** 3
**Confidence:** 3

**Summary:**

This paper studies whether the logical reasoning capability of large language model generalizes. They evaluate deductive, inductive, and abductive reasoning.

First, they replaced semantic words with pure symbols and found that LLMs perform much worse on the Symbolic Tree dataset which consists of family tree relations. In contrast, there's no drop for ProofWriter which consist of fictional facts and rules.

Second, they finetuned Llama2 on symbolic reasoning tasks from one domain (Symbolic Tree), which made the gap disappear in domain, but found that the finetuned model cannot generalize to other domains (ProofWriter, RuDaS, FOLIO).

They concluded that the reasoning abilities of LLMs were confounded by memorizing the semantics, and even if finetuned, it uses template matching instead of truly learning the rules.

**Strengths:**

1. The writing is relatively easy to understand.
2. There are a few interesting empirical findings from the carefully designed experiments, e.g.
(1) Finetuning on symbolic reasoning generalizes to unseen facts, but finetuning on semantics doesn't.
(2) Finetuning on symbolic reasoning can help with generalization in semantic reasoning.
3. The paper found previous works either focusing on a single domain or are confounded by semantics, and try to address their shortcomings.

**Weaknesses:**

1. I think the major weakness is the lack of novelty. Previous works [e.g. Saparov & He] already showed that semantics affects LLMs's reasoning ability, and that if we give new fact and rules contrary to the pretraining prior, the model struggles with learning those new rules. I think the main message of this paper is the same thing and not very new.
2. While I agree that looking at test performance on multiple OOD datasets is important, I hope the authors can explain more clearly whether the datasets contain the same logic rules as the training dataset (LogicTree). If they're different, why do we expect finetuning on LogicTree would generalize at all? Requiring the model to generalize to any novel symbolic rule OOD doesn't seem reasonable to me. Usually for domain generalization one has to specify the boundary of domains. Is this all first-order logic or propositional logic? The delineation seems unclear to me, and I'm not sure inductive reasoning is comparable to deductive reasoning, since we would also not want the model to learn spurious correlations in context. I think the authors should clarify the exact scope of expected generalization in mathematical language. For example, we may want to train on 2-hop but generalize to multi-hop problems, etc.
3. Some minor issues:
(1) All tables and plots: missing error bars
(2) Tables 4, 5, 6 can have more informative row names. The current row names are hard to parse.
(3) Table 6 is lacking context. What is the baseline we are comparing to?

**Questions:**

1. Table 1: Why does induction column miss some values?
2. LLMs can perform simple arithmetics and it couldn't have seen all possible additions / multiplications etc. during training. Doesn't this show it have some ability to learn rules beyond sematics?

---

> ### Author Response · Authors · 2023-11-17
> **Reply to Reviewer Zwsu (part 1/2)**
>
> Thank the reviewer for their comments.
>
>
> >  W1. lack of novelty
>
> We would like to address your concerns regarding the novelty of our work and stress how our findings on paradox 1 differ from the conclusions drawn in prior works. We will discuss paradox 2 subsequently.
>
> While Saparov and He (2022) conclude that "Language models show human-like content effects on reasoning", this is only a subordinate conclusion that suggests LLMs are in-context semantic reasoners influenced by semantic associations. However, their conclusion does not directly imply that "LLMs are not symbolic reasoners," as suggested by Paradox 1: "LLMs are in-context semantic reasoners rather than symbolic reasoners".
>
>
> According to the definition of deductive reasoning from Wikipedia (https://en.wikipedia.org/wiki/Deductive_reasoning), **logical reasoning should be formal and depends only on the form or the syntax of the premises and the conclusion, irrespective of the specific contents of this argument.** To this end, our task decouples semantics and focuses on pure symbolic reasoning (only provided syntax). When we feed these to LLMs, they show significantly worse performance compared to when normal semantic words are fed. This phenomenon indicates that LLMs fail to invoke the basic formal reasoning abilities of humans but instead rely on shallow semantic associations for prediction.
>
> Moreover, we conduct comprehensive experiments on various logical reasoning, including deduction, induction and abduction tasks and find that they perform notably worse in these types of reasoning tasks compared to deduction, regardless of whether semantics or symbols are used. When semantics are decoupled, the drop in performance is even more significant. These findings highlight the considerable room for improvement in LLMs' reasoning abilities and suggest that relying solely on semantics to achieve symbolic reasoning is challenging.
>
> [1]Saparov, Abulhair, and He He. "Language models are greedy reasoners: A systematic formal analysis of chain-of-thought." *arXiv preprint arXiv:2210.01240* (2022).
>
>
>
> > W2. concern about the generalization
>
>
> We would like to clarify that our paper focuses on first-order logical reasoning abilities, which will be emphasized in the revised version. Unlike the inference of machine learning which usually generalizes to the same distribution, an agent with "true" logical reasoning abilities can reason with any given information to solve new, unfamiliar problems, regardless of any prior knowledge or different knowledge representations. In other words, it can also generalize across facts, rules and representations (Sec 1 Line 13):
>
> 1. Facts: We test the same rules as in the training but with different facts (Sec 3.2.1 Line 4), as shown in the testing settings in Table 4.
> 2. Representations: Logical rule reasoning should be independent of different semantic content, including the rep_xx_with_xx examples in Table 4.
> 3. Rules: We test different first-order rules, such as training on $r1(A,B) \land r2(B) \rightarrow r3(B,A)$ and testing on $r1(A,B) \land r2(B,C) \rightarrow r3(C,B)$, as demonstrated in Fig 3. Different hop rules are also considered as different rules. Relevant settings include pf-sym-depth1, pf-sym-depth2, and Rudas in Table 5.
>
> Our comprehensive experimental results show that LLMs fine-tuned on symbolic logic datasets can achieve generalization in (1) and (2) to some extent with the help of template matching. However, they struggle to generalize to novel rules, emphasizing that fine-tuning LLMs does not achieve generalizable logical reasoning.
>
> > W3. some minor issues: (1) All tables and plots: missing error bars (2) Tables 4, 5, 6 can have more informative row names. The current row names are hard to parse. (3) Table 6 is lacking context. What is the baseline we are comparing to?
>
> Re W3 (1) and (2): We have included error bars and refined the row names in Tables 4, 5, and 6 to be more descriptive and concise in the revised version of our paper, with the error bars highlighted in red. Thank you for your suggestions.
>
> Re W3 (3): Table 6 represents the performance of RuDaS and FOLIO of the non-fine-tuned LLaMA2-13B-chat model. The results reveal that, although FOLIO's performance surpasses random, it remains inferior to the non-fine-tuned LLaMA2-13B-chat. This further emphasizes our finding that fine-tuned LLMs struggle to generalize to novel rules.

---

> ### Author Response · Authors · 2023-11-17
> **Reply to Reviewer Zwsu (part 2/2)**
>
> > Q1: Table 1: Why does induction column miss some values?
>
> This is because we did not perform a few-shot setting. Symbolic Tree dataset has a limited number of rules, and these rules exhibit a degree of interconnectedness. For instance, a rule like 'grandAuntOf' can be derived through the composition of rules like 'sisterOf' and 'grandparentOf'. The potential overlap and dependencies among the rules could skew the assessment of the model's true inductive reasoning capabilities. Consequently, we chose to only conduct a zero-shot setting where we provide facts to infer new rules without exposing the model to similar examples.
>
> > Q2: LLMs can perform simple arithmetics and it couldn't have seen all possible additions / multiplications etc. during training. Doesn't this show it have some ability to learn rules beyond sematics?
>
> The review's observation about LLMs performing simple arithmetic tasks raises an intriguing point about their ability to learn rules beyond semantics. However, our research suggests that this ability may not be as robust as it appears. The performance in simple arithmetic tasks could be attributed to data leakage or the application of pre-learned arithmetic solution templates, rather than a genuine understanding or mastery of arithmetic rules.
>
> For instance, when testing GPT-4 with a large number arithmetic problem, such as "128373 + 2879321873=", the model provided an incorrect answer (2879449246 instead of the correct answer 2879450246). This error indicates a limitation in the model's ability to process and apply arithmetic rules, especially in the context of large numbers. In other words, we think if it truly master arithmetics rules (carry or borrow operations), it should process large number. It suggests that while LLMs may display a superficial capability to perform arithmetic operations, they may not truly master the underlying rules.
>
> Therefore, while LLMs exhibit some level of proficiency in arithmetic, this should not be mistaken for a deep or comprehensive understanding of arithmetic rules. Our findings point towards a need for further development in logical reasoning area to enhance the true rule-learning capabilities of LLMs beyond semantic associations.

---

### Author Response · Authors · 2023-11-17
**Response to all reviewers**

We would like to highlight that the main contribution of our work:

We aim to investigate whether LLMs have strict formal reasoning abilities, or simply exploit the superficial semantic associations between questions and answers to make intuitive predictions as well as using template matching of the known knowlege, which is crucially important to AGI development. To answer the research question, we decouple semantics from the language reasoning process and force LLMs to perform symbolic reasoning to finish three kinds of reasoning tasks (deduction/induction/abduction). Furthermore, we also experiment with fine-tuning Llama-2 on pure symbolic reasoning tasks.

* We find that when provided with the syntax of the premises and the conclusion, pre-trained LLMs show significantly worse performance compared to when normal semantic words are fed. This phenomenon indicates that LLMs fail to invoke the basic formal reasoning abilities of humans but instead rely on shallow semantic associations for prediction.

* Moreover, we conduct comprehensive experiments on various logical reasoning, including deduction, induction and abduction tasks and find that they perform notably worse in these types of reasoning tasks compared to deduction, regardless of whether semantics or symbols are used. When semantics are decoupled, the drop in performance is even more significant. These findings highlight the considerable room for improvement in LLMs' reasoning abilities and suggest that relying solely on semantics to achieve symbolic reasoning is challenging.

* When fine-tuning on symbolic reasoning data, supervised fine-tuning considerably narrows the performance gap between symbolic and semantic settings, enabling LLMs to perform logical reasoning independent of semantics. However, when generalized to unseen logical rules, the fine-tuned model's performance significantly drops. This indicates that LLMs probably rely on a template-matching mechanism rather than a genuine formal reasoning process.

* In addition to the aforementioned findings, we have conducted numerous fine-grained experiments. These meticulous experimental results provide valuable insights and further strengthen our work.

---

### Meta-Review · Area_Chair_Sz9h · 2023-12-04

**Metareview:**

This paper studies the generalization of logical reasoning in large language models. They evaluate several types of reasoning, via experiments such as replacing meaningful words with abstract symbols (both for pretrained, and fine-tuned models). They conclude that even fine-tuned LLMs tend to use template matching instead of truly learning the logical rules.

Reviewers agreed that the question posed by this paper is important and timely.
However, most reviewers did not recommend acceptance. The primary issue is the novelty of the findings. Most reviewers felt the findings were not “paradoxes”, and the results were essentially implied by prior works on “context effects” in LLMs. Moreover, several methodological concerns were raised, where it was unclear if certain experimental results were statistically significant. Authors and reviewers were both engaged in the rebuttal period, but not all concerns were addressed.
Thus I must recommend rejection. I encourage the authors to consider the reviewers’ feedback for future revisions.

**Justification For Why Not Higher Score:**

Most reviewers raised concerns about the novelty and framing of this work, and their concerns were not adequately addressed during the rebuttal period.

**Justification For Why Not Lower Score:**

N/A

---

### Decision · Program_Chairs · 2024-01-16

Reject